# CINOC: Cardinality-Invariant Neural Operator Policies for Scalable PDE Control

**Pietro Zanotta** [* 1]   **Dibakar Roy Sarkar** [* 1]   **Honghui Zheng** [* 1]   **Somdatta Goswami** [1]   **Ján Drgoňa** [1]

## Abstract

Controlling partial differential equations (PDEs) with learning-based policies remains fundamentally limited by fixed-dimensional representations: policies trained for a specific sensor, actuator, or agent configuration typically fail when the configuration changes. This limitation is particularly severe in multi-agent PDE control, where policies do not scale across population sizes without retraining. We address this challenge by introducing **C**ardinality **I**nvariant **N**eural **O**perator **C**ontrol (**CINOC**), reformulating PDE control as an operator learning problem that maps state fields to continuous control functions and trains them end-to-end through differentiable PDE solvers, yielding policies that naturally adapt to varying sensor and actuator configurations. Remarkably, CINOC policies trained on small swarms exhibit cardinality invariance, allowing for zero-shot transfer to significantly larger populations as well as robustness to partial agent failure. This scalability arises from agents sharing a common policy and coordinating through their physical environment, which produces an emergent self-normalization effect. To explain this phenomenon, we provide a theorem grounded in mean-field theory demonstrating that policy gradients computed from finite-agent systems converge to those of a continuous control limit. Empirically, we validate CINOC on tracking, stabilization, and density transport across linear, nonlinear, chaotic, and turbulent PDEs.

## 1. Introduction

Partial differential equations (PDEs) govern a vast array of physical processes, from turbulent flows (Kalnay, 2003;

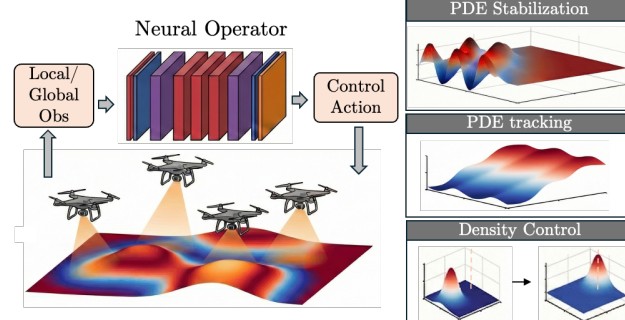

*Figure 1.* **Cardinality-Invariant Neural Operator Policy.** A shared neural operator maps PDE field observations to control actions for multiple agents, enabling zero-shot scalability across agent populations. Framework handles diverse tasks: (top right) chaos stabilization, (middle right) reference tracking, and (bottom right) density transport.

Pope, 2001) to heat transfer and chemical reactions (Crank, 1979; Turing, 1990). Reliable control of these systems is central to many engineering applications, yet remains challenging because PDEs evolve in infinite-dimensional function spaces and exhibit complex spatiotemporal dynamics. Classical control theory provides rigorous analytical frameworks, but these methods typically demand accurate system models, computationally intensive online optimization, or problem-specific designs that do not transfer across domains (Krstic & Smyshlyaev, 2008; Kouvaritakis & Cannon, 2016).

Learning-based approaches offer a promising alternative by approximating control policies from data, bypassing the need for explicit models or online solvers (Farahmand et al., 2017; Rabault et al., 2019; Bhan et al., 2024; Zhang et al., 2024; Becktepe et al., 2026). However, existing methods face a critical architectural bottleneck: they parameterize policies with discretized, fixed-dimensional representations. Consequently, any change in the number of sensors, actuators, or multi-agent swarm size generally requires the policy architecture to be redesigned and retrained from scratch.

We address these limitations by introducing **C**ardinality **I**nvariant **N**eural **O**perator **C**ontrol (**CINOC**), reformulating policy synthesis as an operator learning problem (Lu et al., 2021; Li et al., 2021). Our policy maps the current

---
[*]Equal contribution [1]Department of Civil and System Engineering, Johns Hopkins University, Baltimore, USA. Correspondence to: Pietro Zanotta <pzanott1@jh.edu>.

*Proceedings of the 43rd International Conference on Machine Learning*, Seoul, South Korea. PMLR 306, 2026. Copyright 2026 by the author(s).

PDE state field to a continuous control function over the spatial domain, trained end-to-end through differentiable PDE solvers (Holl et al., 2020). Figure 1 illustrates this framework, which accommodates fixed and mobile actuators (Demetriou, 2010), global and local sensor fields, and centralized- or decentralized agent configurations. In the decentralized setting, agents sharing a common policy coordinate implicitly through the PDE-governed environment. This stigmergic interaction (Theraulaz & Bonabeau, 1999) enables policies trained on small swarms to transfer zero-shot to larger populations, with graceful degradation under partial agent failures.

**Contributions:** We make the following contributions.

**(1) Neural operator framework for PDE control.** We introduce CINOC, which parameterizes PDE control policies as neural operators, yielding configuration-invariant policies

**(2) Differentiable-physics-based policy optimization.** We train control policies end-to-end by differentiating through PDE solvers, providing efficient and scalable policy gradients for a broad class of PDE control problems.

**(3) Cardinality-invariant multi-agent control.** A key discovery in this work is that CINOC's shared policies can induce complex multi-agent control via environment-mediated (stigmergic) coupling. We identify an emergent self-normalization phenomenon that enables cardinality-invariant behavior.

**(4) Mean-field analysis of policy gradient.** To ground these empirical findings, we provide a theoretical analysis linking finite-population gradient learning to a mean-field limit, leading to the cardinality invariance property.

**(5) Empirical evaluation across diverse PDE regimes.** We demonstrate stabilization, tracking, and density transport across diverse PDEs, including nonlinear reaction, diffusion, and chaotic spatiotemporal dynamics, validating scalability across sensing, actuation, and population configurations.

## 2. Related Work

**Classical PDE Control:** Classical methods face a fundamental tension between mathematical rigor and real-time feasibility. Model Predictive Control (MPC) is effective but computationally demanding (Kouvaritakis & Cannon, 2016), while backstepping offers stability guarantees yet is bottlenecked by complex kernel computations (Krstic & Smyshlyaev, 2008; Qi et al., 2024). Adjoint-based methods provide gradients but scale poorly with state dimension (Gunzburger, 2002). To address dimensionality, research has pursued finite-dimensional reductions (Lunasin & Titi, 2015) and optimal actuator/sensor placement (Morris, 2010; Guo & Yang, 2015; Privat et al., 2015). For parabolic systems, explicit stabilization with finitely many actuators

provides constructive guarantees (Kunisch & Rodrigues, 2019). Classical frameworks also address mobile actuator-sensor networks (Demetriou, 2010) and spatially scheduled actuation (Iftime & Demetriou, 2009). While powerful under complete model assumptions, these methods rely on centralized synthesis and fixed actuator configurations.

**RL-based PDE Control:** Reinforcement learning offers a model-free alternative, avoiding explicit system models or adjoint computations. Farahmand et al. (2017) introduced Deep Fitted Q-Iteration with transfer learning capabilities. Addressing high-dimensional action spaces, Pan et al. (2018) proposed action descriptors encoding spatial regularities, inspiring our operator-based formulation where policies output continuous control functions. Applications to fluid control validated this paradigm (Rabault et al., 2019; Pirmorad et al., 2021). Peitz et al. (2024) exploited translational equivariance for zero-shot domain transfer, demonstrating that architectural symmetries enable generalization. Integration with neural operators has enabled adaptive control (Hu et al., 2025). Standardized benchmarks like PDE Control Gym (Bhan et al., 2024) and Controlgym (Zhang et al., 2024) now provide reproducible testbeds.

Multi-agent reinforcement learning (MARL) (Lowe et al., 2017; Hüttenrauch et al., 2019) has been applied to distributed control problems. However, these methods face fundamental limitations for our setting: they require fixed state-action dimensions tied to a specific swarm size, precluding zero-shot transfer across population sizes, and rely on high-variance gradient estimators rather than exact gradients from differentiable physics. Although preliminary MARL studies (Guo et al., 2025; Bousias et al., 2025; Wang et al., 2025) exhibit zero-shot scalability, they achieve this via architectural mechanisms such as parameter sharing, GNNs, and equivariance constraints. In fact, while Permutation-Invariant MARL (PI-MARL) methods achieve variable-size scalability via attention, graph networks, or hypernetworks that embed explicit permutation invariance and equivariance (Jianye et al., 2022; Hu et al., 2021; Wen et al., 2022; Park et al., 2025), a direct empirical comparison is infeasible. Furthermore, while empirical demonstrations of cardinality invariance have recently emerged in fluid dynamics, such as the use of MARL for turbulent drag reduction in channel flows (Guastoni et al., 2023) and the control of three-dimensional Rayleigh-Bénard convection (Vasanth et al., 2025), these works treat the property primarily as an empirical consequence of the system's spatial translational invariance.

To rigorously evaluate our approach against standard continuous control paradigms, we adapted state-of-the-art single-agent and multi-agent reinforcement learning algorithms to our PDE environments. Complete architectural and algorithmic details for these baselines are deferred to Appendix G.

**Differentiable Physics for Control:** Differentiable physics engines enable backpropagation of control objectives through system dynamics, providing scalable gradient-based control across a wide range of applications (Qiao et al., 2020; de Avila Belbute-Peres et al., 2018). In the context of PDE control, Holl et al. (2020) demonstrated end-to-end training with differentiable solvers, ensuring strict physical consistency. Drgoňa et al. (2024) used this paradigm to learn policies for parametric MPC problems. Recent open-source libraries have further broadened accessibility to physics-based learning by providing tools for differentiable simulation and control (Murthy et al., 2021; Gradu et al., 2020; Drgona et al., 2023). These advances have enabled scalable gradient-based control in robotics, including control of quadcopters, locomotion, and contact-rich dynamics (Schwarke et al., 2024; Song et al., 2024; Pan et al., 2026). This paradigm forms the computational backbone of our training algorithm.

**Operator Learning for Control:** Neural operators (NOs), such as DeepONet and FNO (Lu et al., 2021; Li et al., 2021), learn mappings between infinite-dimensional function spaces. NOs can accelerate model-based controllers while preserving theoretical guarantees. Neural backstepping approximates parameter-to-gain kernel mappings (Bhan et al., 2023; Vazquez & Krstic, 2024; Qi et al., 2024; Lamarque et al., 2025), achieving substantial speedups while retaining stability certificates (Bhan et al., 2025; Abdolbaghi et al., 2025). This demonstrates operator learning can encode control-theoretic structures, motivating our investigation of policies as operators. Physics-informed approaches embed PDE residuals and Port-Hamiltonian structure (Li et al., 2025), while safety-critical extensions combine NOs with control barrier functions (Hu & Liu, 2024; Hu et al.).

NOs can serve as differentiable surrogates within optimization loops. Wang et al. (2021) compressed PDE-constrained optimization using self-supervised DeepONets, while de Jong et al. (2025) introduced multi-step DeepONet for MPC. Zhao et al. (2025) achieved drag reduction with zero-shot Reynolds number generalization. Hwang et al. (2022) employs NOs within PDE-constrained optimization, achieving substantial speedups. Fabiani et al. (2025) utilizes local NOs for equation-free control, while Lundqvist & Oliveira (2025) combines NOs for dynamics prediction with physics-informed loss terms and online learning. Similarly, Guven et al. (2025) use DeepOnet, while Sarkar et al. (2025) use time-integrated DeepONet for offline learning of neural policies. While these methods leverage NOs for dynamics representation, our approach instead places operator learning at the core of control synthesis, unifying policy parameterization and optimization within a single operator-learning framework.

The literature establishes powerful tools for PDE control.

However, a critical gap remains: existing approaches rely on fixed-dimensional state-action representations, preventing scalability to varying actuator counts without complete retraining. This architectural constraint also limits decentralized coordination when agents operate with only local observations. Our work addresses this gap by reformulating policy learning as an operator learning problem, directly parameterizing policies as neural operators mapping state fields to control functions. This formulation yields policies with emergent properties including cardinality invariance across actuator counts, effective decentralized coordination despite sparse local sensing, and mean-field consistent gradient learning in the large-population limit.

## 3. Problem Formulation

We consider the optimal control of a time-dependent state field $z(\boldsymbol{x}, t) \in \mathcal{Z}$ over a bounded spatial domain $\Omega \subset \mathbb{R}^d$ and a finite time horizon $t \in [0, T]$. The system is actuated by $M$ agents, where the $i$-th agent is located at $\boldsymbol{\xi}_i(t) \in \Omega$ and applies a localized control intensity $u_i(t) \in \mathbb{R}$. For mobile agents, the position evolves according to a controlled velocity $\boldsymbol{v}_i(t) \in \mathbb{R}^d$. Our goal is to synthesize joint control trajectories for the actions $\boldsymbol{u}^M(t) = \{u_1(t), \ldots, u_M(t)\}$ and velocities $\boldsymbol{v}^M(t) = \{\boldsymbol{v}_1(t), \ldots, \boldsymbol{v}_M(t)\}$ that minimize a cumulative cost subject to the governing PDE dynamics and physical constraints.

Let $\boldsymbol{\psi} \sim \mathcal{D}_{\text{prob}}$ represent a problem instance drawn from a distribution of physical parameters. Each instance $\boldsymbol{\psi} = (z_0, z_{\text{target}}, \boldsymbol{c}_{\text{env}})$ defines the initial condition $z_0(\boldsymbol{x})$, the target state $z_{\text{target}}(\boldsymbol{x})$, and environment-specific constraints $\boldsymbol{c}_{\text{env}}$ (e.g., static obstacles).

The general multi-agent PDE control problem is formulated as the problem of finding the optimal control variables to minimize the expected cost:

$$
\begin{aligned}
\min_{\boldsymbol{u}^M, \boldsymbol{v}^M} \mathcal{J} = \mathbb{E}_{\boldsymbol{\psi} \sim \mathcal{D}_{\text{prob}}} & \left[ \int_0^T \ell\big(z(\boldsymbol{x}, t), \boldsymbol{u}^M(t), \boldsymbol{v}^M(t), \boldsymbol{\psi}\big) dt \right] \\
\text{s.t.} \frac{\partial z}{\partial t} &= \mathcal{F}\big(z(\boldsymbol{x}, t); \boldsymbol{\psi}\big) + \sum_{i=1}^M \mathcal{B}\big(\boldsymbol{x}, u_i(t), \boldsymbol{\xi}_i(t)\big), \\
\dot{\boldsymbol{\xi}}_i(t) &= \boldsymbol{v}_i(t), \quad i = 1, \ldots, M, \\
\boldsymbol{c}\big(z(\boldsymbol{x}, t), & \boldsymbol{u}^M(t), \boldsymbol{v}^M(t), \boldsymbol{\xi}^M(t), \boldsymbol{\psi}\big) \leq 0.
\end{aligned}
\tag{1}
$$

Here, the loss functional $\ell$ maps the global state and control efforts to a scalar cost, penalizing deviations from $z_{\text{target}}$ and excessive energy expenditure. The nonlinear differential operator $\mathcal{F} : \mathcal{Z} \to \mathcal{Z}$ governs the intrinsic PDE dynamics. The actuation map $\mathcal{B} : \Omega \times \mathbb{R} \times \Omega \to \mathcal{Z}$ projects the scalar control input $u_i(t)$ into the spatial domain around the agent's current location $\boldsymbol{\xi}_i(t)$. For static agents, the kinematic constraint simply becomes $\dot{\boldsymbol{\xi}}_i(t) = \boldsymbol{0}$. Finally,

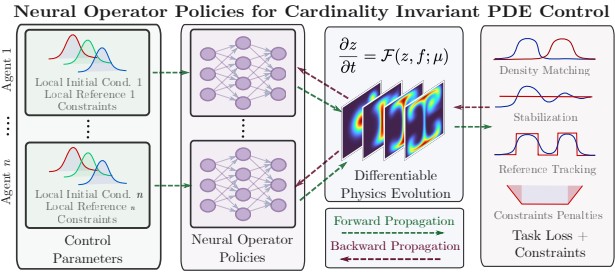

*Figure 2.* Schematic of the multi-agent operator learning for PDE control. The PDE solver acts as a differentiable layer. The operator policy $\mathcal{G}_{\boldsymbol{\theta}}$ (shared among the agents) takes error observations and outputs control actions. Gradients propagate through the solver to update $\boldsymbol{\theta}$.

the vector-valued function $\boldsymbol{c}$ enforces constraints such as maximum control amplitudes, kinematic limits, and spatial obstacle avoidance.

Throughout our analysis and empirical evaluations, we operate under the assumption that the underlying PDE systems are locally observable and controllable given the chosen agent densities, spatial distributions, and sensor radii. A dedicated discussion detailing how these physical constraints inform our framework's applicability and deployment is provided in Appendix D.7.

This formulation describes the fundamental continuous control problem independent of the chosen solution architecture. In Section 4, we introduce our specific methodology, which tackles this problem by parameterizing the decentralized control actions $[u_i(t), \boldsymbol{v}_i(t)]^\top$ as the output of a shared neural operator policy $\mathcal{G}_{\boldsymbol{\theta}}$ acting strictly on local field observations.

## 4. Methodology

We propose a scalable, self-supervised learning framework for PDE control that seamlessly integrates neural operators with differentiable physics. To enable cardinality invariance, the ability to control swarms of varying size $M$ with a single policy, we recast control synthesis as a shared operator learning problem in which all agents query a shared neural operator policy. As illustrated in Figure 2 and detailed in Algorithm 1, our method consists of four coupled differentiable components arranged in a closed loop: (i) an observation operator that samples and processes the PDE state within the field of view of each actuator; (ii) a neural operator policy that maps local observations and agent coordinates to control actions; (iii) a differentiable PDE solver that evolves the system state; and (iv) a learning objective that evaluates control performance and backpropagates gradients through the physics to update the policy parameters.

**Observation operator.** Agents in large scale systems typi-

cally lack global visibility; therefore, we employ an observation operator $\mathcal{O}bs : \mathcal{Z} \times \Omega \rightarrow \mathcal{Z}_{\text{local}}$ that subtracts $z_{\text{target}}$ from $z(\boldsymbol{x}, t)$ to form a task-relevant error representation, and extracts a local field of view $z_i^{\text{patch}}(t) \in \mathcal{Z}_{\text{local}}$ relative to the $i$-th agent's position $\boldsymbol{\xi}_i$, aligned with the PDE mesh:

$$z_i^{\text{patch}}(t) = \mathcal{O}bs\big(z(\boldsymbol{x}, t), \boldsymbol{\xi}_i(t), \boldsymbol{\psi}\big). \quad (2)$$

Processing the local field includes evaluating the pointwise error of the current and target states, as well as spatial derivatives computed on the mesh.

**Neural Operator Policy.** We define the policy $\mathcal{G}_{\boldsymbol{\theta}}$ as a mapping from the local observation space $\mathcal{Z}_{\text{local}}$ and local agent coordinates $\Omega$ to the control space $\mathcal{U}$, $\mathcal{G}_{\boldsymbol{\theta}} : \mathcal{Z}_{\text{local}} \times \Omega \rightarrow \mathcal{U}$.

Each agent $i$ queries this shared operator using its local observation $z_i^{\text{patch}}$ and coordinate $\boldsymbol{\xi}_i$. We parameterize $\mathcal{G}_{\boldsymbol{\theta}}$ using DeepONet (Lu et al., 2021), where a branch net $\text{Br}_{\boldsymbol{\theta}_2}$ encodes the local field and a trunk net $\text{Tr}_{\boldsymbol{\theta}_3}$ encodes the spatial coordinate. A final feed-forward net $\chi_{\boldsymbol{\theta}_1}$ fuses these representations to the control action:

$$u_i(t) = \mathcal{G}_{\boldsymbol{\theta}}(z_i^{\text{patch}}, \boldsymbol{\xi}_i) = \chi_{\boldsymbol{\theta}_1}\left(\text{Br}_{\boldsymbol{\theta}_2}(z_i^{\text{patch}}) \cdot \text{Tr}_{\boldsymbol{\theta}_3}(\boldsymbol{\xi}_i)\right). \quad (3)$$

Here $\cdot$ denotes the dot product. For kinematic actuators, the same operator can synthesize a composite action $[u_i(t), \boldsymbol{v}_i(t)]^\top$. Crucially, the parameters $\boldsymbol{\theta} = \{\boldsymbol{\theta}_1, \boldsymbol{\theta}_2, \boldsymbol{\theta}_3\}$ are shared across all $i = 1, \dots, M$ actuators, ensuring permutation invariance and constant input dimensionality regardless of $M$. Unlike standard MARL frameworks where agents often hold distinct roles, our actuators form a homogeneous swarm deployed to cooperatively solve a unified task. Parameter sharing is therefore a logical reflection of this physical homogeneity.

**Differentiable Physics and Actuation.** To train $\boldsymbol{\theta}$ without expert supervision, we embed the physical dynamics directly into the optimization loop using a Discretize-then-Optimize (DtO) approach. We define the solver $\Psi$ as the transition operator that explicitly solves the discretization form of the PDE dynamics $\mathcal{F}$ from Eq. 1:

$$z_{t+1}, \{\boldsymbol{\xi}_{i,t+1}\}_{i=1}^M = \Psi(z_t, \{\boldsymbol{\xi}_{i,t}\}_{i=1}^M, \boldsymbol{u}^M(t); \Delta t). \quad (4)$$

The solver $\Psi$ aggregates the individual control actions $\boldsymbol{u}^M(t)$ and advances the state field $z_{t+1}$. Simultaneously, it updates the actuator positions based on the agent type. For static actuators positions remain fixed, $\boldsymbol{\xi}_{i,t+1} = \boldsymbol{\xi}_{i,t}$. The policy outputs strictly forcing intensity $u_i$, while for kinetic actuators the policy outputs a composite vector $[u_i, \boldsymbol{v}_i]^\top$. The solver integrates the velocity $\boldsymbol{v}_i$ to update positions, $\boldsymbol{\xi}_{i,t+1} = \boldsymbol{\xi}_{i,t} + \boldsymbol{v}_i \Delta t$. Because $\Psi$ is implemented via automatic differentiation, it provides exact gradients $\partial z_{t+1}/\partial \boldsymbol{\theta}$, linking the system state directly to the policy actions. Notably, our formulation remains compatible with the optimize-then-discretize (OtD) framework (Ricky et al., 2018).

**Learning Objectives.** The training paradigm minimizes the expected cumulative cost over sampled problem instances $\psi$. The training loss function is explicitly parameterized by the swarm size $M$ to balance performance and total control effort. For trajectory tracking tasks, we minimize:

$$\mathcal{J}(\boldsymbol{\theta}) = \sum_{t=0}^{T} \left( \|z_t - z_{\text{target}}\|_{\mathcal{Z}}^2 + \lambda \|\boldsymbol{u}^M(t)\|^2 \right). \quad (5)$$

Here, the weight $\lambda$ penalizes the aggregate control energy. For density matching problems, the tracking error norm is replaced by a Moment Matching loss $\mathcal{W}(z_t, z_{\text{target}})$:

$$\mathcal{J}(\boldsymbol{\theta}) = \sum_{t=0}^{T} \left( \mathcal{W}(z_t, z_{\text{target}}) + \lambda \|\boldsymbol{u}^M(t)\|^2 \right). \quad (6)$$

The parameters $\boldsymbol{\theta}$ are updated via stochastic gradient descent, with gradients computed by backpropagating the loss through the physics solver $\Psi$ (BPTT) (Werbos, 1990).

---

**Algorithm 1** Self-Supervised Operator Policy Training

---

1: **Input:** distribution $\mathcal{D}_{\text{prob}}$, horizon $T$, number of agents $M$
2: **Initialize:** shared policy parameters $\boldsymbol{\theta}$
3: **while** not converged **do**
4:     {**1. Data Sampling**}
5:     Sample instances $\boldsymbol{\psi} = (z_0, z_{\text{target}}, \dots) \sim \mathcal{D}_{\text{prob}}$
6:     Initialize state $z_0$, actuator positions $\{\boldsymbol{\xi}_{i,t=0}\}_{i=1}^{M}$
7:     **for** $t = 0$ to $T$ **do**
8:         {**2. Shared Policy Inference**}
9:         **for** $i = 1$ to $M$ **do**
10:             $z_i^{\text{patch}} \leftarrow \mathcal{O}bs(z_t, \boldsymbol{\xi}_{i,t}, \boldsymbol{\psi})$
11:             $[u_i(t), \boldsymbol{v}_i]^\top \leftarrow \mathcal{G}_{\boldsymbol{\theta}}(z_i^{\text{patch}}, \boldsymbol{\xi}_{i,t})$
12:         **end for**
13:         {**3. Differentiable Physics Step**}
14:         $z_{t+1}, \{\boldsymbol{\xi}_{t+1}\} \leftarrow \Psi(z_t, \{\boldsymbol{\xi}_t\}, \boldsymbol{u}_t^M)$
15:         {**4. Loss**}
16:         $\mathcal{J} \leftarrow \mathcal{J} + \ell(z_{t+1}, \boldsymbol{u}_t^M, \boldsymbol{\psi})$
17:     **end for**
18:     **Update:** $\boldsymbol{\theta} \leftarrow \boldsymbol{\theta} - \eta \nabla_{\boldsymbol{\theta}} \mathcal{J}$     {Backprop via Physics}
19: **end while**

---

## 5. Theoretical Analysis

In this section, we provide a theoretical foundation for the convergence and scalability of our multi-agent framework. We focus on two key questions: (1) Does training on a finite swarm of size $M$ optimize the underlying continuous control problem? and (2) What mechanism allows a policy trained on one swarm size to transfer zero-shot to another?

To establish convergence and scalability, we analyze the system in the large-population limit ($M \to \infty$). For readers less familiar with the functional analytic setting (e.g.,

Gelfand triples, Aubin-Lions lemma (Simon, 1986)) and the mean-field measure convergence concepts utilized in these proofs, we provide a detailed primer in Appendix A.

### 5.1. Mean-Field Gradient Consistency

To ensure CINOC is not simply overfitting to a specific swarm configuration, we analyze the system behavior in the large-population limit ($M \to \infty$). Intuitively, as the number of agents increases, the discrete collection of actuators converges to a continuous actuator density measure. By employing mean-field theory, we define a limiting mean-field forcing operator $\mathcal{B}_\infty(z)$ that acts on this density rather than on discrete agent positions.

Our primary theoretical result establishes that the gradients computed during our discrete multi-agent training are consistent estimators of the gradients for this continuous limit.

**Theorem 5.1** (Consistency of Discrete Policy Gradients (Informal))**.** *As the number of agents $M \to \infty$, the discrete policy gradient $\nabla_{\boldsymbol{\theta}} \mathcal{J}_M$ converges to the mean-field gradient $\mathcal{D}_{\boldsymbol{\theta}} \mathcal{J}_\infty$. (Formal statement and proof in Appendix B).*

*Proof sketch.* The proof relies on the discrete physics uniformly approximates the mean-field physics.

**Uniform Bounds**: We first show that the PDE state and adjoint variables remain uniformly bounded in suitable function spaces regardless of $M$, preventing physical blow-ups as agents are added.

**Compactness & Convergence**: Using the Aubin-Lions lemma (Simon, 1986), we extract strongly convergent subsequences of the state trajectories, showing that the discrete dynamics converge to a unique mean-field limit.

**Gradient Identification**: Finally, we decompose the gradient error into a state-dependent term and a measure-dependent term. We show both vanish in the limit, proving that optimizing the swarm is asymptotically equivalent to optimizing the continuous operator. $\square$

**Significance**: Theorem 5.1 guarantees that our policies learn the fundamental operator dynamics of the PDE rather than becoming dependent on a specific cardinality. This consistency is the theoretical bedrock that allows the learned operator $\mathcal{G}_{\boldsymbol{\theta}}$ to function effectively across different swarm cardinalities.

### 5.2. Conjecture: Emergent Self-Normalization

While Theorem 5.1 assumes normalized forcing ($1/M$), our implementation uses unnormalized superposition ($\sum u_i$). We conjecture that the optimization process induces an implicit normalization via the effort regularization term $\lambda_u$, therefore extending the Theorem 5.1 to the unnormalized

case.

*Conjecture* 1 (Self-Normalization via Effort Regularization). The optimization process induces an implicit scaling where the optimal policy outputs intensities $u_i \approx O(1/M)$. This ensures the total forcing remains bounded while individual effort vanishes.

**Intuition**: This scaling represents a form of cooperative efficiency gain; the solver seeks an equilibrium between minimizing tracking error (which requires a certain total aggregate force) and minimizing control effort (which pushes $u_i \to 0$ for all agents). Because the DeepONet policy shares parameters between all agents and closes the loop through the global physical field $z$, this scaling is discovered sequentially; agents sense the magnitude of the aggregate field and adjust their output accordingly, without ever needing to communicate the total population count M. This mechanism is a key driver of the cardinality invariance observed in our experiments.

## 6. Experiment Results and Discussion

To validate the multi-agent operator learning framework, we analyze the learned policies across four dimensions: (1) control efficacy across diverse set of 1D and 2D PDEs (linear, nonlinear, turbulent, and chaotic dynamics), (2) versatility across distinct control objectives, including stabilization, trajectory tracking, and density transportation, (3) comparative performance against standard single-agent and multi-agent reinforcement learning baselines (Papoudakis et al., 2020; Fujimoto et al., 2018; Yu et al., 2022; Schulman et al., 2017), and (4) invariance to swarm sizes cardinalities. Table 1 details the experimental configurations and reports the average performance of the learned policy alongside the evaluated baselines on all test cases, categorizing each PDE by its dimensionality, control objective, and actuator dynamics.

Details on governing equations, simulation parameters, and further results are provided in Appendix E. The code to reproduce all experiments is available on Github[1]. Furthermore, to contextualize these data-driven approaches against classical optimal control, we benchmarked a Nonlinear Model Predictive Control (NMPC) (Allgöwer & Zheng, 2012) baseline on the 1D KS stabilization task. While theoretically sound, NMPC suffers from severe computational bottlenecks (requiring over 590 ms per optimization step), making it computationally intractable for real-time deployment or scaling to high-dimensional 2D domains. Our neural operator framework bypasses this limitation by shifting the computational burden to the offline training phase (see Appendix G.10 for detailed NMPC timing and perfor-

---

[1] https://github.com/SOLARIS-JHU/CINOC, while complete architectural and algorithmic details for the RL baselines are deferred to Appendix G

mance).

*Table 1.* **Multi-Agent Control Performance.** A distilled comparison of CINOC against uncontrolled evolution, DPC, and the best-performing Reinforcement Learning (RL) baseline for each specific task. The "Best RL" column reports the strongest result among PPO, TD3, MAPPO, and MATD3 (MAPPO for FKPP/Heat/KS tasks, MATD3 for Turbulence, PPO for Density). Comprehensive results for all baselines are provided in Appendix X. Values represent Tracking MSE, $L^2$ Energy/Enstrophy, and Moment Matching Loss (MML); lower is better. See Table 2 for comprehensive baselines.

| Environment | Unctrl. | DPC | Best RL | CINOC |
|---|---|---|---|---|
| *Tracking (MSE)* | | | | |
| FKPP 1D | $0.103 \pm 0.128$ | $1.4e\text{-}4 \pm 1.9e\text{-}4$ | $0.004 \pm 0.010$ | **$4.6e\text{-}5 \pm 7.5e\text{-}5$** |
| Heat 1D | $0.390 \pm 1.144$ | **$7.9e\text{-}5 \pm 1.1e\text{-}4$** | $0.001 \pm 0.002$ | $2.9e\text{-}4 \pm 4.3e\text{-}4$ |
| Heat 2D | $0.176 \pm 0.378$ | $2.1e\text{-}4 \pm 2.2e\text{-}4$ | $0.006 \pm 0.016$ | **$1.5e\text{-}4 \pm 1.8e\text{-}4$** |
| Heat 2D (Obs) | $0.176 \pm 0.378$ | $2.7e\text{-}4 \pm 2.6e\text{-}4$ | $0.004 \pm 0.015$ | **$1.2e\text{-}4 \pm 1.7e\text{-}4$** |
| *Stabilization (Energy/Enstrophy)* | | | | |
| KS 1D | $1.45 \pm 0.69$ | $0.16 \pm 0.15$ | **$0.13 \pm 0.13$** | $0.13 \pm 0.12$ |
| KS 2D | $13.91 \pm 4.61$ | $9.74 \pm 2.32$ | $0.50 \pm 0.24$ | **$0.37 \pm 0.19$** |
| Turbulence 2D | $50.49 \pm 13.29$ | $3.28 \pm 2.19$ | $3.71 \pm 2.46$ | **$3.23 \pm 2.22$** |
| *Density Matching (MML)* | | | | |
| Density Control | $0.027 \pm 0.026$ | **$0.002 \pm 0.010$** | $0.005 \pm 0.005$ | $0.002 \pm 0.002$ |

### 6.1. Stabilizing Spatiotemporal Chaos

**Task Setup.** We first challenge the policy with the 2D Kuramoto-Sivashinsky (KS) equation, a canonical model for pattern formation that exhibits flame-front instabilities and negative diffusion. The system is governed by high-order spatial derivatives that drive energy production at large scales, which is then dissipated at small scales, resulting in fully developed spatiotemporal chaos. We employ a uniform grid of 100 fixed actuators to provide localized control inputs. The goal is to drive the system from a chaotic state to the zero equilibrium $u(\boldsymbol{x}, t) = 0$. This equilibrium is linearly unstable; infinitesimal perturbations in the uncontrolled dynamics grow exponentially until the system returns to the chaotic attractor. This makes the task a challenging test of the policy's ability to manage non-linear amplification and precisely balance energy production with dissipative control.

**Result Evaluation.** Figure 3 illustrates the stabilization performance. The system is evolved to the chaotic attractor (phase $t < 0$) to ensure the policy confronts a realistic, complex state. Upon activating the control at $t = 0$, we observe a rapid suppression of the turbulent structures. While the uncontrolled baseline (Natural Evolution) continues to exhibit cellular fluctuations with magnitudes spanning approximately $[-9.3, 7.6]$, the learned policy induces an exponential decay. By $t = 5.0s$, the policy has effectively eliminated the chaotic shedding, reducing the field to a near-zero state and maintaining this unstable equilibrium despite the system's inherent drive toward instability. Compared to uncontrolled evolution, resulting in an energy level of 12.16, the controlled evolution is able to drive the energy to 0.20.

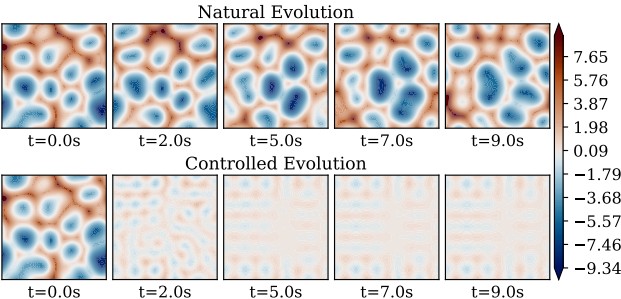

*Figure 3.* **Stabilization of 2D Kuramoto-Sivashinsky chaos.** Top: Natural (uncontrolled) evolution of $u(\boldsymbol{x}, t)$ showing persistent chaotic dynamics. Bottom: Controlled evolution of $u(\boldsymbol{x}, t)$ with policy applied at $t = 0$. The system, initially in a developed chaotic state, rapidly converges to the zero equilibrium within approximately 5 seconds.

Furthermore, as detailed in Table 1, CINOC significantly outperforms both DPC and all RL baselines on the KS 2D task. While DPC struggles to stabilize the system (achieving an average energy of $9.74 \pm 2.32$) and the best RL baseline (MAPPO) reaches $0.50 \pm 0.24$, our policy achieves the lowest energy state of $0.37 \pm 0.19$.

### 6.2. Trajectory Tracking with Actuator Constraints

**Task Setup.** To evaluate the policy's ability to handle non-convex spatial constraints in the multi-agent setting with kinetic actuators, we consider a 2D diffusion control task with static "keep-out" zones and we introduce $K = 3$ circular obstacles $\mathcal{O} \subset \Omega$. We employ 16 kinetic actuators so that the policy outputs, for each agent $i$, a velocity $\boldsymbol{v}_i = \dot{\boldsymbol{\xi}}_i$ governing agent motion and a scalar control intensity $u_i$ for local actuation. The task for the policy drives the system's spatial state along a time-parameterized path, ensuring the continuous evolution of the shape from its initial configuration to a desired terminal state. We also impose kinetic and actuator constraints to better reflect real-world application. The challenge for this example lies in the geometric constraints given by the ostacles: while the physical temperature field, governed by a 2d Heat Equation, diffuses freely through these regions, the actuators are strictly barred from entering them. The policy must therefore sink heat from within the forbidden zones by optimally positioning agents along the permissible boundaries (the hull of $\mathcal{O}$), without having actuators getting too close to the obstacle itself:

$$\min_k \|\boldsymbol{\xi}_i - \boldsymbol{c}_k\| \geq R_{\text{safe}} + r_k. \tag{7}$$

**Results Evaluation.** Figure 4 visualizes the emergent co-ordination strategies under spatial constraints. At $t = 30$ and $t = 80$, rather than forming perimeters around the obstacles, agents cluster within the high-magnitude regions of the temperature field. This suggests the policy prioritizes direct intervention in areas of maximum deviation.

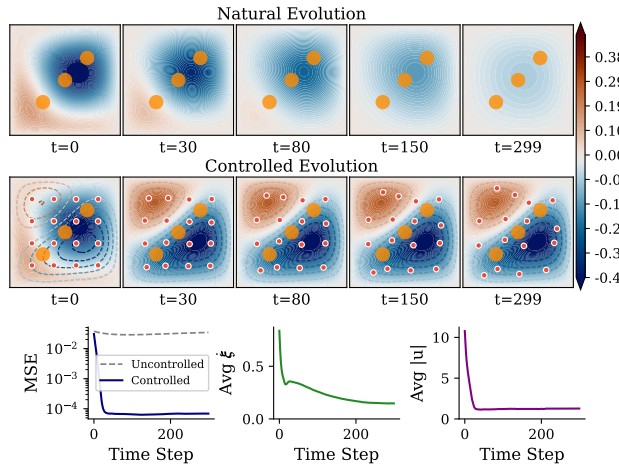

*Figure 4.* **Control under Actuator Constraints.** (Top) Uncontrolled thermal field evolution. (Middle) Controlled evolution where agents (dots) regulate the temperature field toward dashed reference contours while avoiding static obstacles (orange circles). (Bottom) Performance metrics: (left) MSE for controlled vs. uncontrolled convergence; (center) average velocity $\dot{\boldsymbol{\xi}}_i$ of the actuators; (right) average control intensity $|u_i|$.

Despite the presence of three large exclusion zones (orange circles), the bottom-left panel shows the Mean Squared Error (MSE) converging rapidly, reaching a steady state of $\text{MSE} \approx 8 \times 10^{-4}$, significantly lower than the uncontrolled baseline ($MSE \approx 10^{-2}$). The control effort, (Avg $|u_i|$), spikes initially to move agents into position and then stabilizes, indicating that the policy maintains the desired thermal state with minimal energy expenditure once the spatial configuration is established. Quantitatively, as shown in Table 1 for the obstacle-constrained Heat 2D environment, CINOC achieves an MSE of $1.2e\text{-}4$, outperforming the DPC baseline ($2.7e\text{-}4$) and drastically surpassing the best multi-agent RL algorithm, MAPPO ($0.004$). This highlights the advantage of CINOC over both traditional differentiable physics and trial-and-error learning in geometrically constrained tracking scenarios, a trend also reflected in the unconstrained 1D and 2D Heat tasks where CINOC consistently outperforms RL baselines by orders of magnitude. Crucially, while MARL baselines prove capable of managing the restorative forcing required for stabilization, the precise spatiotemporal coordination demanded by trajectory tracking exposes a massive performance gap where CINOC's operator-driven formulation demonstrates a clear superiority.

### 6.3. Density Transport via Flow Manipulation

**Task Setup.** While the previous tasks focused on suppressing instability and trajectory tracking, we now evaluate the policy's capacity for constructive pattern formation. The objective is to transport a passive scalar distribution to match a target configuration $\rho^*$ while strictly conserving mass (details in Appendix E.3), while subject to the 2d Advection-

**Diffusion equation.** This task introduces a specific optimization challenge: when the support of the current density and the target are disjoint, standard pointwise metrics (e.g., MSE, KL-divergence) yield vanishing gradients. We address this by optimizing the Wasserstein distance inspired moment matching (1st and 2nd moment), which leverages the underlying metric geometry to provide informative transport gradients even in the absence of spatial overlap. We employ 9 kinetic agents for this task to influence the field by controlling the local velocity term $\mathbf{v}_i$ rather than using direct mass forcing. Physically, this corresponds to advecting the density mass across the domain rather than locally creating or destroying it.

**Results Evaluation.** Figure 5 showcases the policy controlling $M = 9$ mobile advection agents. The results highlight a sophisticated two-phase strategy: in the initial phase ($t < 75$), agents do not merely push the density from behind. Instead, they organize into a semi-circular "herding" formation (visible at $t = 30, t = 75$). This shape serves a dual purpose: it maximizes the advective velocity toward the target while simultaneously acting as a barrier to prevent the density from diffusing outward. Once the target is reached (see Tracking Error, bottom left), the policy undergoes a phase transition. At $t \approx 75$, the control input (purple line) drops from saturation levels to a low-energy maintenance mode. The agents effectively switch from transporting to station-keeping, applying only minimal actuation required to counteract natural diffusion and hold the shape in place. As reported in Table 1, our approach matches the DPC baseline with a loss of 0.002, while significantly outperforming all standard and multi-agent RL baselines, which fail to achieve a loss lower than 0.005 (PPO) and struggle up to 0.021 (TD3 and MATD3).

## 6.4. Verification of Cardinality Invariance

A central claim of CINOC is the ability to train on $M_{train}$ agents and deploy on $M_{test} \neq M_{train}$ without fine-tuning. We validate the cardinality invariance property (Sec. B.2) by training a single policy for each of our numerical experiments and deploying it zero-shot on different swarm cardinalities. Figure 6 visualizes these results, while Table 3 reports the maximum and minimum parameter counts that keep the error bounded below an arbitrary 250% threshold.

We observe a distinct dimensionality effect where 2D policies generally exhibit broader scaling margins than their 1D counterparts. This stems from the topology of actuator density: while inter-agent distance decays linearly ($M^{-1}$) in 1D, it decays as $M^{-1/2}$ in 2D. Consequently, 1D domains suffer from rapid kernel saturation, whereas 2D domains offer greater geometric capacity to accommodate additional agents without forcing conflicts. Notably, the Density 2D task achieves the highest scalability (up to $64\times$) because the

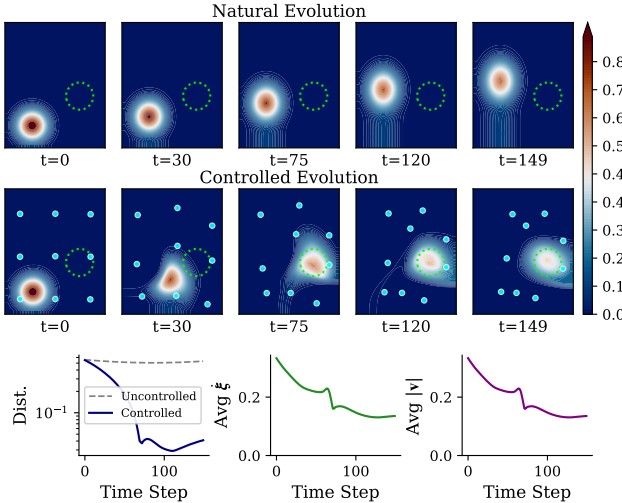

*Figure 5.* **Density Transport via Mobile Advection.** (Top) Uncontrolled evolution of density. (Middle) Controlled evolution where agents (dots) coordinate to "blow" the density blob into the target zone (green circle). (Bottom) Performance metrics: (left) Wasserstein distance showing controlled vs. uncontrolled convergence; (center) average velocity $\dot{\boldsymbol{\xi}}_i$ of the actuators; (right) average control action $|\mathbf{v}_i|$.

policy learns a geometric "containment" strategy around the target zone, rather than applying continuous high-magnitude forcing. This station-keeping behavior allows excess agents to simply join the perimeter without disrupting the physics.

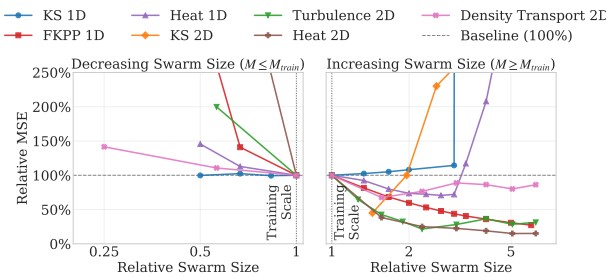

*Figure 6.* **Cardinality invariance across diverse PDE benchmarks.** The plots show the Relative MSE as a function of the swarm size relative to the original training scale. **Left:** Performance regimes for decreasing swarm sizes ($M \leq M_{train}$). **Right:** Performance regimes for increasing swarm sizes ($M \geq M_{train}$). The 100% dashed line indicates the baseline performance at the training population. Policies deployed in 2D environments (e.g., Density 2D, Turbulence 2D) exhibit significantly broader scaling margins than 1D counterparts due to higher geometric capacity and slower inter-agent distance decay. Conversely, chaotic regimes like the Kuramoto-Sivashinsky (KS) equation display the tightest stability margins as minor deviations in the aggregate forcing field are rapidly amplified.

Conversely, chaotic systems like the Kuramoto-Sivashinsky equation display the tightest stability margins (e.g., KS 2D transfers only up to $1.3\times$). Even minor deviations in the aggregate forcing field, introduced by the training-test density

mismatch, are rapidly amplified by the non-linear dynamics. Finally, we note that trajectory-tracking tasks (Fisher-KPP, Heat) generally support lower maximum cardinalities than pure transport. The tracking objective requires precise, localized destructive interference to reshape the state dynamically, a complex balance that is more sensitive to overcrowding.

Last, to demonstrate that this cardinality invariance is an intrinsic property of the operator learning formulation rather than an artifact of the differentiable physics training, we parametrize our policies using DeepONet and train them via standard model-free MARL algorithms. Complete architectural and algorithmic details for these RL experiments are provided in Appendix D.6.

### 6.5. Ablation Studies

We evaluate the robustness of CINOC through extensive ablations on solver fidelity, signal noise, sensor dimension and PDE parameters mismatch. Our results confirm that the proposed effort regularization is critical for emergent self-normalization, while restricting agents to local observation windows is necessary to prevent instability in large swarms. Furthermore, the learned policies exhibit strong robustness to changes in grid resolution and graceful degradation to changes in physical coefficients. We also highlight the role of noise injection during training serving as a key regularizer for cardinality invariance. More details about the ablations in Appendix 6.5.

## 7. Conclusion

In this work, we presented a scalable framework for the control of partial differential equations (PDEs) that integrates neural operators with differentiable physics. By reformulating policy learning as an operator learning problem, we overcame the fixed-dimensional constraints of traditional reinforcement learning approaches, enabling cardinality-invariant policies.

Theoretically, we grounded this capability in a mean-field limit analysis, proving that discrete policy gradients yield consistent estimators of the underlying continuous operator gradients. Furthermore, we identified an emergent self-normalization phenomenon driven by effort regularization, where individual agents automatically scale their output to maintain bounded aggregate forcing as the population grow.

Empirically, the framework proved robust across a diverse suite of linear, nonlinear, and chaotic PDE regimes and control task, including the Kuramoto-Sivashinsky equation and turbulent flows. Crucially, our extensive ablation studies confirmed that these policies remain robust to partial agent failure, sensor noise, grid resolution changes, and small parametric shifts in the underlying physics.

While our framework demonstrates strong performance, it is not without limitations. First, our training pipeline implies a model-based setting, relying on access to a fully differentiable solver and explicit knowledge of the governing equations. Second, while empirically robust on nonlinear systems, our theoretical convergence guarantees are strictly established only for linear, dissipative regimes. Future work will aim to relax these constraints by extending the theoretical analysis to broader classes of nonlinear PDEs. Additionally, we plan to explore training parametric operators conditioned on physical coefficients, enabling zero-shot generalization across diverse dynamical systems. Finally, we aim to scale this framework to high-dimensional, industrially relevant environments, such as 3D turbulent channel flows, to further bridge the gap between data-driven learning and rigorous control theory.

## Acknowledgments

This research is partially supported by the U.S. DOE, Office of Science, ASCR program under the Scientific Discovery through Advanced Computing (SciDAC) Institute "LEADS: LEarning-Accelerated Domain Science", and also partially supported by the Ralph O'Connor Sustainable Energy Institute (ROSEI) at Johns Hopkins University. The authors would like to acknowledge computing support provided by the Advanced Research Computing at Hopkins (ARCH) core facility at Johns Hopkins University and the Rockfish cluster. ARCH core facility (rockfish.jhu.edu) is supported by the National Science Foundation (NSF) grant number OAC1920103. The research efforts of DRS and SG are supported by the National Science Foundation (NSF) under Grant No. 2436738.

## Impact Statement

This paper presents work whose goal is to advance the field of machine learning. There are many potential societal consequences of our work, none of which we feel must be specifically highlighted here.

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

# A. Mathematical Preliminaries

To formalize the multi-agent control problem and subsequently analyze its large-population limit in our theoretical proofs, we rely on foundational concepts from infinite-dimensional functional analysis and mean-field theory.

## A.1. Functional Spaces and PDE Dynamics

We consider the state of the PDE evolving in infinite-dimensional function spaces. Let $H = L^2(\Omega)$ denote the standard Hilbert space of square-integrable functions over the spatial domain $\Omega$, and $V = H_0^1(\Omega)$ denote the Sobolev space of $H^1$ functions whose trace vanishes on the boundary $\partial\Omega$. Physically, these spaces capture essential properties of the system: $H$ typically represents states with finite total energy, while $V$ ensures the field is sufficiently regular, meaning its gradient is square-integrable, and properly respects the boundary conditions. These spaces form a Gelfand triple $V \hookrightarrow H \hookrightarrow V^*$, where $V^*$ is the dual space of $V$, and the embeddings are continuous and dense.

The evolution of the system's state $z(\boldsymbol{x}, t)$ over a time horizon $T$ is analyzed within Sobolev-Bochner spaces, such as $L^2(0, T; V)$. A crucial tool for our theoretical guarantees is the Aubin-Lions lemma. In finite-dimensional spaces, bounded sequences naturally have convergent subsequences, but this is not automatically true in infinite dimensions, and the Aubin-Lions lemma provides the necessary compactness criteria to bridge this gap. For our framework, it guarantees that as we scale the number of agents, the sequence of discrete PDE state trajectories remains well-behaved and does not exhibit infinitely fast, unbounded oscillations. This allows us to extract strongly convergent subsequences, which is a foundational step for analyzing nonlinear evolution equations and proving the existence of unique continuum limits of our discretized system (Alphonse et al., 2023).

## A.2. Mean-Field Theory and Measure Convergence

In our multi-agent configurations, the aggregate spatial influence of $M$ discrete agents located at positions $\boldsymbol{\xi}_i(t)$ can be represented by the empirical measure $\mu_M(t) = \frac{1}{M} \sum_{i=1}^{M} \delta_{\xi_i(t)}$.

To establish the cardinality invariance of our learned policies, we study the system in the mean-field limit as the number of agents $M \to \infty$. In this regime, discrete interacting particle systems can be shown to converge to continuous macroscopic limits (Golse, 2003). This transition from discrete particle descriptions to continuous integral equations relies on the weak-* convergence of the empirical measures $\mu_M(t)$ to a limit probability measure $\mu(\cdot, t)$ (Carrillo & Choi, 2021). We rely on weak-* convergence because discrete particles (modeled as Dirac deltas) cannot converge to a continuous density function under standard norms (such as strong $L^p$ norms or total variation distance). Instead of tracking individual particle paths, weak-* convergence ensures that the *overall integrated influence* of the swarm on the PDE environment admits a well-defined continuum limit as the population scales. This convergence guarantees that the discrete forcing field applied by a finite swarm smoothly transitions into an integral operator defined over the continuous agent density. By establishing this, we can prove that gradients computed during discrete multi-agent training are consistent estimators of the gradients for the continuous control limit.

# B. Full Theoretical Proofs

## B.1. Mean-Field Gradient Consistency

To study the behavior of the system as $M \to \infty$, we establish the functional analytic setting. Let $H = L^2(\Omega)$ and $V = H_0^1(\Omega)$. While CINOC applies to general nonlinear operators $\mathcal{F}$, for the theoretical analysis we restrict our attention to linear, strictly dissipative operators $\mathcal{A}$ to establish rigorous convergence guarantees. We therefore consider the PDE dynamics governed by a linear operator $\mathcal{A} : V \to V^*$ that is strictly dissipative, i.e., $\langle \mathcal{A}z, z \rangle \leq -\alpha \|z\|_V^2$ for some $\alpha > 0$. We define the parabolic regularity constant $C_T := \sup_{t \in [0,T]} \int_0^t \|S(t-s)\|_{\mathcal{L}(H)} \, ds < \infty$, where $\{S(t)\}_{t \geq 0}$ is the analytic $C_0$-semigroup generated by $\mathcal{A}$.

To simplify the following mean-field derivation without loss of generality, we treat the policy $\mathcal{G}_{\boldsymbol{\theta}}$ as a scalar mapping to the control intensities $u$, noting that the convergence of the velocity field $v$ follows an analogous analytic structure. To make the notation lighter, we use assume a continuous observation operator with a FOV of 100% of the field, meaning $\mathcal{Z}_{\text{local}} \equiv H$.

**The Normalized Limit.** To analyze convergence, we define a theoretical *normalized forcing term* operator $\mathcal{B}_M$ acting on

the empirical measure $\mu_M(t) = \frac{1}{M} \sum_{i=1}^{M} \delta_{\boldsymbol{\xi}_i(t)}$:

$$\mathcal{B}_M(z)(\boldsymbol{x}, t) = \frac{1}{M} \sum_{i=1}^{M} b(\boldsymbol{x}, \boldsymbol{\xi}_i(t)) \mathcal{G}_{\boldsymbol{\theta}}(z(\cdot, t), \boldsymbol{\xi}_i(t))$$

$$= \int_{\Omega} b(\boldsymbol{x}, \boldsymbol{y}) \mathcal{G}_{\boldsymbol{\theta}}(z, \boldsymbol{y}) \, d\mu_M(\boldsymbol{y}). \tag{8}$$

In the limit $M \to \infty$, we define the mean-field forcing operator:

$$\mathcal{B}_{\infty}(z)(\boldsymbol{x}, t) = \int_{\Omega} b(\boldsymbol{x}, \boldsymbol{y}) \mathcal{G}_{\boldsymbol{\theta}}(z, \boldsymbol{y}) \, \mu(\boldsymbol{y}, t) \, d\boldsymbol{y}, \tag{9}$$

where $\mu \in L^{\infty}(\Omega \times [0, T])$ is the limiting agent density. The associated mean-field gradient $\mathcal{D}_{\boldsymbol{\theta}} \mathcal{J}_{\infty}$ represents the sensitivity of the limit functional to the parameters $\boldsymbol{\theta}$:

$$\mathcal{D}_{\boldsymbol{\theta}} \mathcal{J}_{\infty} = \int_0^T \int_{\Omega} p_{\infty}(\boldsymbol{x}, t) \, \nabla_{\boldsymbol{\theta}} \mathcal{G}_{\boldsymbol{\theta}}(z_{\infty}, \boldsymbol{y}) \cdot b(\boldsymbol{x}, \boldsymbol{y}) \, \mu(\boldsymbol{y}, t) \, d\boldsymbol{y} \, d\boldsymbol{x} \, dt, \tag{10}$$

where $p_{\infty}$ is the solution to the continuous adjoint equation.

**Theorem B.1** (Consistency of Discrete Policy Gradients). *Let the initial state $z_0 \in H$ be independent of $M$. Assume:*

**(H1) Measure Convergence:** $\mu_M(t) \rightharpoonup^* \mu(\cdot, t) \, dy$ *weakly-\* in $\mathcal{M}(\Omega)$ for a.e. $t \in [0, T]$, with $\mu \in L^{\infty}(\Omega \times [0, T])$.*

**(H2) Policy Regularity:** $\mathcal{G}_{\boldsymbol{\theta}} : H \times \Omega \to \mathbb{R}$ *satisfies:*

   *(i) Uniform boundedness: $|\mathcal{G}_{\boldsymbol{\theta}}(z, \boldsymbol{y})| \leq M$ for all $z \in H$, $\boldsymbol{y} \in \Omega$;*
   *(ii) Lipschitz continuity in state: $|\mathcal{G}_{\boldsymbol{\theta}}(z_1, \boldsymbol{y}) - \mathcal{G}_{\boldsymbol{\theta}}(z_2, \boldsymbol{y})| \leq L_{\mathcal{G}} \|z_1 - z_2\|_H$;*
   *(iii) Continuity in position: $\boldsymbol{y} \mapsto \mathcal{G}_{\boldsymbol{\theta}}(z, \boldsymbol{y})$ is continuous for each fixed $z \in H$;*
   *(iv) Parameter regularity: $\nabla_{\boldsymbol{\theta}} \mathcal{G}_{\boldsymbol{\theta}}$ exists and satisfies (i)–(iii) with constants $M_{\boldsymbol{\theta}}$ and $L_{\mathcal{G}, \boldsymbol{\theta}}$.*

**(H3) Kernel Regularity:** $b \in L^2(\Omega \times \Omega)$ *with $\|b\|_{L^2} \leq L_b$, and $\boldsymbol{y} \mapsto b(\cdot, \boldsymbol{y})$ is continuous as a map into $H$.*

*If $L_{\mathcal{G}} L_b C_T < 1$, then as $M \to \infty$, the discrete policy gradient converges to the mean-field gradient:*

$$\nabla_{\boldsymbol{\theta}} \mathcal{J}_M \to \mathcal{D}_{\boldsymbol{\theta}} \mathcal{J}_{\infty}. \tag{11}$$

*Proof.* **Step 1: Uniform Bounds and Strong State Convergence.**

The discrete state $z_M$ satisfies the mild formulation:

$$z_M(t) = S(t) z_0 + \int_0^t S(t - s) \mathcal{B}_M(z_M)(s) \, ds. \tag{12}$$

By hypothesis (H2)(i) and (H3), the forcing term is uniformly bounded:

$$\|\mathcal{B}_M(z)\|_H \leq L_b M. \tag{13}$$

Under the smallness condition $L_{\mathcal{G}} L_b C_T < 1$, the Picard iteration map $\Phi : C([0, T]; H) \to C([0, T]; H)$ defined by the right-hand side is a strict contraction, guaranteeing a unique solution $z_M$ for each $M$.

Standard energy estimates yield: multiplying the PDE by $z_M$ and integrating, using the dissipativity of $\mathcal{A}$, we obtain

$$\frac{1}{2} \frac{d}{dt} \|z_M\|_H^2 + \alpha \|z_M\|_V^2 \leq \|\mathcal{B}_M(z_M)\|_H \|z_M\|_H \leq L_b M \|z_M\|_H. \tag{14}$$

Applying Grönwall's inequality yields uniform bounds:

$$\sup_M \left( \|z_M\|_{L^{\infty}(0, T; H)}^2 + \|z_M\|_{L^2(0, T; V)}^2 \right) \leq C(z_0, T, L_b, M, \alpha). \tag{15}$$

For the time derivative, from $\partial_t z_M = -\mathcal{A}z_M + \mathcal{B}_M(z_M)$, we have:

$$\|\partial_t z_M\|_{L^2(0,T;V^*)} \le \|\mathcal{A}z_M\|_{L^2(0,T;V^*)} + \|\mathcal{B}_M(z_M)\|_{L^2(0,T;H)} \le C, \tag{16}$$

where we used the continuous embedding $H \hookrightarrow V^*$ and the bound $\|\mathcal{A}z_M\|_{V^*} \le C\|z_M\|_V$.

By the **Aubin–Lions Lemma** (with $V \hookrightarrow\hookrightarrow H \hookrightarrow V^*$), the sequence $\{z_M\}$ is relatively compact in $L^2(0,T;H)$. Hence there exists a subsequence (still denoted $z_M$) such that:

$$z_M \to z_\infty \quad \text{strongly in } L^2(0,T;H). \tag{17}$$

**Step 2: Forcing Convergence and Identification of the Limit.**

We show that $z_\infty$ solves the mean-field equation with forcing $\mathcal{B}_\infty$. Decompose the forcing error:

$$\|\mathcal{B}_M(z_M) - \mathcal{B}_\infty(z_\infty)\|_H \le \underbrace{\|\mathcal{B}_M(z_M) - \mathcal{B}_M(z_\infty)\|_H}_{\text{(a) State error}} + \underbrace{\|\mathcal{B}_M(z_\infty) - \mathcal{B}_\infty(z_\infty)\|_H}_{\text{(b) Measure error}}. \tag{18}$$

*Term (a):* Using the Lipschitz property (H2)(ii) and the kernel bound (H3):

$$\begin{aligned}
\|\mathcal{B}_M(z_M) - \mathcal{B}_M(z_\infty)\|_H &= \left\| \int_\Omega b(\cdot, \boldsymbol{y}) \left[ \mathcal{G}_{\boldsymbol{\theta}}(z_M, \boldsymbol{y}) - \mathcal{G}_{\boldsymbol{\theta}}(z_\infty, \boldsymbol{y}) \right] d\mu_M(\boldsymbol{y}) \right\|_H \\
&\le L_b L_{\mathcal{G}} \|z_M - z_\infty\|_H \to 0,
\end{aligned} \tag{19}$$

since $z_M \to z_\infty$ strongly in $L^2(0,T;H)$.

*Term (b):* For fixed $t$ and $z_\infty(\cdot, t) \in H$, define the test function $\varphi : \Omega \to H$ by $\varphi(\boldsymbol{y}) = b(\cdot, \boldsymbol{y})\mathcal{G}_{\boldsymbol{\theta}}(z_\infty, \boldsymbol{y})$. By hypotheses (H2)(i), (H2)(iii), and (H3), the map $\boldsymbol{y} \mapsto \varphi(\boldsymbol{y})$ is continuous and bounded in $H$. Therefore, by the **Portmanteau Theorem** for weak-* convergence of measures:

$$\int_\Omega \varphi(\boldsymbol{y}) \, d\mu_M(\boldsymbol{y}) \to \int_\Omega \varphi(\boldsymbol{y}) \, \mu(\boldsymbol{y}) \, d\boldsymbol{y} = \mathcal{B}_\infty(z_\infty). \tag{20}$$

Combining terms (a) and (b), we conclude $\mathcal{B}_M(z_M) \to \mathcal{B}_\infty(z_\infty)$ in $L^2(0,T;H)$, and by uniqueness of limits in the mild formulation, $z_\infty$ is the unique solution to the mean-field state equation.

**Step 3: Adjoint Convergence.**

The discrete adjoint $p_M$ satisfies the backward parabolic equation:

$$-\partial_t p_M + \mathcal{A}^* p_M = 2(z_M - z_{\text{ref}}), \quad p_M(T) = 0. \tag{21}$$

Since $z_M \to z_\infty$ strongly in $L^2(0,T;H)$, the source term $2(z_M - z_{\text{ref}})$ converges strongly in $L^2(0,T;H)$. By standard parabolic regularity for the backward equation:

$$\sup_M \left( \|p_M\|_{L^\infty(0,T;H)} + \|p_M\|_{L^2(0,T;V)} + \|\partial_t p_M\|_{L^2(0,T;V^*)} \right) \le C. \tag{22}$$

A second application of the **Aubin–Lions Lemma** yields:

$$p_M \to p_\infty \quad \text{strongly in } L^2(0,T;H), \tag{23}$$

where $p_\infty$ solves the mean-field adjoint equation with source $2(z_\infty - z_{\text{ref}})$.

**Step 4: Gradient Consistency.**

The discrete gradient is:

$$\nabla_{\boldsymbol{\theta}} \mathcal{J}_M = \int_0^T \langle p_M, \nabla_{\boldsymbol{\theta}} \mathcal{B}_M(z_M) \rangle_H \, dt, \tag{24}$$

where $\nabla_{\boldsymbol{\theta}}\mathcal{B}_M(z) = \int_{\Omega} b(\cdot, \boldsymbol{y})\nabla_{\boldsymbol{\theta}}\mathcal{G}_{\boldsymbol{\theta}}(z, \boldsymbol{y})\, d\mu_M(\boldsymbol{y})$.

We decompose the error:

$$
\begin{aligned}
|\nabla_{\boldsymbol{\theta}}\mathcal{J}_M - \mathcal{D}_{\boldsymbol{\theta}}\mathcal{J}_{\infty}| \leq &\int_0^T |\langle p_M - p_{\infty}, \nabla_{\boldsymbol{\theta}}\mathcal{B}_M(z_M)\rangle_H|\, dt \\
&+ \int_0^T |\langle p_{\infty}, \nabla_{\boldsymbol{\theta}}\mathcal{B}_M(z_M) - \nabla_{\boldsymbol{\theta}}\mathcal{B}_{\infty}(z_{\infty})\rangle_H|\, dt.
\end{aligned}
\tag{25}
$$

*First term:* By hypothesis (H2)(iv), $\|\nabla_{\boldsymbol{\theta}}\mathcal{B}_M(z_M)\|_H \leq L_b M_{\boldsymbol{\theta}}$ uniformly. Thus:

$$
\int_0^T |\langle p_M - p_{\infty}, \nabla_{\boldsymbol{\theta}}\mathcal{B}_M(z_M)\rangle_H|\, dt \leq L_b M_{\boldsymbol{\theta}}\|p_M - p_{\infty}\|_{L^2(0,T;H)} \cdot \sqrt{T} \to 0.
\tag{26}
$$

*Second term:* Applying the same decomposition as in Step 2 to $\nabla_{\boldsymbol{\theta}}\mathcal{G}_{\boldsymbol{\theta}}$ (using (H2)(iv)), we obtain $\nabla_{\boldsymbol{\theta}}\mathcal{B}_M(z_M) \to \nabla_{\boldsymbol{\theta}}\mathcal{B}_{\infty}(z_{\infty})$ in $L^2(0, T; H)$. Since $p_{\infty} \in L^2(0, T; H)$:

$$
\int_0^T |\langle p_{\infty}, \nabla_{\boldsymbol{\theta}}\mathcal{B}_M(z_M) - \nabla_{\boldsymbol{\theta}}\mathcal{B}_{\infty}(z_{\infty})\rangle_H|\, dt \to 0.
\tag{27}
$$

Combining both terms, we conclude:

$$
\nabla_{\boldsymbol{\theta}}\mathcal{J}_M \to \mathcal{D}_{\boldsymbol{\theta}}\mathcal{J}_{\infty} \quad \text{as } M \to \infty.
\tag{28}
$$

$\square$

## B.2. Conjecture: Emergent Self-Normalization

While Theorem B.1 relies on normalized forcing ($\frac{1}{M}\sum u_i$), our implementation (Eq. 1) uses unnormalized superposition ($\sum u_i$). In our experiments in fact we operate under the assumption that no communication is allowed among the agents (not even the number of agents), to mimic scenarios where communication is impossible or limited. We conjecture that the optimization process induces an implicit normalization, enabling zero-shot transfer.

*Conjecture* 2 (Self-Normalization via Effort Regularization). Let $\boldsymbol{\theta}_M^*$ be the optimal policy minimizing $\mathcal{J}_M$ with effort penalty $\lambda_u > 0$ and unnormalized forcing. As $M \to \infty$:

1. **Intensity Scaling:** The learned control intensities scale as $u_i^*(t) := \mathcal{G}_{\boldsymbol{\theta}_M^*}(z_M, \boldsymbol{\xi}_i) = O(1/M)$.

2. **Forcing Consistency:** The total forcing $\mathcal{B}_M$ remains bounded, i.e., $\sup_M \|\mathcal{B}_M\|_{L^2(0,T;H)} < \infty$.

3. **Effort Decay:** The total control effort vanishes as $\mathcal{L}_{\text{force}} = \sum_{i=1}^M |u_i^*(t)|^2 = O(1/M) \to 0$.

This scaling emerges from the equilibrium between the tracking cost (requiring $O(1)$ total force) and the effort penalty (pushing $u_i \to 0$). If intensities remained $O(1)$, the total force would grow as $O(M)$, causing massive overshoot and high cost. Gradient descent drives the shared policy $\boldsymbol{\theta}$ to the unique scale $u_i \sim 1/M$ where the field $z$ is controlled effectively while minimizing $\lambda_u\|\boldsymbol{u}^M\|^2$.

This implies a cooperative efficiency gain: larger swarms achieve the same control objective with vanishingly small individual effort. Because the DeepONet policy shares parameters across all agents, this scaling is discovered *stigmergically* via the global state field $z$, without explicit communication of $M$.

*Remark* B.2 (Zero-Shot Transfer). This conjecture provides the mechanism for zero-shot scalability. The policy learns a response function that intrinsically balances the local contribution against the global field magnitude. When deployed with a different swarm size $Q \neq M$, the feedback loop through the physics $z$ automatically adjusts the aggregate forcing to the correct $O(1)$ level.

*Table 2.* **Multi-Agent Control Performance Across Tasks.** Benchmark results comparing CINOC and DPC against uncontrolled evolution and standard single-agent/multi-agent RL baselines. Values represent Tracking MSE, $L^2$ Energy/Enstrophy, and Moment Matching Loss (MML) depending on the task category and Lower is better. Tracking density matching problems involve kinetic actuators, while stabilization tasks use fixed actuators.

| Environment | Unctrl. | PPO | TD3 | MAPPO | MATD3 | DPC | CINOC |
|---|---|---|---|---|---|---|---|
| *Tracking (MSE)* | | | | | | | |
| FKPP 1D | $0.103 \pm 0.128$ | $0.053 \pm 0.091$ | $0.010 \pm 0.016$ | $0.004 \pm 0.010$ | $0.022 \pm 0.050$ | $1.4e\text{-}4 \pm 1.9e\text{-}4$ | $\mathbf{4.6e\text{-}5 \pm 7.5e\text{-}5}$ |
| Heat 1D | $0.390 \pm 1.144$ | $0.016 \pm 0.062$ | $0.006 \pm 0.029$ | $0.001 \pm 0.002$ | $0.016 \pm 0.032$ | $\mathbf{7.9e\text{-}5 \pm 1.1e\text{-}4}$ | $2.9e\text{-}4 \pm 4.3e\text{-}4$ |
| Heat 2D | $0.176 \pm 0.378$ | $0.015 \pm 0.026$ | $0.009 \pm 0.016$ | $0.006 \pm 0.016$ | $0.022 \pm 0.016$ | $2.1e\text{-}4 \pm 2.2e\text{-}4$ | $\mathbf{1.5e\text{-}4 \pm 1.8e\text{-}4}$ |
| Heat 2D (Obs) | $0.176 \pm 0.378$ | $0.016 \pm 0.023$ | $0.012 \pm 0.014$ | $0.004 \pm 0.015$ | $0.015 \pm 0.012$ | $2.7e\text{-}4 \pm 2.6e\text{-}4$ | $\mathbf{1.2e\text{-}4 \pm 1.7e\text{-}4}$ |
| *Stabilization (Energy/Enstrophy)* | | | | | | | |
| KS 1D | $1.45 \pm 0.69$ | $0.13 \pm 0.13$ | $0.13 \pm 0.13$ | $\mathbf{0.13 \pm 0.13}$ | $0.18 \pm 0.18$ | $0.16 \pm 0.15$ | $0.13 \pm 0.12$ |
| KS 2D | $13.91 \pm 4.61$ | $12.96 \pm 4.24$ | $1.66 \pm 0.58$ | $0.50 \pm 0.24$ | $0.91 \pm 0.27$ | $9.74 \pm 2.32$ | $\mathbf{0.37 \pm 0.19}$ |
| Turbulence 2D | $50.49 \pm 13.29$ | $34.52 \pm 12.35$ | $5.57 \pm 3.15$ | $4.48 \pm 2.84$ | $3.71 \pm 2.46$ | $3.28 \pm 2.19$ | $\mathbf{3.23 \pm 2.22}$ |
| *Density Matching (MML)* | | | | | | | |
| Density Control | $0.027 \pm 0.026$ | $0.005 \pm 0.005$ | $0.021 \pm 0.017$ | $0.008 \pm 0.008$ | $0.021 \pm 0.017$ | $\mathbf{0.002 \pm 0.010}$ | $0.002 \pm 0.002$ |

# C. Additional Numerical Experiments

This appendix provides extended qualitative results for the control problems referenced in the quantitative analysis (Table 2). We include visualizations for the 1D Fisher-KPP tracking, 1D and 2D Heat equation tracking in unconstrained domains, and scale-variant stabilization of the 1D Kuramoto-Sivashinsky equation. Further experimental details are reported in Appendix E.

## C.1. Fisher-KPP 1D: Tracking of Nonlinear Traveling Waves

We evaluate the learned policy on the Fisher-KPP equation, a canonical model characterized by the interplay between linear diffusion and non-linear reaction kinetics. This environment serves as a rigorous benchmark for assessing the coordination of 20 kinetic actuators. As illustrated in Figure 7, the emergent control strategy exhibits a distinct bi-phasic behavior. During the initial transient phase, the agents exert high-magnitude control input and maintain elevated velocities (see *the Avg Velocity* panel) to intercept the propagating wavefront. Upon achieving the target distribution at $t \approx 50$, the system undergoes a regime shift to a "maintenance mode." In this state, the agents transition to a quasi-stationary configuration, applying minimal restorative forcing to counteract the natural diffusive dissipation and preserve the desired spatial profile.

## C.2. Heat Equation 1D: Kinetic Source Tracking in Linear Systems

The 1D Heat equation tracking task isolates the control challenge of mitigating rapid dissipation within a linear parabolic system. Similar to the Fisher-KPP results, Figure 8 reveals an efficient allocation of control effort. The 8 kinetic actuators leverage their mobility to dynamically relocate to high-residual regions, indicated by the transient spike in *Avg Velocity*, where they inject thermal energy to match the target manifold. Subsequently, the agents adopt a stationary formation, providing the necessary forcing to balance the smoothing effects of the diffusion operator.

## C.3. Heat 2D: Emergent Coordination in Unconstrained Domains

To decouple the effects of geometric constraints from the emergent multi-agent coordination, we examine a 2D diffusion tracking task in an unconstrained domain. While the physical parameters and actuator count ($M = 16$) are identical to the obstacle-avoidance scenario in Section 6.2, the absence of "keep-out" zones permits a direct coverage strategy.

As visualized in Figure 9, the 16 kinetic agents adopt a dynamic tessellation of the domain, positioning themselves at the centroids of the local error distribution to maximize control authority. This stands in contrast to the "perimeter defense" behavior observed in the presence of obstacles, where agents were forced to act along boundary manifolds. This comparison confirms that the agent configurations are not a fixed artifact of the policy architecture, but rather an adaptive response to the environmental topology and the spatial requirements of the diffusion operator.

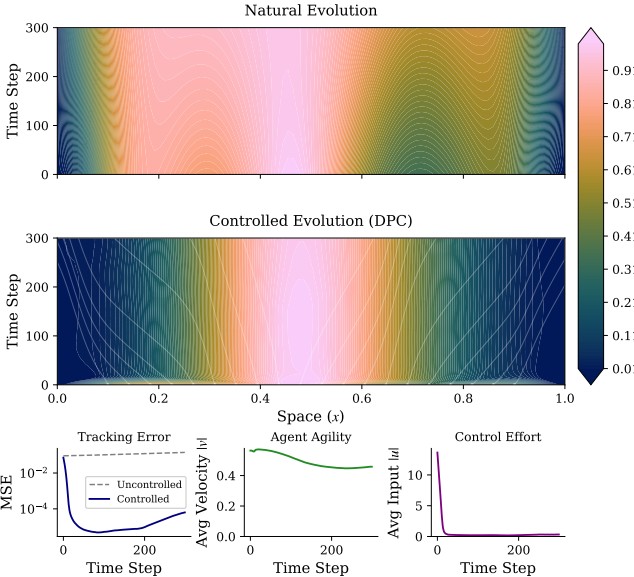

*Figure 7.* **Fisher-KPP 1D Tracking.** (Top) Uncontrolled evolution of the reaction-diffusion wavefront. (Bottom) Controlled trajectory wherein mobile actuators modulate the density field. The performance metrics demonstrate an efficient convergence: an initial high-velocity approach rapidly minimizes the tracking error, followed by a transition to a low-energy station-keeping configuration.

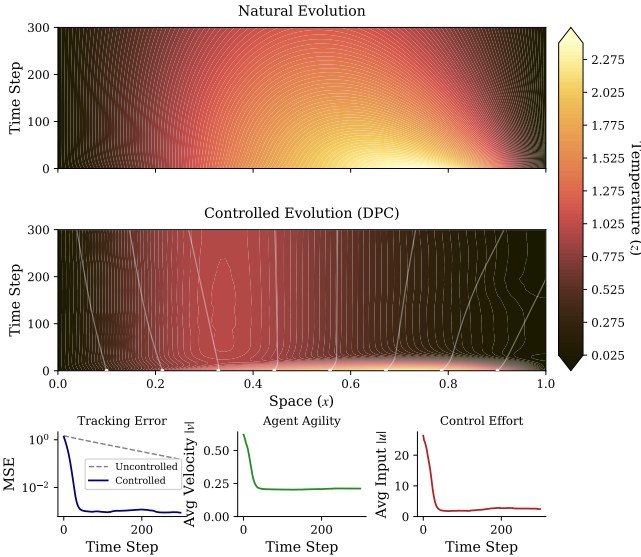

*Figure 8.* **Heat 1D Tracking.** (Top) Natural evolution demonstrating characteristic thermal dissipation. (Bottom) Controlled evolution maintaining a multi-modal target distribution. The policy exhibits rapid convergence (MSE attenuation) followed by a decay in control effort as the system reaches a steady-state maintenance phase.

### C.4. Kuramoto-Sivashinsky 1D: Multi-Scale Stabilization of Chaotic Dynamics

Finally, we investigate the stabilization of the 1D Kuramoto-Sivashinsky (KS) equation across varying spatial scales. To assess the scalability of the training pipeline, we evaluated three separate policies on domains of increasing length ($L \in \{64, 200, 500\}$) with a proportional increase in fixed actuators ($M \in \{30, 80, 200\}$).

Increased domain length in the KS equation corresponds to a higher-dimensional chaotic attractor and a denser spectrum of unstable modes (Figure 10, Left). Despite the increased complexity of the spatiotemporal chaos, the operator-learning framework consistently identifies robust control laws. Energy evolution metrics (Figure 10, Right) show that the policies successfully suppress chaotic fluctuations, driving system energy below $10^{-4}$ within $t = 5$s. These results suggest that the

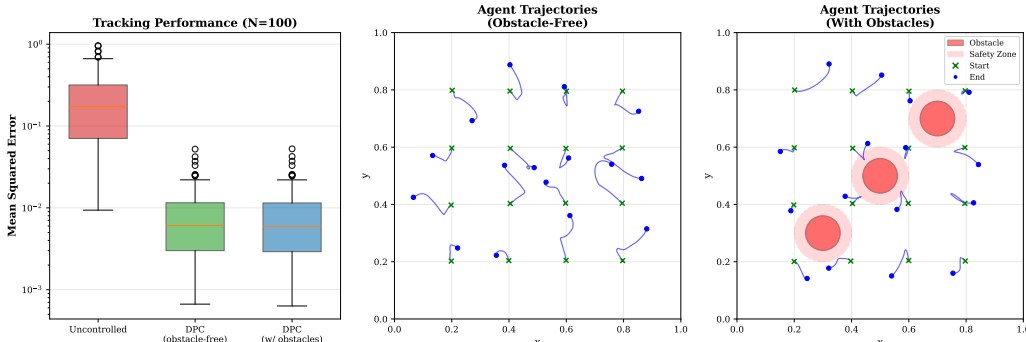

*Figure 9.* **Unconstrained Heat 2D Tracking.** Decentralized DPC performance on 2D heat equation control. (Left) Tracking MSE distribution shows significant error reduction for controlled scenarios regardless of obstacle presence (N=100). (Middle) Obstacle-free agents exhibit more aggressive motion toward high-error regions. (Right) Obstacle-aware agents execute prompt collision avoidance near safety boundaries (dashed circles) while maintaining tracking accuracy. Obstacles are shown as solid red circles.

learned local operators are able to effectively stabilize highly-chaotic systems.

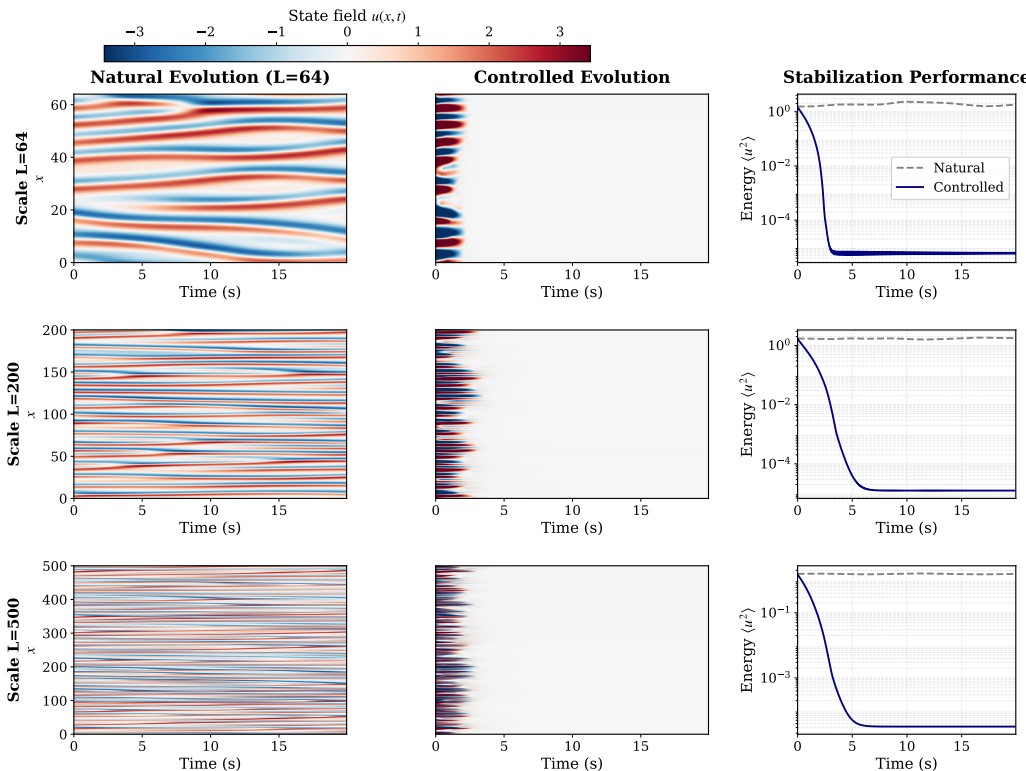

*Figure 10.* **Scale-Invariant Stabilization (KS 1D).** (Left) Natural evolution heatmaps illustrating the increase in chaotic complexity with domain size $L$. (Right) Energy evolution for natural (dashed) vs. controlled (solid) trajectories. The policy consistently achieves stabilization regardless of domain scale, demonstrating effective control of the underlying chaotic manifold.

## D. Ablation studies

### D.1. Empirical Verification of Emergent Scaling

A direct prediction of our Self-Normalization Conjecture (Section B.2) is the inverse scaling of individual control effort with population size. Specifically, for the total aggregate forcing to remain bounded as $M \to \infty$, individual agent intensities must scale approximately as $u_i \sim O(1/M)$. Consequently, the total squared effort metric should decay according to

$\sum_{i=1}^{M} u_i^2 \sim M \cdot (1/M)^2 = O(1/M)$.

We verify this emergent scaling empirically on the FKPP task. Figure 11 plots the mean total squared effort against the number of agents $M$ on a log-log scale. Crucially, this metric is computed only over the last $70\%$ of the control horizon. This averaging window is necessary to capture the steady-state behavior, allowing initial transient dynamics to settle and the stigmergic coordination loop via the PDE field to fully establish itself.

The results highlight the critical role of the effort penalty $\lambda_u$ in driving this cooperative efficiency. For sufficiently large regularization (e.g., $\lambda_u \geq 0.05$ in the provided plot), we observe the predicted monotonic decrease in total effort as $M$ increases, providing evidence for the conjectured $O(1/M)$ scaling. Conversely, when the penalty is negligible (e.g., $\lambda_u = 0.001$), the emergent scaling breaks down: agents fail to regulate down their individual contributions, leading to massive over-actuation and a sharp increase in total effort at larger swarm sizes. This confirms that the effort term provides the necessary optimization pressure to discover scalable, cooperative solutions and the relevance of the penalty effort as a hyperparameter in our setup.

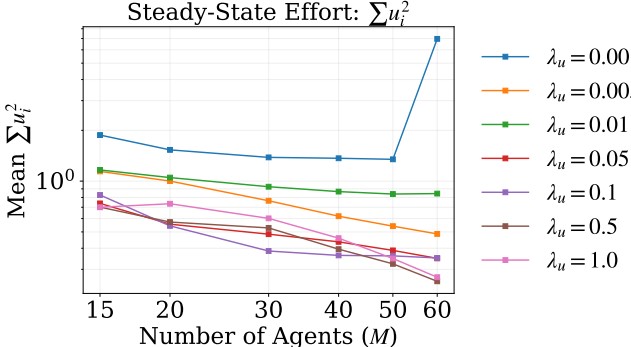

*Figure 11.* **Scaling of Steady-State Total Effort (Fisher-KPP).** The plot shows the mean $\sum u_i^2$ computed over the final $70\%$ of trajectories against the number of agents $M$ on a log-log scale. For sufficient effort penalties ($\lambda_u \geq 0.05$), the total effort decreases with a slope indicating $O(1/M)$ scaling, confirming cooperative self-normalization. Insufficient penalties ($\lambda_u \leq 0.005$) on control inputs lead to a breakdown of this scaling at larger $M$.

*Table 3.* **Cardinality Invariance.** The valid range of inference-time agent counts ($M_{test}$) where the policy maintains stability, defined as a loss $\leq 250\%$ of the training baseline ($M_{test} = M_{train}$). Values in parentheses indicate the scaling factor relative to the training size ($M_{train}$). Gray values denote the specific relative loss percentage at each boundary.

| PDE | Train ($M_{train}$) | Min Stable ($M_{test}$) | Max Stable ($M_{test}$) |
|---|---|---|---|
| FKPP 1D | 30 _100.00_ | 20 (0.7×) _114.97_ | 150 (5.0×) _128.73_ |
| Heat 1D | 30 _100.00_ | 15 (0.5×) _145.65_ | 120 (4.0×) _207.86_ |
| Heat 2D | 16 _100.00_ | 16 (1.0×) _100.00_ | 256 (16.0×) _53.50_ |
| KS 2D | 196 _100.00_ | 144 (0.7×) _45.02_ | 256 (1.3×) _230.10_ |
| KS 1D | 30 _100.00_ | 15 (0.5×) _99.75_ | 90 (3.0×) _114.42_ |
| Turbulence 2D | 64 _100.00_ | 36 (0.6×) _199.80_ | 1024 (16.0×) _75.89_ |
| Density 2D | 16 _100.00_ | 4 (0.2×) _141.53_ | 1024 (64.0×) _83.71_ |

## D.2. Is CINOC Robust to Solver Fidelity?

We evaluate the robustness of our learned policy to the fidelity of the differentiable solver. A critical requirement for practical deployment is that a policy trained on coarse, computationally efficient simulations must retain its efficacy when transferred to high-fidelity environments. To verify this resilience to discretization, we take a policy trained on the KS-2D stabilization task using a coarse solver with $64 \times 64$ grid points and evaluate it zero-shot against a high-fidelity simulation resolving $512 \times 512$ grid points.

Figure 12 (Left) illustrates the performance consistency across solver fidelities. Visually, the spatiotemporal evolution of the field in the top row ($64 \times 64$) and bottom row ($512 \times 512$) exhibits a high degree of coherence, indicating that the policy does not rely on grid-scale artifacts of the coarse solver.

Quantitatively, this robustness is confirmed by the system energy trajectories shown in Figure 12 (Right). The energy decay profile of the high-fidelity simulation (dashed red line) closely tracks that of the low-fidelity training environment (solid blue line). Both solvers are driven to the zero-state equilibrium with comparable rates (the low-fidelity simulation case achieves an energy of $0.83$, while the high-fidelity results in an energy of $0.15$, while the uncontrolled dynamics energy is $11.43$), demonstrating that the neural operator has successfully captured the underlying continuous control physics. This confirms the policy's robustness across discretization levels without retraining.

**Solver Fidelity Robustness in KS-2D Control**

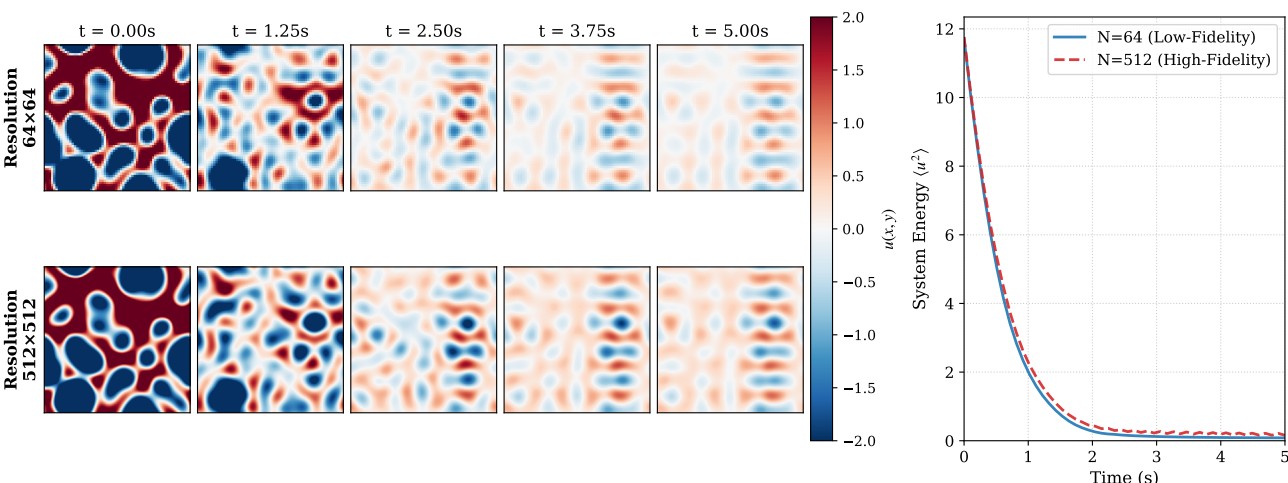

*Figure 12.* **Solver Fidelity Robustness in KS-2D Control.** Left: Comparison of the spatiotemporal evolution of the field $u(x, y)$. The policy, trained on a coarse discretization of $64 \times 64$ (Top), is deployed zero-shot on a high-fidelity solver with $512 \times 512$ grid points (Bottom). Right: Temporal evolution of the System Energy $\langle u^2 \rangle$. The high-fidelity trajectory ($N = 512$, dashed) closely matches the training performance ($N = 64$, solid), confirming the policy is robust to changes in solver discretization.

## D.3. Is the Learned Policy Robust to Signal Corruption?

To assess the robustness of the learned policies, we evaluate performance subject to stochastic signal corruption on the FKPP trajectory tracking task. We distinguish between **actuator uncertainty** ($\sigma_u$), modeled as additive Gaussian noise on the control output, and **observation uncertainty** ($\sigma_z$), representing sensor measurement noise.

**Robustness via Regularization.** Figure 15 illustrates the performance of policies trained in noisy environments and subsequently deployed in low- to mid-noise regimes (for both actuators and sensors). Results indicate that the *Baseline* policy, trained with zero standard deviation for sensor noise ($\sigma_u = 0$) and actuator noise ($\sigma_z = 0$), degrades severely as the swarm scales to $M = 100$. In contrast, policies trained with moderate state noise demonstrate superior performance across the entire population spectrum, even in noise-free scenarios as shown in Figure 13. The injected noise in each example is Gaussian with zero mean.

**Robustness and Stability Limits.** We identify noise injection during training as a technique to enhance the cardinality invariance properties of our policies. As illustrated in Figure 16 where a policy trained with noise-free actuators and sensors (Baseline) and a policy trained with $\sigma_z = 0.01$ and $\sigma_u = 0.02$ are compared in terms of their MSE, the latter policy preserve

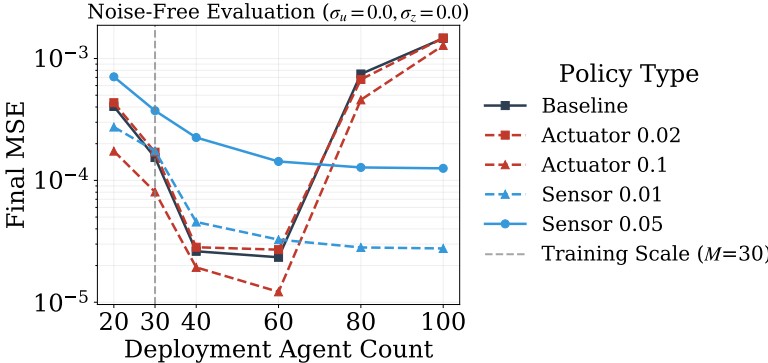

*Figure 13.* **Robustness Analysis.** Performance comparison between models trained with different noise levels. Inference is performed in a noise-less environment. Policies trained with marginal sensor noise exhibit better cardinality invariance.

the self-normalization property for larger swarm sizes compared to the baseline policy, which fails to maintain the necessary inverse effort scaling ($O(1/M)$) as the population grows. This suggests that, in the absence of noise regularization, the policy devolves into competitive behaviors, where agents over-actuate to correct for the perturbations introduced by their neighbors, leading to destructive interference and high-frequency oscillations. In contrast, the noise-conditioned policy maintains the correct scaling slope, effectively decoupling the agents and ensuring stronger cardinality invariance properties, confirming noise-injection as a mean to enhance the robustness of our method.

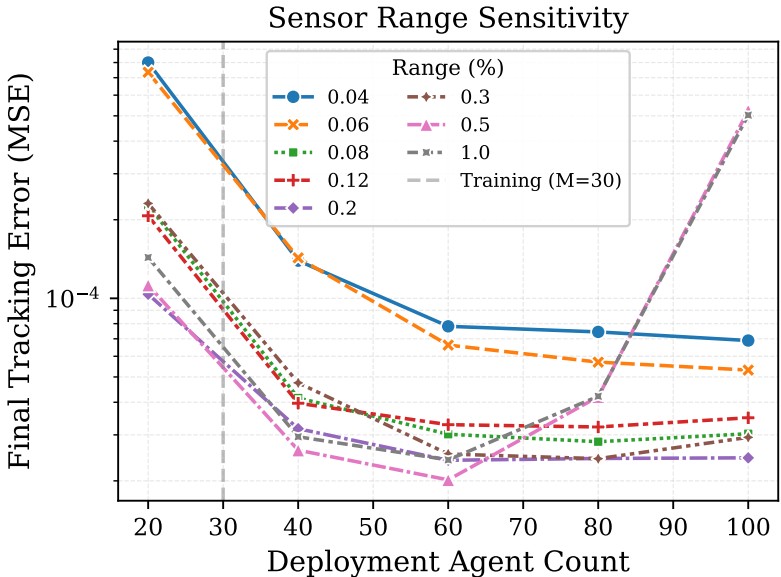

*Figure 14.* **Sensitivity to Field of View (FOV).** Comparison of control error (MSE) against swarm size for different observation radii on the FKPP task. While Global FOV (100%) performs well for small swarms, it creates oscillatory instability at large $M$ due to spurious coupling. Local FOV (10% - 20%) acts as a necessary regularizer, maintaining stability as population scales.

## D.4. What is the Role of Field of View Dimension?

We analyze the impact of the policy's observational range on performance by varying the Field of View (FOV) on the Fisher-KPP trajectory tracking task and comparing the performance of the different models across different swarm cardinalities.

Figure 14 reveals a counter-intuive phenomenon: while access to global information theoretically bounds performance from above, the global policy creates instability and oscillations as the swarm size $M$ increases. Specifically, with a fixed, small network architecture the global policy degrades significantly for $M > 50$, whereas the local policy remains stable. We attribute this to the fact that the Branch network compresses the input field into a low-dimensional latent embedding and when the input is the global state, the network must encode the entire domain's complexity into this fixed bottleneck and

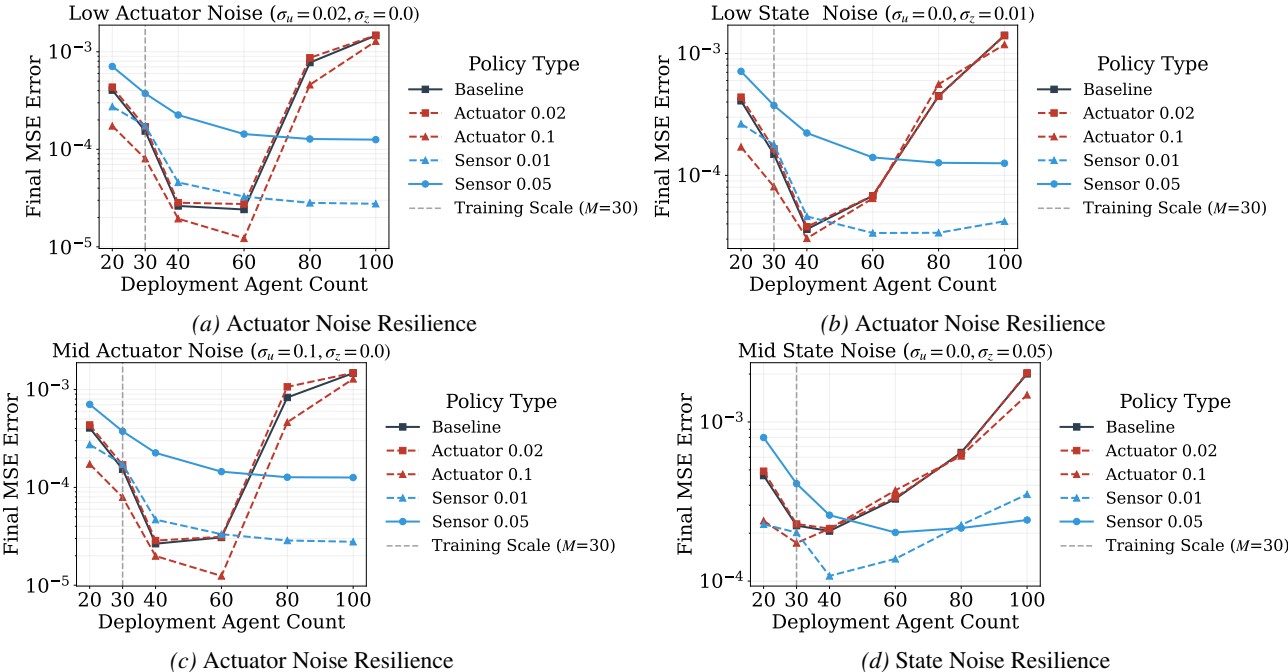

*(a)* Actuator Noise Resilience  *(b)* Actuator Noise Resilience

*(c)* Actuator Noise Resilience  *(d)* State Noise Resilience

*Figure 15.* **Robustness Analysis.** Comparative evaluation of Mean Squared Error (MSE) against deployment population size ($M$) in low- and mid-noise environments. The *Baseline* policy, trained in a noise-free environment, (solid black) degrades significantly as $M$ increases, indicating overfitting to the training physics. Policies trained with sensor-noise injection (dashed blue) exhibit robust scalability, reducing error by up to an order of magnitude at $M = 100$.

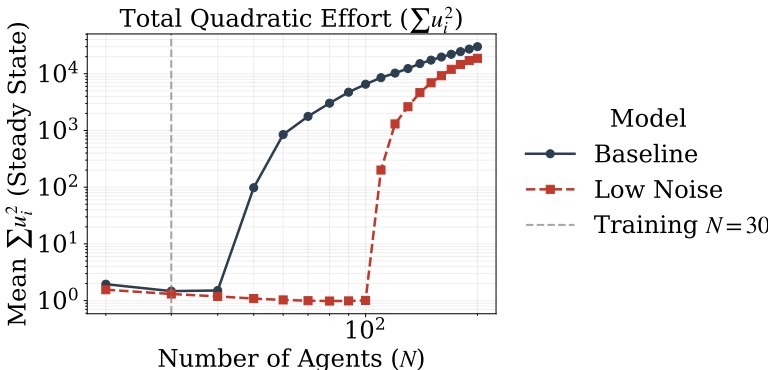

*Figure 16.* **The Role of Noise for more Robust Self-Normalization.** Log-log plot of total control effort vs. agent count. The Baseline policy (black), trained on a zero-noise environment, diverges from the ideal $O(1/M)$ scaling slope at high $M$, indicating a loss of coordination and the onset of competitive over-actuation. The policy trained in a low noise-scenario ($\sigma_z = 0.01$ and $\sigma_u = 0.02$) maintains the correct power law, preserving self normalization at higher cardinalities and ensuring better cardinality-invariant properties.

high-frequency local error features, critical for immediate stabilization, are dilute by the global signal, preventing the agent from reacting precisely to local disturbances.

Furthermore, global observation creates an all-to-all dependency graph. If Agent $i$ observes a perturbation caused by Agent $j$ at the opposite end of the domain, a small network may erroneously trigger a corrective action. Given the delay inherent in physical transport (diffusive time scales), this reaction arrives out of phase, inducing system-wide oscillations.

### D.5. Can the Policy Transfer to Unseen Physics?

To verify that the learned operator $\mathcal{G}_\theta$ encodes a generalized control policy, we evaluate its zero-shot transfer performance across variations in the underlying PDE coefficients. This evaluation spans two distinct control objectives: trajectory tracking

on reaction-diffusion dynamics and stabilization of 2D turbulent flow.

We first stress the policy on the Fisher-KPP *trajectory tracking* task by perturbing the dynamics relative to the training configuration ($\nu_{train} = 0.005, \rho_{train} = 3.0$). As shown in Figure 17, the policy maintains precise tracking even under extreme shifts, such as the complete removal of diffusion ($\nu \to 0$) or the doubling of reaction rates ($\rho \to 8.0$). This confirms that the agents do not rely solely on passive diffusion to distribute control influence, nor are they brittle to faster instability growth when tracking dynamic targets.

We further validate this on the *stabilization* of a 2D turbulent flow trained at a viscosity of $\nu_{train} = 5 \times 10^{-4}$. We deploy this fixed policy into environments with progressively lower viscosities (down to $2 \times 10^{-4}$), effectively increasing the Reynolds number. Figure 18 demonstrates that the policy successfully suppresses the energy cascade across this spectrum. Additionally, Figure 19 reveals a scalable synergy: increasing the actuator count consistently reduces steady-state error, effectively compensating for the harder dynamics of high-Reynolds-number flows.

We emphasize that these results demonstrate a degree of inherent robustness to parameter mismatch coming from the control feedback loop, rather than full generalization to arbitrary physics. We do not claim that the current method solves the parametric control problem, as the policy has no access to the physical coefficients. Extending this framework to parametric policies, which explicitly condition on physical parameters to enable broad generalization, remains a promising direction for future work.

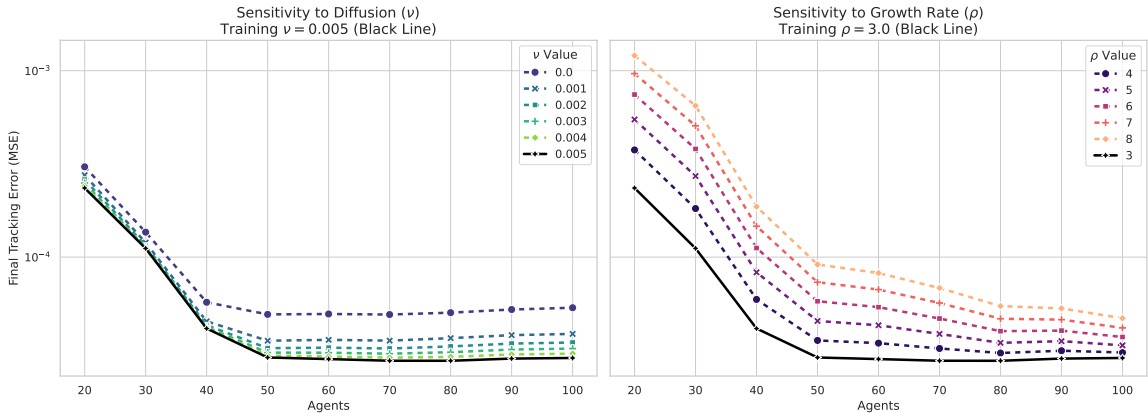

*Figure 17.* **Zero-Shot Robustness to Parametric Shifts (FKPP Tracking).** The policy, trained on fixed parameters ($\nu = 0.005, \rho = 3$), is deployed on mismatched dynamics to test generalization. (Left) The agent exhibits graceful degradation: tracking remains stable even when natural diffusion is eliminated ($\nu = 0.0$), maintaining coherent behavior despite the expected increase in tracking error. (Right) Agents track targets in systems with large reaction rates, resulting in bounded performance deviation.

### D.6. Model-Free MARL Validation of Cardinality Invariance

To further demonstrate that this cardinality invariance is an intrinsic property of the operator learning formulation rather than an artifact of the differentiable physics (model-based) training, we parameterized our policies using DeepONet and trained them via standard model-free multi-agent reinforcement learning (MARL) algorithms. Specifically, we evaluated Multi-Agent Proximal Policy Optimization (MAPPO) and Multi-Agent Twin Delayed DDPG (MATD3) on the 2D turbulent flow stabilization task.

**Network Architecture:** The DeepONet actor architecture for both MARL models utilizes a branch network consisting of a 2-layer Convolutional Neural Network (32 and 64 channels with $3 \times 3$ kernels and max pooling) followed by a dense layer with 128 units to process the local observation patches. The trunk network is composed of Fourier features (4 frequency bands) combined with a 2-layer MLP (64 and 128 units) to encode the spatial coordinates of the agents. These representations are merged via an element-wise product and passed through a final MLP to output the control actions. The centralized critic networks process the flattened global state (and joint actions for MATD3) using a 3-layer MLP with 512 and 256 hidden units.

**Training and Hyperparameters:** The MAPPO and MATD3 algorithms were trained for a total of 1,000,000 timesteps. For MAPPO, we utilized an actor and critic learning rate of $3 \times 10^{-4}$ (with linear decay), a clip ratio of 0.2, and 4 PPO

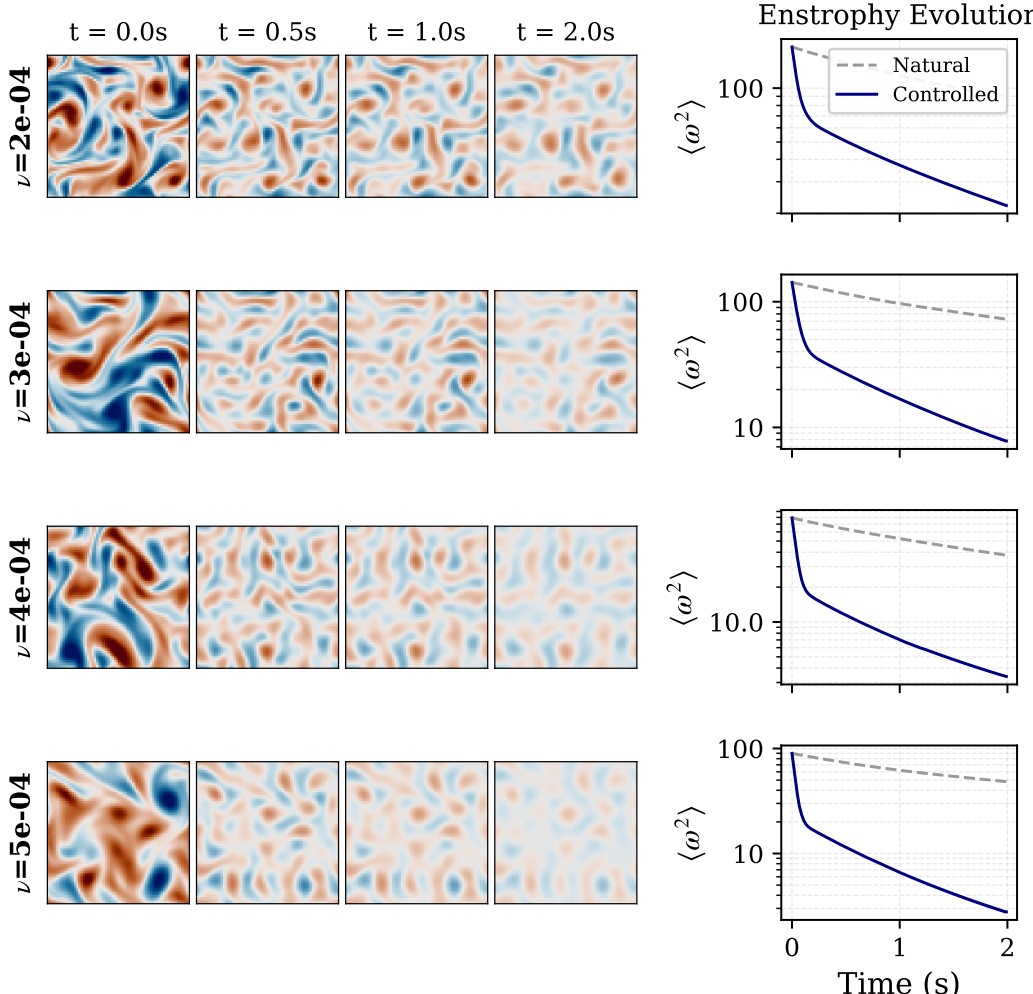

*Figure 18.* **Zero-Shot Transfer in 2D Turbulence (Stabilization).** The policy, trained at $\nu = 5 \times 10^{-4}$, is evaluated on more turbulent flows. (Left) Vorticity fields show suppression of chaotic structures even at $\nu = 2 \times 10^{-4}$. (Right) Enstrophy evolution confirms exponential decay across all tested regimes.

optimization epochs per rollout with a minibatch size of 200. For MATD3, the models were trained using a batch size of 256, an actor learning rate of $1 \times 10^{-4}$, and a critic learning rate of $5 \times 10^{-4}$. The MATD3 target policy smoothing noise was set to 0.2 (clipped at 0.5) with a policy update delay frequency of 2 steps.

As illustrated in Figure 20, the zero-shot scalability curve of the RL-trained policies closely mirrors that of CINOC. Both MAPPO and MATD3 exhibit robust cardinality invariance, maintaining low relative enstrophy when generalizing to swarms significantly larger than the training size ($N = 64$). This confirms that the emergent scalability is fundamentally tied to mapping the multi-agent control problem to an operator learning framework, independent of the underlying optimization scheme.

### D.7. Practical Guidelines and Rules of Thumb for Deployment

We appreciate that for practitioners, universal heuristics for deploying multi-agent PDE control would be highly desirable. However, the reality of deploying decentralized control across vastly different physical regimes, ranging from purely diffusive processes to chaotic, advective flows, is that optimal configurations are inherently dependent on the specific PDE and the downstream task. Because characteristic length scales, energy dissipation rates, and instability mechanisms vary so drastically between equations like Fisher-KPP and Navier-Stokes, a single, universal set of deployment rules is currently out of reach.

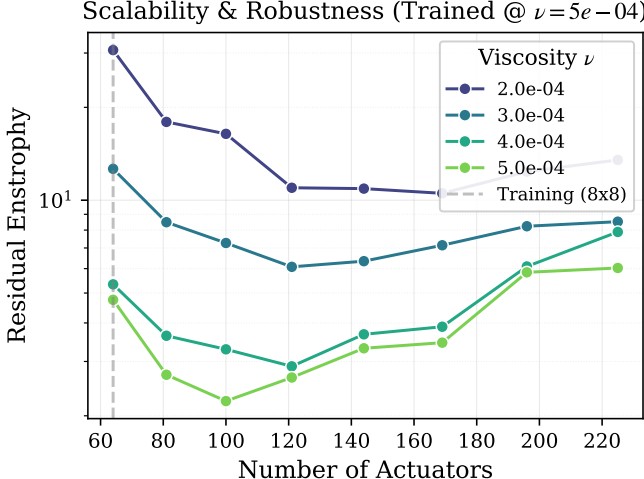

*Figure 19.* **Scalability under Physical Shift.** Residual enstrophy vs. agent count for different fluid viscosities. Performance improves monotonically with swarm size, showing that increased actuation density can compensate for the tougher dynamics of lower viscosity flows.

While deploying this framework on a novel PDE currently requires task-specific empirical tuning, our experiments reveal several consistent heuristics to guide the selection of agent density, sensing radii, and actuation widths based on the domain's unique physical properties.

### D.7.1. AGENT POPULATION ($M$) AND CONTROLLABILITY

The minimum number of agents, $M_{\min}$, is fundamentally bounded by the theoretical controllability and observability of the underlying PDE. For instance, in the 1D Kuramoto-Sivashinsky (KS) equation, the number of linearly unstable spatial modes dictates a strict mathematical minimum for the number of actuators required to stabilize the system; deploying fewer agents makes stabilization physically impossible regardless of the policy's quality.

**Rule of Thumb:** Identify the highest-frequency spatial mode or characteristic length scale of the target instability (e.g., the Taylor microscale in turbulence or the wavelength of the most unstable mode). Ensure the baseline agent population $M_{\text{train}}$ provides sufficient spatial sampling to address these modes, conceptually similar to a spatial Nyquist-Shannon criterion.

### D.7.2. ACTUATION RANGE ($\sigma$)

The width of the Gaussian control kernel, $\sigma$, must balance numerical stability with control precision. As noted in our scaling analysis, inter-agent distance decays linearly ($M^{-1}$) in 1D, but as $M^{-1/2}$ in 2D.

**Rule of Thumb:** To prevent grid-scale numerical artifacts and ensure smooth, differentiable gradients, $\sigma$ should be large enough to span at least 3–5 spatial grid points of the discretized solver. However, to prevent destructive interference and saturation at higher cardinalities, $\sigma$ must remain strictly smaller than the average inter-agent distance ($\propto L/M_{\text{train}}$ in 1D, or $\propto \sqrt{A/M_{\text{train}}}$ in 2D).

### D.7.3. FIELD OF VIEW (FOV) AND SENSING RADIUS

While a global FOV theoretically maximizes available information, our ablation studies (Appendix D.4) demonstrate that it can induce oscillatory instability in large swarms due to out-of-phase reactions to distant perturbations.

**Rule of Thumb:** The FOV should be restricted to the local characteristic length scale of the PDE's dynamics (e.g., the typical width of a flame front or vortex) plus a buffer accounting for the advective/diffusive transport expected during the control horizon. For wave-like or highly chaotic systems, a strictly local FOV (e.g., 10%-20% of the spatial domain) acts as a necessary regularizer to decouple agents, prevent spurious coupling, and preserve cardinality invariance.

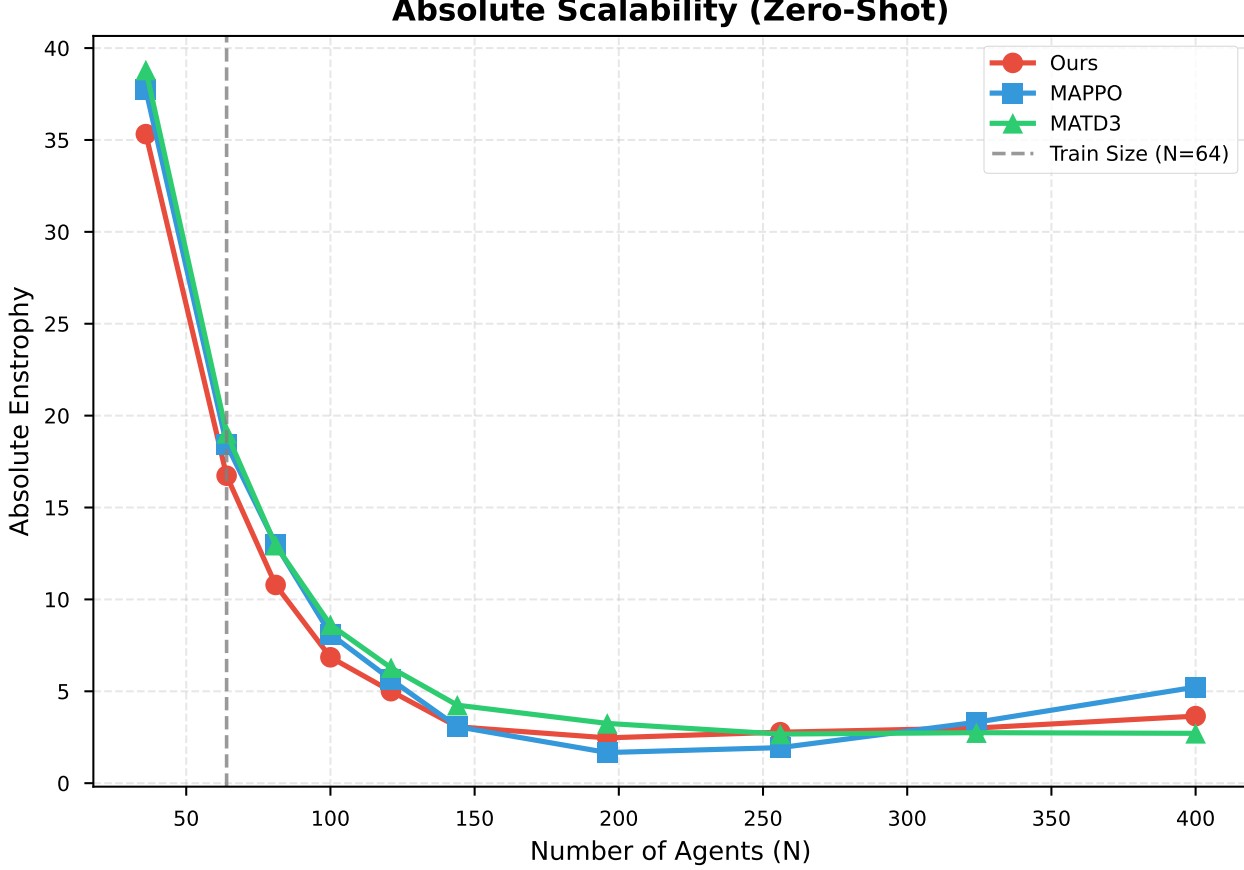

*Figure 20.* **Relative Scalability (Zero-Shot) of MARL-trained Operator Policies.** Comparison of zero-shot scalability on the 2D Turbulence stabilization task. The plot evaluates Relative Enstrophy (%) as a function of the inference-time agent count ($M$), with all models trained at $M = 64$. DeepONet policies trained with model-free algorithms (MAPPO and MATD3) exhibit almost identical cardinality invariance properties to CINOC, confirming that zero-shot scalability is a fundamental property of the operator parameterization rather than the training modality.

## E. Experimental Details

This section details the architectural configurations, training hyperparameters, and numerical discretization schemes used for our experiments. A quantitative performance comparison between the controlled evolution and the uncontrolled evolution is reported in Table 1.

### E.1. Trajectory Tracking

For trajectory tracking tasks, we instantiate the general control formulation (Equation 1) with kinetic (mobile) actuators and a tracking-specific objective. The optimization problem is defined as:

$$\min_{\boldsymbol{\theta}} \quad \mathcal{J}(\boldsymbol{\theta}) = \mathbb{E}_{z_0, \boldsymbol{\xi}_0} \left[ \int_0^T (\lambda_t \mathcal{L}_{\text{track}} + \lambda_u \mathcal{L}_{\text{effort}} + \lambda_s \mathcal{L}_{\text{safe}}) \, dt \right]$$

$$\text{s.t.} \quad \frac{\partial z}{\partial t} = \mathcal{F}(z) + \sum_{i=1}^M b(\boldsymbol{x}, \boldsymbol{\xi}_i(t)) u_i(t) \tag{29}$$

$$\dot{\boldsymbol{\xi}}_i(t) = v_i(t)$$

$$|u_i(t)| \leq u_{\max}, \quad |\boldsymbol{v}_i(t)| \leq v_{\max}, \quad \boldsymbol{\xi}_i(t) \in \Omega$$

$$[\boldsymbol{u}(t), \boldsymbol{v}(t)] = \mathcal{G}_{\boldsymbol{\theta}}(z(\boldsymbol{x}, t), \boldsymbol{\xi}(t))$$

Here, $\mathcal{F}$ denotes the PDE-specific differential operator (e.g., diffusion, reaction-diffusion), while the forcing term instantiates the actuation operator $\mathcal{B}$ from Equation 1 as:

$$\mathcal{B}(\boldsymbol{x}, \boldsymbol{u}(t)) = \sum_{i=1}^M b(\boldsymbol{x}, \boldsymbol{\xi}_i(t)) u_i(t) \tag{30}$$

where $u_i(t)$ represents the control intensity at actuator $i$, $\boldsymbol{\xi}_i(t)$ is the actuator position, and $v_i(t)$ governs the actuator's velocity. The spatial influence of each actuator is modeled via a Gaussian kernel:

$$b(\boldsymbol{x}, \boldsymbol{\xi}_i) = \frac{1}{(2\pi\sigma^2)^{d/2}} \exp\left(-\frac{|\boldsymbol{x} - \boldsymbol{\xi}_i|^2}{2\sigma^2}\right) \tag{31}$$

where $\sigma$ is fixed to represent the physical actuation range, mimicking real-world actuator characteristics. This superposition structure allows the total forcing field to be a weighted sum of localized actuator contributions, naturally accommodating varying numbers of agents without architectural changes.

The composite loss function comprises three components:

$$\mathcal{L}_{\text{track}} = \int_\Omega (z(\boldsymbol{x}, t) - z_{\text{target}}(\boldsymbol{x}))^2 \, d\boldsymbol{x},$$

$$\mathcal{L}_{\text{effort}} = \frac{1}{M} \sum_{i=1}^M \left( u_i(t)^2 + \lambda_v v_i(t)^2 + \lambda_a \dot{v}_i(t)^2 \right), \tag{32}$$

$$\mathcal{L}_{\text{safe}} = \frac{1}{M} \sum_{i=1}^M \left[ \left( \sum_{j=1}^M b(\boldsymbol{x}, \boldsymbol{\xi}_j(t)) u_j(t) \right)^2 + \sum_{j \neq i} \text{ReLU}(R_{\text{safe}} - |\boldsymbol{\xi}_i - \boldsymbol{\xi}_j|)^2 \right].$$

The tracking term $\mathcal{L}_{\text{track}}$ measures the $L^2$ deviation from the target profile. The effort term $\mathcal{L}_{\text{effort}}$ penalizes control intensity, kinetic energy ($\lambda_v v_i^2$), and acceleration ($\lambda_a \dot{v}_i^2$) to promote energy-efficient policies. The safety term $\mathcal{L}_{\text{safe}}$ enforces collision safety by penalizing inter-agent distances below the safe radius $R_{\text{safe}}$ and constrains the total forcing magnitude to prevent actuator saturation.

The constraints ensure that control intensities remain bounded by $u_{\max}$, actuator velocities respect kinematic limits $v_{\max}$, and agents remain within the spatial domain $\Omega$ throughout the control horizon.

**Implementation.** Our differentiable PDE solvers are implemented in JAX (Bradbury et al., 2018), with density transport problems utilizing PhiFlow (Holl & Thuerey, 2024) for advection-diffusion dynamics. For large-scale or high-fidelity simulations, the Tesseract framework (Häfner & Lavin, 2025) provides scalable alternatives.

**Heat 1D** We instantiate the problem formulation for the one-dimensional heat equation with mobile source terms. The system is defined over the spatial domain $\Omega = [0, 1]$ subject to homogeneous Dirichlet boundary conditions, $z(0, t) =$

$z(1, t) = 0$. The intrinsic dynamics are governed by linear diffusion:

$$\frac{\partial z}{\partial t} = \nu \frac{\partial^2 z}{\partial x^2} + \sum_{i=1}^{M} u_i(t) b(x, \xi_i(t)) \tag{33}$$

where $\nu = 0.2$ represents the diffusion coefficient. We simulate the environment with $M = 8$ controllable agents, each having an actuator width of $\sigma = 0.1$. The problem requires the policy to steer the field from an initial state $z_0(x)$ to a target state $z_{\text{target}}(x)$ within a fixed time horizon. To ensure robustness, both $z_0$ and $z_{\text{target}}$ are sampled from smooth Gaussian Random Fields (GRF) with Radial Basis Function (RBF) kernels. We utilize length scales of $l = 0.2$ for initial conditions and $l = 0.4$ for targets to generate diverse topological challenges. The physics are discretized on a spatial grid of $N_x = 100$ points using a Crank-Nicolson scheme with a fixed timestep $\Delta t = 0.001$.

**Fisher-KPP 1D.**  We apply the framework to the Fisher-Kolmogorov-Petrovsky-Piskunov (FKPP) equation, a fundamental model for nonlinear reaction-diffusion processes such as population dynamics, biological invasions, and epidemic spread. The system evolution on $\Omega = [0, 1]$ is governed by:

$$\frac{\partial z}{\partial t} = \nu \frac{\partial^2 z}{\partial x^2} + \rho z(1 - z) + \sum_{i=1}^{M} u_i(t) b(x, \xi_i(t)). \tag{34}$$

We select a diffusion coefficient $\nu = 0.005$ and a reaction rate $\rho = 3.0$, placing the system in a regime where both spatial transport and local logistic growth are significant. The control objective is achieved by $M = 20$ agents with an actuator influence width of $\sigma = 0.05$.

Due to the semi-linear nature of the equation, we employ an operator splitting numerical scheme. At each timestep $\Delta t = 0.001$, the non-linear reaction and control forcing are integrated via an explicit Euler step, followed by a fully implicit solution of the linear diffusion operator using a tridiagonal solver. To ensure physically meaningful population densities ($z \in [0, 1]$), the initialization and target data are not drawn from raw GRFs. Instead, we generate a latent GRF $g(x)$ and apply a transformation $z(x) \propto \exp(g(x)) \sin^2(\pi x)$, where the sine envelope strictly enforces Dirichlet boundary conditions while the exponential ensures positivity.

**Heat 2D**  We extend the experimental evaluation to the two-dimensional diffusion setting on the unit square $\Omega = [0, 1]^2$. The dynamics are governed by the 2D heat equation with isotropic diffusion:

$$\frac{\partial z}{\partial t} = \nu \left( \frac{\partial^2 z}{\partial x^2} + \frac{\partial^2 z}{\partial y^2} \right) + \sum_{i=1}^{M} u_i(t) b(\boldsymbol{x}, \boldsymbol{\xi}_i(t)) \tag{35}$$

subject to homogeneous Dirichlet boundary conditions $z(\boldsymbol{x}, t) = 0$ for all $\boldsymbol{x} \in \partial\Omega$. We increase the swarm size to $M = 16$ agents to cover the larger domain area, with a corresponding increase in actuator width to $\sigma = 0.15$ and a collision safety radius of $R_{\text{safe}} = 0.08$.

Due to the increased computational cost of fully implicit schemes in 2D, the physics solver utilizes the Peaceman-Rachford Alternating Direction Implicit (ADI) method. This splits the time evolution into two half-steps, implicit in $x$ then implicit in $y$, allowing for the inversion of tridiagonal matrices rather than the full sparse Hessian. The domain is discretized on a $32 \times 32$ spatial grid. Initial and target configurations are sampled from 2D Gaussian Random Fields. To satisfy the boundary constraints without introducing discontinuities, we apply a 2D bridge correction that subtracts a bilinear interpolation of the field's boundary values.

**Heat 2D with Obstacles**  To evaluate the collision avoidance capabilities of the policy in constrained environments, we introduce static forbidden regions $\mathcal{O} \subset \Omega$. The physical dynamics, solver configuration, and domain discretization follow the setup described in Section E.1.

The environment includes $K = 3$ circular obstacles defined by centers $\boldsymbol{c}_k$ and radii $r_k = 0.08$, positioned at $(0.15, 0.50)$, $(0.85, 0.50)$, and $(0.50, 0.15)$. The forbidden set is defined as $\mathcal{O} = \bigcup_{k=1}^{K} \{\boldsymbol{x} : \|\boldsymbol{x} - \boldsymbol{c}_k\|_2 \leq r_k\}$. Consequently, the optimization problem is modified to include a soft penalty term that discourages obstacle violation. The safety constraint for the $i$-th agent is updated to:

$$\min_{j \neq i} \|\boldsymbol{\xi}_i - \boldsymbol{\xi}_j\| \geq R_{\text{safe}} \quad \text{and} \quad \min_{k} \|\boldsymbol{\xi}_i - \boldsymbol{c}_k\| \geq R_{\text{safe}} + r_k \tag{36}$$

where $R_{\text{safe}}$ remains 0.08. Target field generation and boundary corrections are identical to the unconstrained case.

## E.2. Stabilization

The problem definition for our stabilization tasks is akin to Equation 29, with the difference that $z_{\text{target}}$ is chosen to be 0. We also employ fixed actuators to showcase our paradigm in different settings.

**Kuramoto-Sivashinsky 1D (KS)**    We investigate the control of spatiotemporal chaos using the Kuramoto-Sivashinsky equation, a canonical model for pattern formation in flame fronts and thin-film flows. The system evolves on a periodic domain $\Omega = [0, L]$ according to:

$$\frac{\partial z}{\partial t} + z\frac{\partial z}{\partial x} + \frac{\partial^2 z}{\partial x^2} + \frac{\partial^4 z}{\partial x^4} = \sum_{i=1}^{M} u_i(t)b(x, \xi_i). \tag{37}$$

We set the domain size to $L = 22.0$, a regime exhibiting sustained chaotic behavior. The control interface consists of $M = 8$ static actuators ($\dot{\xi}_i = 0$) equidistantly spaced across the domain, modulating their intensity $u_i(t)$ with a spatial influence width of $\sigma = 1.0$. The objective is to suppress the intrinsic turbulence and stabilize the system to the zero state $z(x) \equiv 0$.

Unlike the diffusion-dominated examples, we utilize a pseudo-spectral numerical scheme to accurately resolve high-order derivatives. The linear operator $\mathcal{L} = -\partial_x^2 - \partial_x^4$ is computed diagonally in Fourier space ($k^2 - k^4$), while the non-linear advection term $uu_x$ is computed in real space and transformed via FFT. Time integration is performed using a semi-implicit Crank-Nicolson method with $\Delta t = 0.05$ on a grid of $N_x = 128$ modes. To ensure robust stabilization capabilities, training initial conditions are not random noise but rather "spun-up" states: random perturbations evolved autonomously for $T_{\text{warm}} = 5000$ time units to project the system onto its chaotic attractor before control begins.

**Kuramoto-Sivashinsky 2D**    Extending the chaotic stabilization task described in Sec. E.2, we consider the two-dimensional variant on a periodic domain $\Omega = [0, L]^2$ with $L = 32.0$. The dynamics are governed by the equation:

$$\frac{\partial z}{\partial t} = -\nabla^2 z - \nabla^4 z - \frac{1}{2}|\nabla z|^2 + \sum_{i=1}^{M} u_i(t)b(\boldsymbol{x}, \boldsymbol{\xi}_i). \tag{38}$$

The system is actuated by a dense grid of $M = 100$ static agents arranged in a $10 \times 10$ lattice, with an actuator width of $\sigma = 1.2$.

The numerical challenges in 2D require a more sophisticated solver than the semi-implicit schemes used in 1D. We employ the Fourth-order Exponential Time Differencing Runge-Kutta (ETDRK4) method, which provides superior stability for stiff PDEs by treating the linear operator in Fourier space. Spatial derivatives are computed spectrally on a $64 \times 64$ grid with the $\frac{2}{3}$-rule applied for de-aliasing the non-linear term. To align the control frequency with the characteristic timescales of the turbulence while maintaining numerical precision, we employ temporal action repetition: the physics integration proceeds at a fine timestep $\Delta t = 0.005$, while the control policy is queried every $k = 20$ substeps (effective control $\Delta t_c = 0.1$). As in the 1D case, training is performed on fully developed chaotic states generated via a spin-up procedure.

**Turbulence 2D**    We address the control of fully developed decaying isotropic turbulence. The system is modeled by the incompressible Navier-Stokes equations in their vorticity-streamfunction formulation, which casts the dynamics as a non-linear advection-diffusion problem for the vorticity $\omega$:

$$\frac{\partial \omega}{\partial t} + (\boldsymbol{q} \cdot \nabla)\omega = \nu\nabla^2\omega + \sum_{i=1}^{M} u_i(t)b(\boldsymbol{x}, \boldsymbol{\xi}_i). \tag{39}$$

Unlike passive scalar advection, the velocity field $\boldsymbol{q}$ is coupled to the vorticity via the stream function Poisson equation $\nabla^2\psi = -\omega$, where $\boldsymbol{q} = (\partial_y\psi, -\partial_x\psi)$. This non-linear self-advection drives the turbulent energy cascade. We simulate the system with viscosity $\nu = 5 \times 10^{-4}$ on a periodic domain $\Omega = [0, 1]^2$, using a Pseudo-Spectral solver with RK4 integration ($\Delta t = 0.01$) and $\frac{3}{2}$-rule de-aliasing to resolve the non-linear interactions. Similarly to the KS problems outlined above, we train the policy to drive the system to 0 after evolving the system to a turbulent state.

## E.3. Density Matching

For density matching tasks, we shift from controlling source terms to manipulating the advection field itself. The objective is to transport a passive scalar distribution $\rho(\boldsymbol{x}, t)$ to match a static target configuration $\rho^*(\boldsymbol{x})$ while conserving total mass:

$$
\begin{aligned}
\min_{\boldsymbol{\theta}} \quad & \mathcal{J}(\boldsymbol{\theta}) = \mathbb{E}_{\rho_0, \boldsymbol{\xi}_0} \left[ \lambda_w \mathcal{W}(\rho(T), \rho^*) + \int_0^T (\lambda_m \mathcal{L}_{\text{mass}} + \lambda_u \mathcal{L}_{\text{effort}} + \lambda_s \mathcal{L}_{\text{safe}}) \, dt \right] \\
\text{s.t.} \quad & \frac{\partial \rho}{\partial t} + \nabla \cdot (\rho \boldsymbol{q}) = \nu \nabla^2 \rho \\
& \boldsymbol{q}(\boldsymbol{x}, t) = \boldsymbol{q}_{\text{nat}} + \sum_{i=1}^M \boldsymbol{v}_i(t) b(\boldsymbol{x}, \boldsymbol{\xi}_i(t)) \\
& \dot{\boldsymbol{\xi}}_i(t) = \alpha \boldsymbol{v}_i(t) \\
& |\boldsymbol{v}_i(t)| \le v_{\max}, \quad \boldsymbol{\xi}_i(t) \in \Omega
\end{aligned}
\tag{40}
$$

This formulation differs fundamentally from trajectory tracking (Eq. 29) in several key aspects. First, the velocity field $\boldsymbol{q}$ is constructed as a superposition of a background drift $\boldsymbol{q}_{\text{nat}}$ and local flow fields injected by mobile agents. Second, the primary objective $\mathcal{W}$ employs a moment-matching loss (computed via center-of-mass distance) rather than pointwise $L^2$ error.

In the density transport task, the initial density $\rho(\boldsymbol{x}, t)$ and the target density $\rho^*(\boldsymbol{x})$ often have disjoint supports (i.e., they do not overlap). In this regime, pointwise metrics such as Mean Squared Error (MSE) or Cross-Entropy fail to provide informative gradients, as the derivative of the loss with respect to position is zero in the empty space between distributions.

To provide a dense gradient signal that guides the agents to transport mass across the domain, we minimize the discrepancy between the first and second statistical moments of the distributions. Let $\tilde{\rho} = \rho / \int_\Omega \rho \, d\boldsymbol{x}$ be the normalized probability measure of the density field. We compute the spatial centroid (first moment) $\boldsymbol{\mu} \in \mathbb{R}^2$:

$$
\boldsymbol{\mu}(\rho) = \int_\Omega \boldsymbol{x} \, \tilde{\rho}(\boldsymbol{x}) \, d\boldsymbol{x}.
\tag{41}
$$

To ensure the transported density matches the shape of the target, we also compute the spatial spread (approximated by the standard deviation of the distribution relative to its centroid):

$$
\sigma(\rho) = \left( \int_\Omega \|\boldsymbol{x} - \boldsymbol{\mu}(\rho)\|^2 \, \tilde{\rho}(\boldsymbol{x}) \, d\boldsymbol{x} \right)^{1/2}.
\tag{42}
$$

The total Moment Matching loss is defined as the weighted sum of the centroid distance (Wasserstein-1 approximation) and the spread difference:

$$
\mathcal{W}(\rho, \rho^*) = \|\boldsymbol{\mu}(\rho) - \boldsymbol{\mu}(\rho^*)\|_2 + \lambda_{\text{var}} |\sigma(\rho) - \sigma(\rho^*)|,
\tag{43}
$$

where we set $\lambda_{\text{var}} = 0.5$ in our experiments. This formulation provides a convex gradient landscape that pulls the density toward the target location regardless of spatial overlap.

Last, the mass conservation penalty is defined as $\mathcal{L}_{\text{mass}} = \left( \int_\Omega \rho(\boldsymbol{x}, t) \, d\boldsymbol{x} - m_0 \right)^2$, ensuring the policy transports rather than creates or destroys density.

**Advection-Diffusion 2D (AD)**   We instantiate this formulation on a rectangular domain $\Omega = [0, 1] \times [0, 1.25]$ with zero-flux (Neumann) boundary conditions. The control interface comprises $M = 9$ mobile agents that inject velocity with Gaussian spatial influence (width $\sigma = 0.2$) and maximum intensity $v_{\max} = 0.8$. The diffusion coefficient $\nu = 0.01$ creates an advection-dominated regime with mild diffusive smoothing.

To maintain stability under high-velocity transport, we employ a Semi-Lagrangian scheme. The advective term is resolved via backward trajectory tracing ($\boldsymbol{x}_{\text{src}} = \boldsymbol{x} - \boldsymbol{q}\Delta t$) with bilinear interpolation of the density field. This approach permits a relatively large timestep $\Delta t = 1.0$ on a $64 \times 80$ grid without violating CFL stability conditions. The simulation horizon is $T = 150$ timesteps.

The loss function prioritizes terminal-state accuracy with weight $\lambda_w = 4.0$ on the moment-matching term, while strongly penalizing mass loss ($\lambda_m = 6.0$) to prevent the policy from trivially minimizing error by artificially removing density. The effort and safety penalties ($\lambda_u, \lambda_s$) follow the same structure as in trajectory tracking.

# F. Training and Architecture Specifications

## F.1. Training Hyperparameters

Table 4 summarizes the optimization settings used for training the neural operator policies across all problem domains. All experiments employ the Adam optimizer with gradient clipping to ensure stable training dynamics. The choice of batch size reflects a trade-off between sample efficiency and GPU memory constraints, with smaller batches (4-16) used for high-resolution 2D problems and larger batches (32) for 1D cases. The turbulent flow problem requires fewer epochs (50) due to its higher per-epoch computational cost, while the density transport task benefits from extended training (1000 epochs) to learn the complex advection dynamics.

*Table 4.* Training hyperparameters for all benchmark problems.

| Problem | Batch Size | Epochs | LR Schedule | Initial LR | Grad Clip |
|---------|------------|--------|-------------|------------|-----------|
| FKPP 1D | 32 | 500 | Exp. decay | 1e-3 | 1.0 |
| Heat 1D | 32 | 500 | Exp. decay | 1e-3 | 1.0 |
| Heat 2D | 16 | 500 | Exp. decay | 1e-3 | 1.0 |
| KS 1D | 32 | 500 | Exp. decay | 1e-3 | 1.0 |
| KS 2D | 4 | 500 | Warmup+Cosine | 5e-4 $\rightarrow$ 1e-3 $\rightarrow$ 1e-5 | 1.0 |
| Turb. 2D | 4 | 500 | Warmup+Cosine | 1e-4 $\rightarrow$ 5e-4 $\rightarrow$ 1e-6 | 1.0 |
| Density | 4 | 1000 | Exp. decay | 1e-3 | 1.0 |

**Learning Rate Schedules:**

- **Exponential decay**: $\eta_t = \eta_0 \cdot \gamma^{t/T_{\text{decay}}}$ where $\gamma = 0.5$.

- **Warmup + Cosine**: Linear warmup to peak LR followed by cosine annealing to minimum LR.

## F.2. Loss Function Weights

The total loss function balances multiple objectives: tracking accuracy, control effort minimization, constraint satisfaction, and problem-specific regularization. Table 5 presents the weight coefficients for each loss component. The tracking weight ($\lambda_{\text{track}}$) is consistently the largest term to prioritize control performance. The effort penalty ($\lambda_{\text{effort}}$) is kept small to avoid over-penalizing control authority. Boundary violation penalties ($\lambda_{\text{bound}}$) are large (10-100) to enforce hard constraints. The collision avoidance weight ($\lambda_{\text{coll}}$) varies from 1.0 to 20.0 depending on agent density and obstacle presence. Problem-specific terms include enstrophy regularization for turbulence control and mass conservation penalties for density transport.

*Table 5.* Loss function weight coefficients. Components: $\mathcal{L}_{\text{track}}$ (state tracking), $\mathcal{L}_{\text{effort}}$ (control regularization), $\mathcal{L}_{\text{bound}}$ (boundary violations), $\mathcal{L}_{\text{coll}}$ (collision avoidance), $\mathcal{L}_{\text{accel}}$ (velocity smoothness).

| Problem | $\lambda_{\text{track}}$ | $\lambda_{\text{effort}}$ | $\lambda_{\text{bound}}$ | $\lambda_{\text{coll}}$ | $\lambda_{\text{accel}}$ | **Other** |
|---------|------------|-------------|-------------|------------|------------|-------|
| FKPP 1D | 5.0 | 0.001 | 100.0 | 1.0 | 0.1 | — |
| Heat 1D | 5.0 | 0.001 | 100.0 | 1.0 | 0.1 | — |
| Heat 2D | 5.0 | 0.001 | 100.0 | 20.0 | 0.1 | — |
| KS 1D | 10.0 | 0.001 | — | — | — | — |
| KS 2D | 100.0 | 1e-4 | — | — | — | — |
| Turb. 2D | — | 1e-5 | — | — | — | Enstrophy: 1.0 |
| Density | 4.0 (hold) | 0.001 | 10.0 | 14.0 | 0.1 | $\lambda_{\text{mass}} = 6.0, \lambda_{\text{smooth}} = 0.1$ |

## F.3. Policy Network Architecture

All policies follow the DeepONet framework with problem-specific adaptations to the branch and trunk networks. Table 6 details the architectural choices for each benchmark. For 1D problems, we use purely MLP-based branch networks that process the local field observations. For 2D problems, we employ convolutional layers in the branch network to efficiently extract spatial features from the high-dimensional observation patches. The trunk network consistently uses Fourier features

(4 frequency bands) combined with an MLP to encode agent positions, enabling the policy to learn position-dependent control strategies. The output dimensionality varies by problem: kinetic actuators produce both forcing ($u$) and velocity ($v$) outputs, while static actuators generate only forcing terms.

Table 6. Neural operator policy architectures. Fourier features use 4 frequency bands: $[\sin(2\pi k\xi), \cos(2\pi k\xi)]$ for $k \in \{1, 2, 4, 8\}$.

| Problem | Branch Net | Trunk Net | Output |
|---------|-----------|-----------|--------|
| FKPP 1D | MLP (64, 64) | Fourier (4 freq) + MLP (32, 32) | $u, v$ |
| Heat 1D | MLP (64, 64) | Fourier (4 freq) + MLP (32, 32) | $u, v$ |
| Heat 2D | CNN (16, 32) + MLP | Fourier (4 freq) + MLP (32, 32) | $u, v$ |
| KS 1D | MLP (64, 64) | Fourier (4 freq) + MLP (32, 32) | $u$ only |
| KS 2D | CNN (16, 32) + MLP | Fourier (4 freq) + MLP (32, 32) | $u$ only |
| Turb. 2D | CNN (32, 64) + MLP | Fourier (4 freq) + MLP (32, 32) | $u$ only |
| Density | CNN (16, 32) + MLP | Fourier + Direction vectors | $v$ only |

## F.4. Local Observation Specifications

To enable scalable multi-agent control, each actuator observes only a local patch of the global state field, as detailed in Table 7. The observation window size is chosen to balance information content with computational efficiency: 1D problems use sliding windows covering 8% of the domain (20 points), while 2D problems employ square patches of size $12 \times 12$ or $20 \times 20$. All observations include the tracking error field and its spatial gradients, computed via finite differences. For periodic domains (KS and Turbulence), observations wrap around boundaries to maintain consistency. The turbulence problem uses vorticity ($\omega$) instead of the velocity field to provide a more compact representation of the flow dynamics.

Table 7. Local observation specifications for each agent. Each observation includes the tracking error and its spatial gradients computed via finite differences.

| Problem | Observation Type | Patch/Window Size | Channels |
|---------|-----------------|-------------------|----------|
| FKPP 1D | 1D sliding window | 8% of domain $\to$ 20 points | error + $\nabla$error |
| Heat 1D | 1D sliding window | 8% of domain $\to$ 20 points | error + $\nabla$error |
| Heat 2D | 2D local patch | $12 \times 12$ | error + $\nabla_x$ error + $\nabla_y$ error |
| KS 1D | 1D periodic window | 4 points $\to$ 20 | error + $\nabla$error |
| KS 2D | 2D periodic patch | $12 \times 12$ | error + $\nabla_x + \nabla_y$ |
| Turb. 2D | 2D periodic patch | $20 \times 20$ | $\omega + \nabla_x\omega + \nabla_y\omega$ |
| Density | Full Field | – | error + $\nabla$error |

## F.5. Control Saturation Limits

Control inputs are bounded to ensure physical feasibility and numerical stability, as specified in Table 8. The bounds are enforced via $\tanh$ activation functions in the policy output layer. Forcing magnitude limits ($u_{\max}$) range from 1.0 for the sensitive KS 1D system to 75.0 for the high-Reynolds turbulent flow, reflecting the different energy scales of each problem. Velocity limits ($v_{\max}$) for kinetic actuators are set to allow meaningful displacement within each control timestep while preventing excessively rapid motion that could destabilize the coupled system. Static actuators (KS and Turbulence) generate only forcing terms and maintain fixed positions throughout the episode.

## F.6. Data Generation and Initial Conditions

Training data is generated by sampling diverse initial and target states to ensure policy generalization, as detailed in Table 9. For linear and weakly nonlinear systems (Heat, FKPP), we use Gaussian Random Fields (GRF) with different length scales ($\ell$) for initial and target distributions to create meaningful tracking tasks. For chaotic systems (KS 1D, KS 2D), initial conditions are evolved from the chaotic attractor with extended warmup periods (200-5000 time units) to ensure the system resides on its strange attractor at training onset. The turbulent flow is initialized from a developed turbulent state and controlled toward the zero equilibrium. The density transport problem uses randomly placed Gaussian blobs to simulate diverse scalar distribution patterns.

*Table 8.* Control input saturation limits enforced via $\tanh(\cdot)$ activation. $u$: forcing term; $v$: actuator velocity.

| Problem | $u_{\max}$ | $v_{\max}$ | Control Type |
|---------|------------|------------|--------------|
| FKPP 1D | 40.0 | 2.0 | Forcing + Velocity |
| Heat 1D | 40.0 | 2.0 | Forcing + Velocity |
| Heat 2D | 40.0 | 5.0 | Forcing + 2D Velocity |
| KS 1D | 1.0 | — | Forcing only (static) |
| KS 2D | 10.0 | — | Forcing only (static) |
| Turb. 2D | 75.0 | — | Forcing only (static) |
| Density | — | 0.8 | Velocity only (push) |

*Table 9.* Data generation protocols for training: initial condition (IC) types, target distributions, and warmup procedures. GRF denotes Gaussian Random Field.

| Problem | IC Type | Target | Warmup |
|---------|---------|--------|--------|
| FKPP 1D | GRF ($\ell = 0.2$) | GRF ($\ell = 0.4$) | — |
| Heat 1D | GRF ($\ell = 0.2$) | GRF ($\ell = 0.4$) | — |
| Heat 2D | 2D GRF ($\ell = 0.25$) | 2D GRF ($\ell = 0.4$) | — |
| KS 1D | Chaotic attractor | Zero state | 5000 time units |
| KS 2D | Chaotic attractor | Zero state | 200 time units |
| Turb. 2D | Turbulent state | Zero state | Warmup evolution |
| Density | Gaussian blobs | Gaussian blobs | — |

## F.7. Hardware Configuration

Experiments were executed on a local workstation equipped with an Intel(R) Core(TM) Ultra 9 275HX processor, featuring 24 physical cores and a maximum clock speed of 5.4 GHz. High-speed parallel computations for the PDE solvers and neural networks were offloaded to an NVIDIA GeForce RTX 5090 Laptop GPU with 24 GB of dedicated GDDR7 VRAM and a peak SM clock rate of 3090 MHz.

## F.8. Software Environment

The computational environment was hosted on Windows Subsystem for Linux (WSL2) running Ubuntu 22.04 with Linux Kernel 6.6.87. We utilized the JAX framework for high-performance numerical computing and hardware acceleration. The software stack versioning is summarized in Table 10.

*Table 10.* Software stack and library versions used for evaluation.

| Component | Local Workstation |
|-----------|-------------------|
| OS | Ubuntu 22.04 (WSL2) |
| Linux Kernel | 6.6.87 |
| NVIDIA Driver | 581.57 |
| JAX Backend | CUDA (GPU) |
| JAX Version | 0.8.1 |
| Optimization | Optax (latest) |
| Job Scheduler | N/A |

## G. Baseline Reinforcement Learning Benchmarks

To evaluate the proposed framework, we benchmarked it against standard continuous control reinforcement learning (RL) and multi-agent reinforcement learning (MARL) algorithms. This section details the shared architectures, algorithm implementations, and environment-specific configurations used across all baseline comparisons. For a complete list of training hyperparameters, please refer to Appendix F.

## G.1. Baseline Algorithms and Network Architectures

We evaluated multiple algorithms, spanning on-policy, off-policy, centralized, decentralized, and differentiable predictive control paradigms to benchmark the 1D Heat Equation environment. Unless otherwise specified, all actor and critic networks share a common backbone: a Multi-Layer Perceptron (MLP) with two hidden layers of 256 units and ReLU activations. To maintain representational stability under large spatiotemporal state variations, the final hidden representation $x$ undergoes a soft $L_2$ normalization trick: $x = x/(||x||_2 + 1.0)$ prior to the output heads. Action outputs are bounded using $\texttt{tanh}$ activations scaled by the environment's physical limits.

- **Centralized RL (TD3 & PPO):** These baselines treat the swarm as a single entity. The actor observes the flattened global state (current PDE state, target state, and all agent coordinates) and outputs the joint action space for all agents simultaneously. The critics evaluate the global state to predict a single joint $Q(s, a)$ for TD3 or a global state-value $V(s)$ for PPO. PPO actors additionally learn a state-independent $\log(\sigma)$ standard deviation parameter for exploration.

- **Decentralized MARL (MATD3 & MAPPO):** These baselines follow the Centralized Training with Decentralized Execution (CTDE) paradigm. Decentralized actors share weights and process strictly localized observations (a concatenation of a localized spatial patch of the error field and a Positional Encoding of their coordinates) to output individual actions. The centralized critics observe the full global state to compute joint $Q$-values (MATD3) or individual, agent-specific state-values $V_i(s)$ (MAPPO) to solve spatial credit assignment.

- **Differentiable Predictive Control (DPC):** As a non-RL baseline, we utilize a DPC framework. Unlike the RL agents that learn via a learned value function and trial-and-error, the DPC policy exploits the exact gradients of the differentiable JAX-based PDE solver. The network is trained directly via Backpropagation Through Time (BPTT) by unrolling the dynamics over $T$ steps and minimizing the trajectory loss.

## G.2. Hyperparameters: Fisher-KPP 1D Environment

Table 11 details the specific network inputs, optimization schedules, and algorithm-specific hyperparameters utilized for the Fisher-KPP 1D benchmark.

For spatial state dimensions, the global state representation consists of the current PDE state ($N_{grid} = 100$), the target PDE state ($N_{grid} = 100$), and all agent coordinates ($N_{agents} = 20$), totaling 220 dimensions. The joint action space encompasses 40 dimensions (2 actions $\times$ 20 agents). Decentralized actors observe a local spatial patch combined with a Sinusoidal Positional Encoding (PE).

*Table 11.* Hyperparameters and Architectural Details for the FKPP 1D Benchmark

| Algorithm | Actor Input | Critic Input | Learning Rate (Actor / Critic) | Batch Size | Alg-Specific Hyperparameters |
|---|---|---|---|---|---|
| **DPC** | Global | N/A | Exp. Decay ($10^{-3} \rightarrow 0$, step 2k) | 32 | Updates: 500 Epochs, Grad Clip: 1.0 Loss Weights: Track (5.0), Bound (100.0) |
| **Centralized TD3** | Global | Global + Joint Act | Constant ($10^{-4}$ / $5 \times 10^{-4}$) | 256 | Updates: 100k, Buffer: 125k, Grad Clip: 1.0 $\tau = 0.005$, Policy Delay: 2, Warmup: 500 |
| **Centralized PPO** | Global | Global | Linear Decay ($3 \times 10^{-4}$ / $10^{-3}$) | 1024 | Rollout: 128, PPO Epochs: 4, Grad Clip: 0.5 Clip $\epsilon = 0.2$, Ent. Coef: 0.01, VF Coef: 0.5 |
| **MATD3 (MARL)** | Local Patch + PE | Global + Joint Act | Linear Decay ($10^{-4}$ / $5 \times 10^{-4}$) | 1024 | Updates: 100k, Buffer: 125k, Grad Clip: 1.0 $\tau = 0.005$, Policy Delay: 2, Warmup: 500 |
| **MAPPO (MARL)** | Local Patch + PE | Global | Linear Decay ($3 \times 10^{-4}$ / $3 \times 10^{-4}$) | 1024 | Rollout: 300, PPO Epochs: 4, Grad Clip: 0.5 Clip $\epsilon = 0.2$, Ent. Coef: 0.01, VF Coef: 0.5 |

## G.3. Hyperparameters: Heat 1D Environment

Table 12 details the specific network inputs, optimization schedules, and algorithm-specific hyperparameters utilized for the Heat 1D benchmark.

For the Heat 1D task, the swarm consists of fewer agents ($N_{agents} = 8$). Consequently, the flattened global state representation encompasses 208 dimensions (current PDE state $N_{grid} = 100$, target PDE state $N_{grid} = 100$, and 8 agent coordinates). The centralized joint action space spans 16 dimensions (2 actions $\times$ 8 agents). As with the FKPP setup, decentralized MARL actors observe a localized spatial patch augmented with a PE.

*Table 12.* Hyperparameters and Architectural Details for the Heat 1D Benchmark

| Algorithm | Actor Input | Critic Input | Learning Rate (Actor / Critic) | Batch Size | Alg-Specific Hyperparameters |
|---|---|---|---|---|---|
| **DPC** | Global | N/A | Exp. Decay ($10^{-3} \to 0$, step 2k) | 32 | Updates: 500 Epochs, Grad Clip: 1.0 |
| | | | | | Loss Weights: Track (5.0), Bound (100.0), Accel (0.1) |
| **Centralized TD3** | Global | Global + Joint Act | Constant ($10^{-4}$ / $5 \times 10^{-4}$) | 256 | Updates: 100k, Buffer: 125k, Grad Clip: 1.0 |
| | | | | | $\tau = 0.005$, Policy Delay: 2, Warmup: 500 |
| **Centralized PPO** | Global | Global | Linear Decay ($3 \times 10^{-4}$ / $10^{-3}$) | 1024 | Rollout: 128, PPO Epochs: 4, Grad Clip: 1.0 |
| | | | | | Clip $\epsilon = 0.2$, Ent. Coef: 0.01, VF Coef: 0.5 |
| **MATD3 (MARL)** | Local Patch + PE | Joint Local Obs + Joint Act | Linear Decay ($10^{-4}$ / $5 \times 10^{-4}$) | 256 | Updates: 100k, Buffer: 125k, Grad Clip: 1.0 |
| | | | | | $\tau = 0.005$, Policy Delay: 2, Warmup: 500 |
| **MAPPO (MARL)** | Local Patch + PE | Global | Linear Decay ($3 \times 10^{-4}$ / $3 \times 10^{-4}$) | 1024 | Rollout: 300, PPO Epochs: 4, Grad Clip: 0.5 |
| | | | | | Clip $\epsilon = 0.2$, Ent. Coef: 0.01, VF Coef: 0.5 |

## G.4. Hyperparameters: Heat 2D Environment

Table 13 outlines the network architectures, optimization configurations, and algorithm-specific hyperparameters for the Heat 2D benchmark.

Due to the increased spatial complexity of the 2D grid ($32 \times 32$), the flattened global state representation expands significantly to 2,080 dimensions (1,024 current state, 1,024 target state, and $16 \times 2$ agent coordinates). The centralized joint action space spans 48 dimensions ($[u, v_x, v_y]$ per agent). Decentralized MARL actors observe a localized 2D spatial patch combined with a 2D PE.

*Table 13.* Hyperparameters and Architectural Details for the Heat 2D Benchmark

| Algorithm | Actor Input | Critic Input | Learning Rate (Actor / Critic) | Batch Size | Alg-Specific Hyperparameters |
|---|---|---|---|---|---|
| **DPC** | Global | N/A | Exp. Decay ($10^{-3} \to 0$, step 2k) | 16 | Updates: 500 Epochs, Grad Clip: 1.0 |
| | | | | | Loss Weights: Track (5.0), Coll (20.0), Accel (0.1) |
| **Centralized TD3** | Global | Global + Joint Act | Constant ($10^{-4}$ / $5 \times 10^{-4}$) | 256 | Updates: 100k, Buffer: 125k, Grad Clip: 1.0 |
| | | | | | $\tau = 0.005$, Policy Delay: 2, Warmup: 500 |
| **Centralized PPO** | Global | Global | Constant ($3 \times 10^{-4}$ / $10^{-3}$) | 1024 | Rollout: 128, PPO Epochs: 4, Grad Clip: 0.5 |
| | | | | | Clip $\epsilon = 0.2$, Ent. Coef: 0.01, VF Coef: 0.5 |
| **MATD3 (MARL)** | Local Patch + PE | Joint Local Obs + Joint Act | Constant ($10^{-4}$ / $5 \times 10^{-4}$) | 256 | Updates: 100k, Buffer: 125k, Grad Clip: 1.0 |
| | | | | | $\tau = 0.005$, Policy Delay: 2, Warmup: 500 |
| **MAPPO (MARL)** | Local Patch + PE | Global | Linear Decay ($3 \times 10^{-4}$ / $3 \times 10^{-4}$) | 1024 | Rollout: 128, PPO Epochs: 4, Grad Clip: 0.5 |
| | | | | | Clip $\epsilon = 0.2$, Ent. Coef: 0.01, VF Coef: 0.5 |

## G.5. Hyperparameters: Heat 2D Environment (with Obstacles)

Table 14 outlines the network architectures, optimization configurations, and algorithm-specific hyperparameters for the Heat 2D benchmark with static obstacles.

Operating on a $32 \times 32$ spatial grid, the flattened global state representation encompasses 2,080 dimensions (1,024 current state, 1,024 target state, and $16 \times 2$ agent coordinates). The centralized joint action space spans 48 dimensions ($[u, v_x, v_y]$ per agent). Decentralized MARL actors observe a localized 2D spatial patch combined with a 2D PE. All models incorporate steep penalties for breaching the safety radius ($R_{safe\_obstacle} = 0.04$) of the three stationary obstacles.

*Table 14.* Hyperparameters and Architectural Details for the Heat 2D (Obstacles) Benchmark

| Algorithm | Actor Input | Critic Input | Learning Rate (Actor / Critic) | Batch Size | Alg-Specific Hyperparameters |
|---|---|---|---|---|---|
| **DPC** | Global | N/A | Exp. Decay ($10^{-3} \to 0$, step 2k) | 16 | Updates: 500 Epochs, Grad Clip: 1.0 |
| | | | | | Loss Weights: Track (5.0), Coll-Agent (20.0), Coll-Obs (100.0) |
| **Centralized TD3** | Global | Global + Joint Act | Constant ($10^{-4}$ / $5 \times 10^{-4}$) | 256 | Updates: 100k, Buffer: 125k, Grad Clip: 1.0 |
| | | | | | $\tau = 0.005$, Policy Delay: 2, Warmup: 500 |
| **Centralized PPO** | Global | Global | Constant ($3 \times 10^{-4}$ / $10^{-3}$) | 1024 | Rollout: 128, PPO Epochs: 4, Grad Clip: 0.5 |
| | | | | | Clip $\epsilon = 0.2$, Ent. Coef: 0.01, VF Coef: 0.5 |
| **MATD3 (MARL)** | Local Patch + PE | Joint Local Obs + Joint Act | Constant ($10^{-4}$ / $5 \times 10^{-4}$) | 256 | Updates: 100k, Buffer: 125k, Grad Clip: 1.0 |
| | | | | | $\tau = 0.005$, Policy Delay: 2, Warmup: 500 |
| **MAPPO (MARL)** | Local Patch + PE | Global | Linear Decay ($3 \times 10^{-4} \to 0$ / $3 \times 10^{-4} \to 0$) | 1024 | Rollout: 128, PPO Epochs: 4, Grad Clip: 0.5 |
| | | | | | Clip $\epsilon = 0.2$, Ent. Coef: 0.01, VF Coef: 0.5 |

## G.6. Hyperparameters: Kuramoto-Sivashinsky (KS) 1D Environment

Table 15 outlines the network architectures, optimization configurations, and algorithm-specific hyperparameters utilized for the highly chaotic Kuramoto-Sivashinsky (KS) 1D benchmark.

In this stabilization task, 8 static actuators are distributed across a spatial domain of $N_{grid} = 128$. For the centralized RL baselines, the models observe only the current dimensional PDE state, with the actor outputting an 8-dimensional joint action vector. The DPC baseline observes an augmented 264-dimensional global state (current state, target state, and fixed actuator coordinates). Notably, to maintain stability against the rapidly diverging KS dynamics, the off-policy RL algorithms utilize significantly lower learning rates compared to other environments, and observation states are scaled by a normalization factor of 5.0.

*Table 15.* Hyperparameters and Architectural Details for the KS 1D Benchmark

| Algorithm | Actor Input | Critic Input | Learning Rate (Actor / Critic) | Batch Size | Alg-Specific Hyperparameters |
|---|---|---|---|---|---|
| DPC | Global | N/A | Exp. Decay ($10^{-3} \to 0$, step 2k) | 32 | Updates: 500 Epochs, Grad Clip: 1.0 |
| | | | | | Loss Weights: Track (10.0), Effort (0.001) |
| Centralized TD3 | Global | Global + Joint Act | Constant ($10^{-6}$ / $5 \times 10^{-5}$) | 512 | Updates: 100k, Buffer: 200k, Grad Clip: 1.0 |
| | | | | | $\tau = 0.005$, Policy Delay: 2, Warmup: 500 |
| Centralized PPO | Global | Global | Constant ($3 \times 10^{-4}$ / $10^{-3}$) | 1024 | Rollout: 128, PPO Epochs: 4, Grad Clip: 0.5 |
| | | | | | Clip $\epsilon = 0.2$, Ent. Coef: 0.01, VF Coef: N/A |
| Indep. TD3 (MARL) | Local Patch + PE + $\mu$ | Local Obs + PE + Local Act | Constant ($10^{-6}$ / $5 \times 10^{-5}$) | 512 | Updates: 100k, Buffer: 125k, Grad Clip: 1.0 |
| | | | | | $\tau = 0.005$, Policy Delay: 2, Warmup: 500 |
| MAPPO (MARL) | Global + Pos (1) | Global | Linear Decay ($3 \times 10^{-4} \to 0$ / $3 \times 10^{-4} \to 0$) | 1024 | Rollout: 128, PPO Epochs: 4, Grad Clip: 0.5 |
| | | | | | Clip $\epsilon = 0.2$, Ent. Coef: 0.01 |

## G.7. Hyperparameters: Kuramoto-Sivashinsky (KS) 2D Environment

Table 16 details the specific network inputs, optimization schedules, and algorithm-specific hyperparameters utilized for the highly complex Kuramoto-Sivashinsky (KS) 2D benchmark.

This environment scales the stabilization task to a $64 \times 64$ spatial grid with a dense array of $N_{agents} = 100$ static actuators. Consequently, the flattened spatial state spans 4,096 dimensions, and the joint action space encompasses 100 dimensions. The DPC baseline observes an augmented 8,392-dimensional global state (current state, target state, and fixed 2D actuator coordinates). To maintain stability against the rapidly diverging 2D chaotic dynamics, the off-policy RL algorithms utilize significantly reduced learning rates, and all RL/MARL state observations are scaled down by a normalization factor of 5.0.

*Table 16.* Hyperparameters and Architectural Details for the KS 2D Benchmark

| Algorithm | Actor Input | Critic Input | Learning Rate (Actor / Critic) | Batch Size | Alg-Specific Hyperparameters |
|---|---|---|---|---|---|
| DPC | Global | N/A | Cosine Decay ($10^{-3} \to 10^{-5}$, warmup 50) | 4 | Updates: 500 Epochs, Grad Clip: 1.0 |
| | | | | | Loss Weights: Track (100.0), Effort ($10^{-4}$) |
| Centralized TD3 | Global | Global + Joint Act | Constant ($10^{-6}$ / $5 \times 10^{-5}$) | 128 | Updates: 100k, Buffer: 12.5k, Grad Clip: 1.0 |
| | | | | | $\tau = 0.005$, Policy Delay: 2, Warmup: 500 |
| Centralized PPO | Global | Global | Linear Decay ($10^{-4} \to 0$ / $10^{-3} \to 0$) | 1600 | Rollout: 50, PPO Epochs: 4, Grad Clip: 0.5 |
| | | | | | Clip $\epsilon = 0.2$, Ent. Coef: 0.001, Target KL: 0.05 |
| Indep. TD3 (MARL) | Local Patch + PE + $\mu$ | Local Obs + PE + Local Act | Constant ($10^{-6}$ / $5 \times 10^{-5}$) | 128 | Updates: 100k, Buffer: 12.5k, Grad Clip: 1.0 |
| | | | | | $\tau = 0.005$, Policy Delay: 2, Warmup: 500 |
| MAPPO (MARL) | Local Patch + PE | Global | Linear Decay ($3 \times 10^{-4} \to 0$ / $3 \times 10^{-4} \to 0$) | 1600 | Rollout: 50, PPO Epochs: 4, Grad Clip: 0.5 |
| | | | | | Clip $\epsilon = 0.2$, Ent. Coef: 0.01 |

## G.8. Hyperparameters: Turbulence 2D Environment

Table 17 details the specific network inputs, optimization schedules, and algorithm-specific hyperparameters utilized for the decaying Turbulence 2D benchmark.

To preserve the spatial alignment between the 2D fluid dynamics ($64 \times 64$ grid) and the 64 static actuators (arranged in an $8 \times 8$ grid), all centralized baselines (DPC, TD3, PPO) replace standard Multi-Layer Perceptrons with Fully Convolutional Networks (FCNs). The actors downsample the 4,096-dimensional global vorticity field directly into an $8 \times 8 \times 1$ spatial action grid. Centralized critics utilize Global Average Pooling (GAP) to compute global values.

For the decentralized MARL baselines, the local observation is expanded into a $20 \times 20 \times 3$ tensor patch containing the local vorticity alongside its spatial gradients ($\nabla_x, \nabla_y$). This patch is flattened and concatenated with a PE before passing

through the dense layers. Due to the extreme chaos of the system, state observations across all off-policy RL algorithms are divided by a normalization factor of 50.0.

*Table 17.* Hyperparameters and Architectural Details for the Turbulence 2D Benchmark

| Algorithm | Actor Input | Critic Input | Learning Rate (Actor / Critic) | Batch Size | Alg-Specific Hyperparameters |
|---|---|---|---|---|---|
| DPC | Global (FCN) | N/A | Warmup Cosine ($10^{-4} \rightarrow 5 \times 10^{-4} \rightarrow 10^{-6}$) | 4 | Updates: 500 Epochs, Grad Clip: 1.0 |
| | | | | | Loss Weights: Enstrophy (1.0), Effort ($10^{-5}$) |
| Centralized TD3 | Global (FCN) | Global + Spatial Act (FCN) | Cosine Decay ($10^{-4} \rightarrow 10^{-5} / 10^{-4} \rightarrow 10^{-5}$) | 128 | Updates: 100k, Buffer: 100k, Grad Clip: 1.0 |
| | | | | | $\tau = 0.005$, Policy Delay: 2, Warmup: 500 |
| Centralized PPO | Global (FCN) | Global (FCN) | Cosine Decay ($10^{-5} \rightarrow 0 / 10^{-3} \rightarrow 0$) | 1600 | Rollout: 150, PPO Epochs: 4, Grad Clip: 0.5 |
| | | | | | Clip $\epsilon = 0.1$, Ent. Coef: 0.001, Target KL: 0.05 |
| Indep. TD3 (MARL) | Local Patch + PE | Local Patch + PE + Act | Constant ($3 \times 10^{-5} / 3 \times 10^{-5}$) | 512 | Updates: 50k, Buffer: 100k, Grad Clip: 1.0 |
| | | | | | $\tau = 0.005$, Policy Delay: 2, Warmup: 500 |
| Indep. PPO (MAPPO) | Local Patch + PE | Local Patch + PE | Linear Decay ($3 \times 10^{-4} \rightarrow 0 / 3 \times 10^{-4} \rightarrow 0$) | 400 | Rollout: 150, PPO Epochs: 4, Grad Clip: 0.5 |
| | | | | | Clip $\epsilon = 0.2$, Ent. Coef: 0.01 |

## G.9. Hyperparameters: Density 2D Environment (Navier-Stokes)

Table 18 outlines the network architectures, optimization configurations, and algorithm-specific hyperparameters for the Density 2D benchmark.

In this environment, a swarm of $N_{agents} = 9$ mobile injectors operates over a $64 \times 80$ fluid grid. The flattened global state encompasses 10,258 dimensions (5,120 current density, 5,120 target density, and 18 agent coordinates). The centralized joint action space spans 18 dimensions ($[v_x, v_y]$ per agent).

*Table 18.* Hyperparameters and Architectural Details for the Density 2D Benchmark

| Algorithm | Actor Input | Critic Input | Learning Rate (Actor / Critic) | Batch Size | Alg-Specific Hyperparameters |
|---|---|---|---|---|---|
| DPC | Global | N/A | Exp. Decay ($10^{-3} \rightarrow 0$, step 2k) | 4 | Updates: 1000 Epochs, Grad Clip: 1.0 |
| | | | | | Loss Weights: Track (10.0), Coll (1000.0) |
| Centralized TD3 | Global | Global + Joint Act | Constant ($10^{-4} / 5 \times 10^{-4}$) | 256 | Updates: 100k, Buffer: 50k, Grad Clip: 1.0 |
| | | | | | $\tau = 0.005$, Policy Delay: 2, Warmup: 500 |
| Centralized PPO | Global | Global | Linear Decay ($10^{-4} \rightarrow 0 / 5 \times 10^{-4} \rightarrow 0$) | 1600 | Rollout: 50, PPO Epochs: 4, Grad Clip: 0.5 |
| | | | | | Clip $\epsilon = 0.2$, Ent. Coef: 0.001, Target KL: 0.05 |
| MATD3 (MARL) | Global + One-Hot ID (10,267) | Global + Joint Act | Constant ($10^{-4} / 5 \times 10^{-4}$) | 256 | Updates: 100k, Buffer: 50k, Grad Clip: 1.0 |
| | | | | | $\tau = 0.005$, Policy Delay: 2, Warmup: 500 |
| MAPPO (MARL) | Global Obs Stack | Global | Linear Decay ($3 \times 10^{-4} \rightarrow 0 / 3 \times 10^{-4} \rightarrow 0$) | 1600 | Rollout: 50, PPO Epochs: 4, Grad Clip: 0.5 |
| | | | | | Clip $\epsilon = 0.2$, Ent. Coef: 0.001 |

This section provides detailed hyperparameter specifications for all experiments. These tables supplement the problem descriptions and results already presented in the main appendix sections.

## G.10. Nonlinear MPC (NMPC) Baseline

To contextualize the performance of our neural operator policies against classical optimal control, we also benchmarked a Nonlinear Model Predictive Control (NMPC) algorithm (Allgöwer & Zheng, 2012) on the 1D Kuramoto-Sivashinsky (KS) stabilization task. The NMPC solves a finite-horizon open-loop optimal control problem at each time step, explicitly utilizing the differentiable KS physics solver.

Table 19 details the tracking performance and computational timing of the NMPC baseline. While the NMPC achieves competitive stabilization performance (Final Window MSE of $1.37 \times 10^{-4}$), it highlights the fundamental limitation of online optimization for PDE control: computational latency. The solver requires an average of 592.09 ms per step (reaching up to 1.7 seconds during complex chaotic transients). In multi-agent or 2D settings, this online computational cost scales poorly, rendering real-time control intractable. In contrast, our learned neural operator policy evaluates in a fraction of a millisecond, demonstrating the critical advantage of amortizing the optimization cost during offline training.

*Table 19.* NMPC Performance and Timing Statistics for the 1D KS Equation

| Parameter / Metric | Mean | Std. Dev. |
|---|---|---|
| *System Setup* | | |
| Samples Evaluated | 10 | |
| Rollout Steps | 400 | |
| Prediction Horizon | 20 | |
| *Control Performance* | | |
| Final Window MSE | $1.3797 \times 10^{-4}$ | $4.0648 \times 10^{-5}$ |
| *Timing Statistics (ms)* | | |
| Per-step MPC Solve Time[a] | 592.09 | 27.91 |
| Full Rollout Time | 237105.59 | |

[a] Optimization only. The absolute minimum solve time was 558.96 ms, and the maximum was 1730.05 ms.

