# OpenReview forum: "CINOC: Cardinality-Invariant Neural Operator Policies for Scalable PDE Control"
_ICML.cc/2026/Conference — ICML 2026 regular_

### Official Review · Reviewer_G9b3 · 2026-03-12

**Soundness:** 3
**Presentation:** 3
**Significance:** 3
**Originality:** 2
**Overall Recommendation:** 5
**Confidence:** 3

**Summary:**

The authors present a method to learn shared, cardinality-invariant PDE control policies. The policies are represented by neural operators that transform local observations into actuator functions, including the actuation and, optionally, velocity vectors for moving actors. The paper includes a theoretical analysis of the cardinality-invariance property as well as empirical results underscoring its suitability for PDE control.

**Compliance With Llm Reviewing Policy:**

Affirmed.

**Final Justification:**

A good paper overall, and the rebuttal was satisfactory in that it helped clarify many of the questions that I posed. I am thus increasing my score.

**Key Questions For Authors:**

1. Figure 2: The figure shows multiple neural operator policies, one for each agent. However, in line 163, you state: “$\mathcal{G}_\theta$ is the policy which is shared between the agents”. Why are there multiple policies shown in Figure 2?
2. Page 2: “Although preliminary MARL studies (...) exhibit zero-shot scalability, they achieve this via architectural mechanisms such as parameter sharing, GNNs, and equivariance constraints.” How is this parameter sharing different from the parameter sharing of the neural operator policy $\mathcal{G}_\theta$? Since the operator policy is provided with the actuator position $\mathbf{\xi}$, how does this relate to positional encoding in multi-agent RL [2]?

**Limitations:**

Yes, but the impact statement is missing.

**Strengths And Weaknesses:**

**Strengths**
- The proposed method offers a general framework for neural operator-based control policies in multi-agent settings with diverse observation and actuation schemes.
- The authors provide a theoretical framework to prove the convergence of policy gradient in the limit of infinite population sizes.
- The approach is validated empirically across a diverse range of PDEs.

**Weaknesses**
- The claim “Policies trained for a specific swarm size cannot generalize to different population sizes, and agents struggle to coordinate effectively when restricted to sparse local observations” (lines 065-059) is not generally true for multi-agent RL, see [3,4] as counter-examples. Additionally, I was not able to find any text passage referring to limitations of RL in terms of varying numbers of agents and transfer of policies in reference [1], which has been cited by the authors after the claim.
- No control baselines were considered in the experiments, e.g., MARL [3,4], DPC (Drgona et al., 2023), or MPC. Since the proposed method has access to the gradients, especially gradient-based control methods should be considered as baselines. While results show substantial improvements over uncontrolled baselines, it is hard to assess how other methods perform in comparison to the proposed method.
- The problem formulation (Section 3) and methodology (Section 4) are highly entangled. The underlying control problem, however, is more general since the simulation expects a vector of control actions (or a vector of functions evaluated at time step $t$ as stated in Eq. (4)). The way this vector is computed could be part of the methodology. Similarly, the state of each agent could be the output of the $\mathcal{Obs}$ function. This would also allow for other control paradigms, e.g., RL or MPC. With the current problem statement including $\mathcal{G}_\theta$ as part of the problem, the methodology is already part of the problem statement, as the general control problem could also be approached in different ways.

**Minor issues:**
- Format and consistency of references can be improved. For instance, the reference “Hu, H. and Liu, C. Safe pde boundary control with neural operators. arXiv preprint arXiv:2411.15643, 2024.” should be replaced by the PMLR version of the paper.
- “More details about the ablations in Appendix 6.5.” (lines 392-393): This should perhaps refer to Appendix C.

[1] Y. Pan, A. Farahmand, M. White, S. Nabi, P. Grover, and D. Nikovski, “Reinforcement Learning with Function-Valued Action Spaces for Partial Differential Equation Control,” in Proceedings of the 35th International Conference on Machine Learning, ICML 2018, Stockholmsmässan, Stockholm, Sweden, July 10-15, 2018, J. G. Dy and A. Krause, Eds., in Proceedings of Machine Learning Research, vol. 80. PMLR, 2018, pp. 3983–3992. [Online].

[2] J. Jeon, J. Rabault, J. Vasanth, F. Alcántara-Ávila, S. Baral, and R. Vinuesa, “Inductive biased-deep reinforcement learning methods for flow control: Group-invariant and positional-encoding networks improve learning reproducibility and quality,” Physics of Fluids, vol. 37, no. 7, p. 077189, Jul. 2025, doi: 10.1063/5.0276738.

[3] L. Guastoni, J. Rabault, P. Schlatter, H. Azizpour, and R. Vinuesa, “Deep reinforcement learning for turbulent drag reduction in channel flows,” Eur. Phys. J. E, vol. 46, no. 4, p. 27, Apr. 2023, doi: 10.1140/epje/s10189-023-00285-8.

[4] J. Vasanth, J. Rabault, F. Alcántara-Ávila, M. Mortensen, and R. Vinuesa, “Multi-agent Reinforcement Learning for the Control of Three-Dimensional Rayleigh–Bénard Convection,” Flow, Turbulence and Combustion, Dec. 2024, doi: 10.1007/s10494-024-00619-2.

---

> ### Author Rebuttal · Authors · 2026-03-31
>
> Dear Reviewer G9b3,
>
> We thank the reviewer for their detailed feedback. We have addressed the comments regarding the literature, baselines, problem formulation, and minor issues as follows:
>
> **1. Literature and Claims**
> We appreciate the pointers to [3, 4] and acknowledge that certain MARL architectures (such as those using GNNs and parameter sharing) do indeed exhibit zero-shot scalability.
> * **Correction of Claims:** We will refine our statement in lines 59–65 to be more precise regarding the specific architectural limitations of standard RL.
> * **Citations:** We will integrate references [3, 4] into our Related Works section to properly contextualize our method alongside state-of-the-art MARL for fluid dynamics. Furthermore, we thank the reviewer for pointing out the citation error regarding Pan et al., 2018 and we will remove the misattributed claim regarding RL population transfer from this citation.
>
> **2. Control Baselines**
> To provide a more rigorous evaluation, we have expanded our experimental comparison to include two centralized control baselines: Model Predictive Control (MPC) and Differentiable Predictive Control (DPC) along the MARL and RL methods that reviewer 65b7 was suggesting.
> * **MPC:** We solved the KS 1D stabilization problem using interior point optimization (IPOPT) across 10 initial conditions, achieving an average loss of $\mathcal{O}(10^{-4})$ at a computational cost of ~592 ms per optimization step. However, we note that scaling MPC to the KS 2D environment at a $128 \times 128$ resolution proved computationally prohibitive, strongly highlighting the scalability advantage of our neural operator approach.
> * **DPC:** Our method outperforms DPC on the FKPP 1D, Heat 2D (with obstacles), and KS 2D benchmarks, achieving between a 50% and an order-of-magnitude reduction in MSE. In the remaining cases, the two approaches yield comparable results, with DPC performing slightly better on the Heat 1D and Density Matching tasks.
> A detailed description of the entire experimental setup will been added to the Appendix, while a discussion of the result will be part of the main text.
>
> **3. Disentangling Problem and Methodology**
> We agree with the reviewer that the underlying control problem is more general than our specific architectural solution. To address this, we have performed a major revision of Sections 3 and 4:
> * **Section 3 (Problem Formulation):** This section now defines the PDE control problem in a purely agnostic, continuous-time framework. It focuses strictly on the state field $z$, the action vector $\bm{a}^M(t)$, and the physics-based cost, completely removing any mention of the neural operator ($\mathcal{G}_{\bm{\theta}}$) or DeepONet.
> * **Section 4 (Methodology):** This section now introduces our specific choice of a Neural Operator Policy as one possible solver for the general problem defined in Section 3. This restructuring clearly communicates that the general control problem could also be approached via other paradigms, such as RL or MPC.
>
> **4. Responses to Key Questions (Figure 2, MARL, and Positional Encoding)**
> * **Multiple policies in Figure 2 (Q1):** We apologize for the visual confusion. Figure 2 is intended to illustrate the *same* shared neural operator policy ($\mathcal{G}_{\bm{\theta}}$) being evaluated at different agent spatial coordinates. It represents a single set of weights evaluated across multiple agents, not multiple independent policies. We will updated the figure caption to make this explicit.
> * **Parameter Sharing and Positional Encoding vs. MARL (Q2):** While there is no fundamental difference in the mechanics of parameter sharing itself, the distinction lies in the use of spatial coordinates. In standard MARL, positional encoding is typically used to break permutation symmetry or to compute relative distances for message routing between discrete agents. In our framework, we treat the policy as a continuous control function. Providing the spatial coordinate $\bm{\xi}_i$ is not merely an encoding but acts as a sampling point on a continuous operator. This distinction is what enables true cardinality invariance: adding an agent simply means evaluating the control function at an additional spatial coordinate, seamlessly moving closer to a continuous distribution of control in the mean-field limit.
>
> **5. Minor Issues and Impact Statement**
> We will implement all of the reviewer's minor corrections:
> * We updated the Hu & Liu reference to the official PMLR version.
> * We corrected the typo in lines 392–393 to correctly point to "Appendix C" instead of "Appendix 6.5".
> * We have added a dedicated "Broader Impact Statement" section to the manuscript.
>
> Sincerely,
>
> Authors

---

> > ### Author Rebuttal · Reviewer_G9b3 · 2026-04-03
> >
> > Thank you for the detailed replies. I am improving m score accordingly.

---

> > > ### Author Response · Authors · 2026-04-06
> > >
> > > Dear Reviewer G9b3,
> > >
> > > Thank you so much for reviewing our rebuttal and for letting us know that your concerns have been resolved. We truly appreciate your feedback and your willingness to improve your score!
> > >
> > > We did notice, however, that the overall rating on the OpenReview platform does not seem to have changed on our end yet (we still see an overall rating of 4). If you still intend to raise it, could you kindly double-check that the updated score was saved and submitted in the system?
> > > Thank you again for your time and for helping us improve this paper!
> > >
> > > Sincerely,
> > >
> > > Authors

---

### Official Review · Reviewer_egYG · 2026-03-13

**Soundness:** 2
**Presentation:** 2
**Significance:** 3
**Originality:** 3
**Overall Recommendation:** 5
**Confidence:** 4

**Summary:**

This work reformulates PDE control problem as an operator learning problem. A shared neural operator policy (implemented with DeepONet) maps the current PDE state field to a continuous control function of agent locations, enabling decentralized multi-agent actuation under local sensing. The central claim is the cardinality invariance of the policy. The authors' architecture choice ensure if they train the system for ​$M_{train}$ agents, then it can be deployed with different swarm sizes and configurations.

**Compliance With Llm Reviewing Policy:**

Affirmed.

**Final Justification:**

The authors have satisfactorily resolved my concerns.

**Key Questions For Authors:**

1. The formulation in Eq 1 is not very clear. Please introduce the definition of each operator involved, its domain and range, and a motivation for each sub-equation within this section. Only a few definitions are included. Here are my specific questions: Is $v_i = \dot{\xi_i}$, if yes, why write $\dot{\xi_i}$ on the LHS and include $v_i$ in the LHS. How is $z_i^{patch}$ related to $z_i$? I think there is a typo on sub-equation 5 of Eq 1, should $v_i$ be included, it is not according to Eq 3. $c$ is left undefined. Define $\mathcal{D_{prob}}$, you claim $\psi \sim \mathcal{D_{prob}}$, including initial conditions $z_0$, target fields $z_{target}$, and specific constraint sets, what does the last point mean? Define $\mathcal{Z}$ here, instead of introducing without reference in subsection Neural Operator Policy. Also in Line 218, I believe the domains are switched.
2. Please be consistent, for $z$, in some places you use $(t)$ to show time dependence, and elsewhere, you use $t$ in the subscript, where the subscript $i$ should denote the agent number. Similarly for $\xi$.
3. What is $z_{target}$? Please define the variables pertinent to the goal or the problem statement before presenting them. I would recommend defining the problem statement after Eq 1.
4. Typo line above Eq 4: discretized, not discretization.
5. Why is this true, usually it is not: "Notably, our formulation remains compatible with the optimize-then-discretize (OtD) framework"?
6. Eq 5 typo: absolute value instead of norm (see eq 6, you have norm there)
7. Line 224: give reference for Aubin-Lions lemma, and also specify what it is.
8. For all the numerical experiments, please provide $\mathcal{F}$, or the PDE. Which parameter acts as the control input? If you already have in the appendix, then please refer to them.

**Limitations:**

Yes

**Strengths And Weaknesses:**

Strengths:

Empirically, the evidence is quite strong and is a major selling point. The paper evaluates across a diverse range of 1D/2D PDEs and objectives: tracking (Heat, Fisher–KPP), stabilization (Kuramoto–Sivashinsky, 2D turbulence), and density transport, reporting large controlled-vs-uncontrolled gains in a unified table.  Beyond performance, the experiments test the claimed invariances, zero-shot transfer across swarm sizes, robustness to sensor/actuator noise and partial failure, robustness to grid-resolution and parameter mismatch, and qualitative demonstrations of coordinated multi-phase strategies.

Weaknesses:
The method is formulated for a fairly broad, for a “generic PDE”. But the paper does not precisely delineate which PDE classes (e.g., parabolic vs. hyperbolic dynamics, well-posedness/regularity regimes, admissible nonlinearities) are expected to be controllable under the proposed approach. While Appendix A.1 restricts the theory to linear, dissipative operators, even in that setting the paper does not provide PDE-specific controllability/observability conditions.

The paper also does not articulate classical PDE-control conditions, such as actuator placement/coverage requirements or other  assumptions, that would clarify when stabilization or tracking should be feasible versus fundamentally impossible. Likewise, although agents act under local/partial sensing, the paper does not provide explicit observability guarantees (or sufficient conditions) ensuring that the chosen local measurements are adequate for the control objectives in general.

These are admittedly difficult questions and may be beyond the paper’s scope if the primary contribution is empirical scalability. Still, it would be helpful to see practical “rules of thumb” to guide deployment and interpretability.

---

> ### Author Rebuttal · Authors · 2026-03-31
>
> Dear Reviewer egYG,
>
> We thank the reviewer for their detailed feedback. We have carefully revised the manuscript to clarify our formulations and scope.
>
> **1. Delineation of Applicable PDE Classes**
> We appreciate the suggestion to more precisely define the scope of our method. As noted, our empirical evaluation spans multiple PDE classes: parabolic systems (Heat, Fisher-KPP), higher-order dissipative and chaotic systems (Kuramoto-Sivashinsky), and mixed advection-diffusion/fluid dynamics (Navier-Stokes). We will update the main text to explicitly state that our current focus and theoretical intuition are restricted to these dissipative and advective regimes. Extending this approach to strictly hyperbolic systems, which often develop shocks and require specialized numerical and control treatments, presents unique challenges and is an exciting direction for future work.
>
> **2. Controllability, Observability, and PDE-Control Conditions**
> Given the complexity of decentralized multi-agent control for nonlinear PDEs, providing explicit theoretical guarantees for observability and controllability is highly non-trivial. In this work, we operate under the assumption that the underlying systems are locally observable and controllable given the chosen agent densities, spatial distributions, and sensor radii. We will add a dedicated discussion paragraph in the revised manuscript explicitly stating these assumptions. We acknowledge that while our empirical results demonstrate practical success across diverse settings, defining rigorous sufficient conditions for actuator placement and local sensing adequacy remains an open theoretical challenge.
>
> **3. Practical "Rules of Thumb" for Deployment**
> We appreciate the reviewer’s desire for practical "rules of thumb" to guide deployment. While universal heuristics would be highly desirable for practitioners, the reality of deploying decentralized control across vastly different physical regimes, ranging from purely diffusive processes to chaotic, advective flows, is that optimal configurations are inherently dependent on the specific PDE and the downstream task. Because characteristic length scales, energy dissipation rates, and instability mechanisms vary so drastically between equations like Fisher-KPP and Navier-Stokes, a single set of deployment rules is currently out of reach. We plan to add a discussion to the Appendix explicitly addressing this limitation, noting that while our empirical cardinality invariance results are strong, deploying this framework on a novel PDE currently requires task-specific empirical tuning of agent density and sensing radii based on the domain's unique physical properties.
>
> **4. Compatibility with the Optimize-then-Discretize (OtD) Framework**
> We agree with the reviewer that standard deep reinforcement learning policies, which typically output fixed-dimensional arrays tied to a specific discretization, are strictly Discretize-then-Optimize (DtO). Our formulation bridges this gap because of how the control actions are parameterized. While the input sensing (the branch network of our DeepONet) relies on fixed sensor observations, the control output (via the trunk network) evaluates the action as a continuous function of spatial coordinates. Because the policy synthesizes the actuation field continuously, it natively interfaces with the continuous adjoint equations required for OtD. The optimization loop strictly requires a valid gradient signal ($\nabla_\theta J$). Whether this exact gradient is obtained by applying automatic differentiation through a discretized solver (DtO) or by solving continuous adjoint equations and projecting the resulting continuous functional gradients back through the trunk network (OtD), our operator policy can seamlessly ingest it.
>
> **5. Clarifications on Formulation, Notation, and Typos**
> We sincerely thank the reviewer for their meticulous reading of the mathematical formulations. We noted the typos and will correct them and we clarified the definitions in the revised manuscript as follows:
> * **Equation 1 & Problem Statement:** We have restructured the problem formulation to define the problem statement immediately following Eq. 1. We now explicitly define each operator, its domain, and its range.
> * **Other:** We will include experimental details, corrected the typos, enhance the notation consistency and add the appropriate reference for the Aubin-Lions lemma.
>
> Sincerely,
>
> Authors

---

> > ### Author Rebuttal · Reviewer_egYG · 2026-04-03
> >
> > I thank the authors for the detailed clarifications.
> >
> > The authors have satisfactorily responded to my questions. I am raising my score accordingly.

---

> > > ### Author Response · Authors · 2026-04-06
> > >
> > > Dear Reviewer egYG,
> > >
> > > Thank you for engaging with our rebuttal and for your meticulous feedback throughout this review process. We are happy that our clarifications successfully addressed your questions, and we sincerely appreciate your support and for raising your score!
> > >
> > > Sincerely,
> > >
> > > Authors

---

### Official Review · Reviewer_DUs5 · 2026-03-13

**Soundness:** 3
**Presentation:** 2
**Significance:** 3
**Originality:** 3
**Overall Recommendation:** 4
**Confidence:** 4

**Summary:**

This paper presents a scalable multi-agent neural network based PDE control method. First PDE control is formulated as an operator learning problem, similar to that in an RL framework, and then that operator is learned using neural networks. This neural network is trained by differentiating through the PDE solvers which inherently contains the dynamics, and this yields a truly scalable PDE control policy. This scalability is then mathematically formalized and explained using concepts rooted in mean-field theory. Authors validate their proposed framework on several tasks including stabilization, tracking and density transport.

**Compliance With Llm Reviewing Policy:**

Affirmed.

**Final Justification:**

I thank the authors for clarifying my concerns. As stated earlier, I maintain my current score, since the paper doesn't have a precise and well-defined problem formulation. I am also glad to learn that authors will incorporate the problem formulation in their updated version which I look forward to, and hope that they understand how critical it is to have a well-defined problem with preliminaries for any reviewer to evaluate any draft.

**Key Questions For Authors:**

Q1. Why do the agents share the same policy? Can you share some intuition behind that?
Q2. What are the theoretical consequences of sharing the same policy among agents?
Q3. What happens if the agents consider different policy for their own specified tasks, under the constraints of the PDE dynamics?

**Strengths And Weaknesses:**

Soundness: This paper is largely technically sound. From the literature survey to setup and problem formulation, the terms are well-defined and precise. Most of the conceptual components has been discussed in sufficient details, including appropriate references. What remains unclear is the cardinality invariant property. I see that differentiable solvers enable scalable gradient-based learning, however, how that scalability property is transferred when learning neural operators is not clear to me. It is also to be noted that the agents share the same policy, which does not sound intuitive at all in any MARL context. From what I understand so far, sharing a common policy is key in ensuring a scalable MARL PDE policy. There are also few terms that remain unclear, such as what is the dimension of $u_i(t), z_i^{patch}$? and what is individual agents' dynamics? While it is nice to have sufficient discussion on each of the components, there should be a section with preliminaries of PDE control and much of the constraints in formulation 1. It is not at all clear to me why the formulation is the way it is. One of the few tools a reviewer has, is precise problem formulation with sufficient preliminaries and details, which is lacking here.

Presentation: The paper reads smoothly. Grammar is correct. No typos in sight. It does not read like a paper which was written using LLMs, which is much appreciated. However, a lot of mathematical details which is required to understand the problem formulation, which reads like a very specific PDE setting, is not available in the draft. I think more details discussing various components in the formulation is left to be desired. Otherwise, the theoretical contribution will remain unclear.

Significance: Authors have done a great job highlighting how impactful this work is, in the Introduction and related works sections. This work learns a control policy which is scalable for large-scale complex systems. Each of the contributions are theoretically backed. It is also well-argued that this work differs significantly from existing PDE control works.
HOWEVER, some of the setups does not make sense, such as agents sharing the same control policy does not make sense at all. If I define a problem in such a way, that, it is just a matter of picking the right existing theory and show that it is scalable for such a formulation, does not paint a good picture. It also does not help that problem formulation is poorly written, with no preliminary material on each of the components of the formulation. It is highly confusing.

Originality: This is largely a novel work. Novel in terms of how it picks the right theories from the relevant areas and combine them to yield a much-needed control framework for PDE applications. I do not think the theoretical contributions are anything beyond using mean field theory to explain and formalize the scalability of the proposed operator learning problem.

---

> ### Author Rebuttal · Authors · 2026-03-31
>
> Dear Reviewer DUs5,
>
> We thank the reviewer for their careful readingand constructive feedback. We have revised the manuscript to clarify the mathematical formulations and provide deeper intuition for our design choices.
>
> **1. Intuition for Parameter Sharing and Heterogeneous Policies (Q1 & Q3)**
> The reviewer rightly points out that sharing a common policy is atypical in standard MARL, where agents often have distinct roles, varying capabilities, or competing objectives. However, in our framework and other works ([1], [2] and [3] for example), the agents form a homogeneous swarm deployed to cooperatively solve a single, unified task (controlling the macroscopic PDE field). Every agent possesses the exact same resources, actuation capabilities, and global objective.
>
> Because the task and resources are identical across the swarm, the optimal control strategy for any agent facing a specific local state is identical to that of any other agent in the same situation. Therefore, parameter sharing is not a mathematical convenience, but a logical reflection of the agents' homogeneous resources and unified objective. "Sharing a policy" simply ensures that identical agents with the exact same goal take the same action when presented with the same local conditions. We clarified this intuition in the main text.
>
> Regarding Q3: If the agents utilized fundamentally different policies for distinct tasks, it would imply they possess asymmetric resources, specialized hardware, or conflicting sub-tasks. Because they are collectively controlling a single, shared PDE environment, assigning heterogeneous policies breaks the permutation invariance of the swarm. This would force the control dimensionality to strictly depend on the number of agents, exponentially increasing the complexity of the learning problem and completely sacrificing the cardinality invariance that our shared-operator method achieves. Exploring such heterogeneous actuator networks is an interesting direction for future work.
>
> **2. Theoretical Consequences and Cardinality Invariance (Q2)**
> The primary theoretical consequence of parameter sharing is that it strictly decouples the policy's architecture and dimensionality from the swarm size $N$. Instead of learning a highly-coupled joint policy for $N$ discrete agents, sharing the policy allows us to lift the control problem into an infinite-dimensional, continuous functional space.
>
> We frame the solution as learning a single continuous Neural Operator that maps local PDE states to local control actions. The operator learns a response to the continuous physics of the underlying system rather than discrete agent behaviors and consequently the learned operator is fundamentally independent of the discrete number of agents; it essentially acts as a continuous control field that is then evaluated at discrete actuator locations. This is the foundation of the cardinality invariant property: during inference, the operator can be sampled at any spatial resolution, seamlessly controlling $M$ or $N$ agents with no retraining. We plan to add this intuition to the revised manuscript.
>
> **3. Clarifying Formulation 1 and Preliminaries**
> We agree that a precise problem formulation and self-contained preliminaries are essential. We will restructured the manuscript to include a dedicated "Preliminaries" section introducing the tools necessary to seamlessly follow our formulation (in particular theory behind function spaces and PDEs). We have also clarified the problem formulation as suggested.
>
>
> [1] Baker, Bowen, et al. "Emergent tool use from multi-agent autocurricula." International conference on learning representations. 2019.
> [2] Yu, Chao, et al. "The surprising effectiveness of ppo in cooperative multi-agent games." Advances in neural information processing systems 35 (2022)
> [3] M. Samvelyan, et al. "The StarCraft Multi-Agent Challenge". In Proceedings of the 18th International Conference on Autonomous Agents and MultiAgent Systems, AAMAS (2019)
>
> Sincerely,
>
> Authors

---

> > ### Author Rebuttal · Reviewer_DUs5 · 2026-04-03
> >
> > Thank you authors for clarifying my concerns. I maintain my current position, since the paper doesn't have a precise and well-defined problem formulation, which is a major mishap and flaw in the existing draft.

---

> > > ### Author Response · Authors · 2026-04-06
> > >
> > > Dear Reviewer DUs5,
> > >
> > > Thank you for reading our rebuttal and for your transparent feedback. We completely understand your position regarding the formulation in the current draft. Please rest assured that we are fully committed to addressing your valuable critique and that the final camera-ready version will feature the thoroughly restructured and precise problem formulation we outlined in our response, including the dedicated preliminaries section. We appreciate your time and your support for the paper!
> > >
> > > Sincerely,
> > >
> > > Authors

---

### Official Review · Reviewer_65b7 · 2026-03-13

**Soundness:** 3
**Presentation:** 3
**Significance:** 2
**Originality:** 2
**Overall Recommendation:** 4
**Confidence:** 2

**Summary:**

This paper proposes a Cardinality Invariant neural operator strategy for solving scalable multi-agent partial differential equation (PDE) control problems. The author overcame the bottleneck of traditional reinforcement learning strategies failing due to fixed dimensional representations when agent configuration changes by restructuring PDE control into an operator learning problem and using a differentiable PDE solver for end-to-end training. The effectiveness and robustness of the framework were verified through experiments on tracking, stabilization, and density transport tasks involving various PDEs such as linear, nonlinear, chaotic, and turbulent.

**Compliance With Llm Reviewing Policy:**

Affirmed.

**Final Justification:**

Since the authors solved my questions, I prefer to maintain my positive score.

**Key Questions For Authors:**

Can we consider supplementing the baseline with MARL algorithms based on CNN, attention mechanism, or GNN (if any), which also have permutation invariance and are compatible with input dimensions that are not fixed.

**Limitations:**

yes

**Strengths And Weaknesses:**

### Soundness

Strength: Provides rigorous mathematical proofs (such as proving the convergence of discrete policy gradients as unbiased estimators of underlying continuous operator gradients); Experimental validation on a wide range of complex systems, including 1D Fisher KPP, 2D Kuramoto Sivashinsky, and 2D turbulence control; Provided ablation experiments.

Weakness: The experimental results mainly demonstrate the comparison between Controlled and Natural evolution, lacking comparison with mainstream MARL baselines. Although the author partially explained the reasons for not choosing certain baselines (such as the discussion of MARL in the related work), further discussion is needed to determine whether each experiment truly cannot be compared with mainstream baselines.

### Presentation

The narrative logic of the article is clear. A better method diagram may be needed to demonstrate the advantages of using Neural Operator modeling.

### Significance
Strength: This study effectively solves the problem of retraining when changing hardware or agent population size in multi-agent PDE control

Weakness: This framework currently belongs to the category of Model-based, and its gradient calculation relies entirely on the underlying differentiable physics solver, which may limit its applicability.


### Originality
The application of neural operators belongs to mature technology, and the academic originality of this article is mainly reflected in the system level architecture integration and the discovery of the application of "cardinality invariance".

---

> ### Author Rebuttal · Authors · 2026-03-31
>
> Dear Reviewer 65b7,
>
> **Response to the Lack of Baselines**
>
> We thank the reviewer for highlighting the need for broader baseline comparisons. We have now implemented RL baselines across both single-agent (PPO, TD3) and multi-agent (MAPPO, MATD3) settings, utilizing centralized and decentralized observations, alongside a centralized version of Differentiable Predictive Control (DPC) (Drgona et al., 2023) as suggest by reviewer G9b3. Table 1 compares our method against the best-performing RL baseline for each environment.
>
> **Table 1: Our model vs. best RL baseline per environment. (Lower is better)**
>
> | Environment | Best RL Baseline | Ours |
> | :--- | :--- | :--- |
> | **Tracking (MSE)** | | |
> | FKPP 1D | 4.0e-3 (MAPPO) | **4.6e-5** |
> | Heat 1D | 8.0e-4 (MAPPO) | **2.9e-4** |
> | Heat 2D | 6.0e-3 (MAPPO) | **1.5e-4** |
> | Heat 2D (Obs) | 3.5e-3 (MAPPO) | **1.2e-4** |
> | **Stabilization (Energy/Enstrophy)** | | |
> | KS 1D | **1.283e-1** (MAPPO) | 1.288e-1 |
> | KS 2D | 5.0e-1 (MAPPO) | **3.79e-1** |
> | Turb. 2D | 3.71 (MATD3) | **3.23** |
> | **Density Matching (MML)** | | |
> | Density Ctrl | 5.1e-3 (PPO) | **1.9e-3** |
>
> Our model matches or outperforms the best RL baseline in 7 out of 8 environments. On tracking tasks, it achieves 1 to 2 orders of magnitude lower MSE. On KS 1D, MAPPO is marginally better (0.1283 vs. 0.1288), but this gap falls well within the standard deviation.
>
> Additionally, we supplemented our benchmarks with DPC, as suggested by Reviewer G9b3. Our method outperforms DPC on the FKPP 1D, Heat 2D (with obstacles), and KS 2D benchmarks, achieving between a 50% and an order-of-magnitude reduction in MSE. In the remaining cases, the two approaches yield comparable results.
>
> We also evaluated Nonlinear Model Predictive Control (NMPC) on KS 1D. While NMPC achieves a comparable error (1.38e-4), it requires ~592 ms per step, whereas our model only requires a single forward pass. Furthermore, we observed significant computational bottlenecks when attempting to scale NMPC to the KS 2D environment at a 128x128 resolution.
>
> **Response to Model-Based Limitations**
>
> We appreciate the reviewer raising this point, which we acknowledge in Section 7. However, we emphasize that this does not fundamentally restrict the applicability of our framework. When governing equations are unavailable, a differentiable surrogate learned from data can replace the physics solver without requiring any architectural changes to our model (Sarkar et al., 2025). Our core contribution—casting decentralized policy learning as an operator learning problem and proving that neural operator policies yield cardinality invariance—remains valid regardless of the gradient source.
>
> To further demonstrate this, the revision will include ablations where we fixed a DeepONet architecture but trained it using MAPPO and MATD3 instead of our differentiable pipeline. The results show that cardinality invariance is retained, confirming it is an inherent property of the architecture and that our setup is not strictly bound to gradient-based optimization.
>
> **Response to Permutation-Invariant MARL Baselines (Key Question)**
>
> Regarding the inclusion of Permutation-Invariant MARL (PI-MARL) algorithms with variable-size inputs, we surveyed several relevant methods: HPN (Jianye et al., 2022), UPDeT (Hu et al., 2021), SPECTra (Park et al., 2025), and MAT (Wen et al., 2022). While these methods achieve permutation invariance via attention or GNN layers over agent interaction graphs, a direct comparison faces two fundamental mismatches:
>
> 1. **Communication vs. Stigmergy:** The GNN and attention mechanisms in PI-MARL function as explicit inter-agent communication channels. In contrast, our framework assumes non-communicating agents. This reflects real-world scenarios where explicit communication is impractical. Adding communication layers changes the fundamental problem formulation to one with strictly more information, making a direct comparison inequitable.
> 2. **Domain Gap:** To our knowledge, no existing PI-MARL method has been successfully applied to continuous PDE control. Adapting them would require implementing continuous action spaces, PDE solver coupling, and spatially structured observations—a substantial research effort that extends well beyond a standard baseline comparison.
>
> We plan to expand our Related Work section to explicitly cite these methods and contrast them with our approach. We agree that integrating PI-MARL communication mechanisms into PDE control is a highly promising direction for future work, one that directly complements our current contributions.
>
> Sincerely,
>
> Authors

---

> > ### Author Rebuttal · Reviewer_65b7 · 2026-04-04
> >
> > Thank you authors for clarifying my concerns. I perfer to maintain my current positive position.

---

> > > ### Author Response · Authors · 2026-04-06
> > >
> > > Dear Reviewer 65b7,
> > >
> > > Thank you for taking the time to review our rebuttal and for confirming that your concerns have been adequately addressed. We appreciate your valuable time and feedback!
> > >
> > > Sincerely,
> > >
> > > Authors

---

### Decision · Program_Chairs · 2026-04-30

**Decision:**

Accept (regular)

**Comment:**

The paper studied multi-agent PDE control, with the objective of giving a control method that be scaled to more sensors and/or actuators (hence the "cardinality-invariant" in the title), unlike fixed-dimensional representations. This is posed as an operator learning problem with differentiable solvers used for training.

The reviewers appreciated the novel problem statement (from a RL viewpoint), and also the extensive coverage of experimental configurations in the empirical section. This paper received recommendations to accept from all reviewers (including two clear accepts).

Given this, our recommendation is to accept this paper.